# Lrig1-expression confers suppressive function to CD4$^+$ cells and is essential for averting autoimmunity via the Smad2/3/Foxp3 axis

Jae-Seung Moon[1,9,11], Chun-Chang Ho[1,10,11], Jong-Hyun Park [2,3], Kyungsoo Park[4], Bo-Young Shin[1,10], Su-Hyeon Lee [1], Ines Sequeira [5], Chin Hee Mun[6], Jin-Su Shin[1,10], Jung-Ho Kim[7], Beom Seok Kim [7], Jin-Wook Noh[7], Eui-Seon Lee[7], Ji Young Son[7], Yuna Kim[1], Yeji lee[7], Hee Cho[1], SunHyeon So[7], Jiyoon Park [1], Eunsu Choi[7], Jong-Won Oh [1], Sang-Won Lee[6], Tomohiro Morio [8], Fiona M. Watt [5], Rho Hyun Seong [4] & Sang-Kyou Lee [1,7] ✉

Regulatory T cells (T$_{reg}$) are CD4$^+$ T cells with immune-suppressive function, which is defined by Foxp3 expression. However, the molecular determinants defining the suppressive population of T cells have yet to be discovered. Here we report that the cell surface protein Lrig1 is enriched in suppressive T cells and controls their suppressive behaviors. Within CD4$^+$ T cells, T$_{reg}$ cells express the highest levels of Lrig1, and the expression level is further increasing with activation. The Lrig1$^+$ subpopulation from T helper (Th) 17 cells showed higher suppressive activity than the Lrig1$^-$ subpopulation. Lrig1-deficiency impairs the suppressive function of T$_{reg}$ cells, while Lrig1-deficient naïve T cells normally differentiate into other T cell subsets. Adoptive transfer of CD4$^+$Lrig1$^+$ T cells alleviates autoimmune symptoms in colitis and lupus nephritis mouse models. A monoclonal anti-Lrig1 antibody significantly improves the symptoms of experimental autoimmune encephalomyelitis. In conclusion, Lrig1 is an important regulator of suppressive T cell function and an exploitable target for treating autoimmune conditions.

Regulatory T cell (T$_{reg}$) is an essential population of T cells to suppress the immune responses[1,2] and is critical for peripheral tolerance to control autoimmune diseases[3,4]. However, the tumor development or progression is accelerated by the presence of T$_{reg}$ cells in the tumor microenvironment through inhibition of anti-tumor immune responses[5,6]. Foxp3 is a master transcription factor known to be

necessary for T$_{reg}$ cell differentiation, maturation, and suppressive functions in a TGF-β-dependent manner[1,7]. It has been reported that dysfunction of T$_{reg}$ cells, such as decreased number of T$_{reg}$ cells or their impaired suppressive activity, is a critical factor in the onset or maintenance of multiple sclerosis (MS), type 1 diabetes (T1D), rheumatoid arthritis (RA), and inflammatory bowel disease (IBD)[8–11]. And

[1]Department of Biotechnology, Yonsei University College of Life Science and Biotechnology, Seoul, Republic of Korea. [2]Center for Brain Disorders, Brain Science Institute, Korea Institute of Science and Technology, Seoul, Republic of Korea. [3]Division of Bio-Medical Science & Technology, KIST School, Korea University of Science and Technology, Seoul, Republic of Korea. [4]Department of Biological Sciences and Institute of Molecular Biology and Genetics, Seoul National University, Seoul, Republic of Korea. [5]Centre for Stem Cells and Regenerative Medicine, King's College London, Guy's Hospital, London, UK. [6]Division of Rheumatology, Department of Internal Medicine, Yonsei University College of Medicine, Seoul, Republic of Korea. [7]Good T cells, Inc., Seoul, Republic of Korea. [8]Department of Pediatrics and Developmental Biology, Graduate School of Medical and Dental Sciences, Tokyo Medical and Dental University (TMDU), Tokyo, Japan. [9]Present address: Division of Immunology and Rheumatology, Department of Medicine, Stanford University School of Medicine, Stanford, CA, USA. [10]Present address: Good T cells, Inc., Seoul, Republic of Korea. [11]These authors contributed equally: Jae-Seung Moon, Chun-Chang Ho. ✉e-mail: sjrlee@goodcells.co.kr

the previous study demonstrated that the proliferation of $T_{reg}$ cells is defective in patients with relapsing-remitting multiple sclerosis after T cell receptor (TCR) stimulation because of altered IL-2/STAT5 signaling, which caused the decreased expression of Foxp3[12–14]. Therefore, identifying the surface protein specific to $T_{reg}$ cells is a fundamental element to utilize the functional enhancement or the adoptive transfer of $T_{reg}$ cells for clinical applications.

Several surface proteins, including CD25, CTLA-4, CD127, and CD177, have been discovered as differentially expressed proteins on $T_{reg}$ cells[15–19]. CD25, the IL-2Rα chain, is the most common surface marker used for isolating $T_{reg}$ cells. CD4$^+$CD25$^+$ T cells are heterogeneous and contain suppressive subpopulations expressing Foxp3[20,21]. However, CD25 can also be expressed on activated effector T cells and memory subsets, and CD25$^-$ T cells can be highly suppressive Foxp3$^+$ $T_{reg}$ cells in particular experimental settings[22,23]. Further, to define human $T_{reg}$ cells, CD4$^+$CD25$^+$CD127$^-$ combination is generally used, and additional markers such as CD45RA or CD62L can be used to isolate the highly purified $T_{reg}$ cells[24–26]. CD177 is a glycosyl-phosphatidylinositol (GPI)-linked surface glycoprotein and is known to be highly expressed in neutrophils[27]. Recently, CD177 has been suggested as a surface molecule specifically expressed in a subpopulation of tumor infiltrated $T_{reg}$ cells[28]. However, all these known $T_{reg}$ cell-specific proteins are also highly expressed on the surface of various other pro-inflammatory T cell subsets under chronic antigen stimulation[29–33].

Furthermore, it has been reported that $T_{reg}$ cells exhibit plasticity properties, such as the expression of inflammatory cytokines associated with other T helper (Th) cells while maintaining Foxp3 expression in certain disease conditions[34–36]. In addition to $T_{reg}$ cells, recent studies have shown that Th17 cells can transdifferentiate into another suppressive T cell population during inflammation resolution or in a tumor microenvironment[37,38]. Therefore, a new surface molecule highly specific to a suppressive population of T cells, not other effector T cells, needs to be identified to provide new therapeutics to patients with autoimmunity.

In this study, Lrig1 is identified as a surface protein on the mouse and human $T_{reg}$ cells, whose Lrig1 expression is significantly higher than other T cell subsets or activated T cells. The suppressive activity of $T_{reg}$ cells is positively correlated with the expression of Lrig1 on the surface, and CD4$^+$Lrig1$^+$ T cells show a similar level of suppressive potential to CD4$^+$Foxp3$^+$ T cells. Moreover, we show that the Lrig1$^+$ subpopulation from Th17 cells shows a comparable level of suppressive potential to that of Lrig1$^+$ $T_{reg}$ cells. Agonistic monoclonal antibody (mAb) specific to Lrig1 or adoptive transfer of CD4$^+$Lrig1$^+$ T cells exhibits significant therapeutic efficacy in experimental autoimmune encephalomyelitis (EAE), IBD, and lupus animal models. This therapeutic potential through Lrig1 is mediated by the increase of Foxp3$^+$ T cells via induction of Smad2/3 phosphorylation leading to *Foxp3* expression at the transcription level. Therefore, Lrig1 on the surface of CD4$^+$ T cells is important for the suppressive population of Th17 and $T_{reg}$ cells in mice and humans, and Lrig1-targeting mAb or adoptive transfer of Lrig1$^+$ T cells can be new therapeutics for treating autoimmune diseases by targeting or regulating the immune-suppressive T cells.

## Results

### The expression of Lrig1 is enriched in the activated mouse and human regulatory T cells
Using next-generation sequencing and functional gene network analysis, we identified Lrig1 as a new surface protein enriched in $T_{reg}$ cells. As shown in Fig. 1a, b, the expression of Lrig1 at mRNA and Lrig1$^+$ T cell number were significantly higher in mouse $T_{reg}$ cells than that in other T cell subsets. And the level of Lrig1$^+$ cells was very low in CD19$^+$ immune cell populations, and CD4$^+$ T cells showed a higher level of Lrig1$^+$ population than CD8$^+$ T cells in

mouse mesenteric and inguinal lymph nodes (LN) (Supplementary Figs. 1 and 2a). Lrig1 protein on the surface of induced $T_{reg}$ (i$T_{reg}$) cells can be induced in a TGF-β1-dependent manner, while IL-2 didn't change Lrig1 expression on the CD4$^+$Foxp3$^+$ T cells (Fig. 1c and Supplementary Fig. 2b). Consistently, the level of LRIG1-positive cells in human i$T_{reg}$ cells was higher than that on other T cell subsets. However, the amount of LRIG1$^+$ human i$T_{reg}$ cells varies from 19.69% to 62.39% individually (Fig. 1d). The Rapamycin[39]- or Everolimus[40]-mediated expansion of human i$T_{reg}$ cells increased LRIG1 protein compared to normal i$T_{reg}$ cells (Supplementary Fig. 2c).

When the Lrig1-expressing cells among CD4$^+$Foxp3$^+$ T cells in the mouse lymphoid organs was examined, sub-populations of Foxp3$^+$ T cells from spleen, thymus, or inguinal LN expressed Lrig1 on the surface, respectively (Fig. 1e). CD4$^+$Lrig1$^+$ T cells expressed a higher level of immuno-suppressive markers, including PD-1, CD25, Foxp3, or IL-10, than CD4$^+$Lrig1$^-$ T cells (Fig. 1f). It has been reported that the activated $T_{reg}$ (a$T_{reg}$) cells have CD44$^{high}$ and CD62L$^{low}$ surface phenotypes[41]. We next examined the functional phenotype of CD4$^+$Lrig1$^+$ T cells in the spleen or inguinal LN. $T_{reg}$ cells in the activated status included a higher number of Lrig1$^+$ cells than those in the resting status (r$T_{reg}$) (Supplementary Fig. 3a). Consistently, a much higher level of a$T_{reg}$ cells can be found in the CD4$^+$Foxp3$^+$Lrig1$^{high}$ T cells than in the CD4$^+$Foxp3$^+$Lrig1$^{low}$ T cells from spleen or inguinal LN (Supplementary Fig. 3b). These data suggest that Lrig1 is a new surface protein enriched in the activated $T_{reg}$ cells, and its expression is substantially low in other T cell subsets and non-T immune cells.

### Lrig1 is functionally required for the suppressive subpopulation of CD4$^+$ T cells
To evaluate the suppressive potential of CD4$^+$Lrig1$^+$ T cells, the effector T cells were co-cultured with CD4$^+$Lrig1$^+$, CD4$^+$Lrig1$^-$ T cells, or CD4$^+$Foxp3$^+$ T cells as a positive control. As shown in Fig. 2a, the suppressive activity of CD4$^+$Lrig1$^+$ T cells was comparable to that of CD4$^+$Foxp3$^+$ T cells, and CD4$^+$Lrig1$^-$ T cells had lower suppressive activity than CD4$^+$Lrig1$^+$ T cells. In addition, CD4$^+$Lrig1$^+$ T cells had higher expression level of Foxp3, CD25, PD-1, or CTLA4 compared to CD4$^+$Lrig1$^-$ T cells (Supplementary Fig. 4). Interestingly, the secretion level of the immune-suppressive cytokines such as IL-10 from CD4$^+$Lrig1$^+$ T cells was higher than that of CD4$^+$Lrig1$^-$ T cells, or even CD4$^+$Foxp3$^+$ T cells (Fig. 2b).

To examine whether the presence of Lrig1 on the surface is important for the suppressive activity of Foxp3$^+$ T cells, CD4$^+$Foxp3$^+$ T cells isolated from Foxp3-IRES-GFP mice were separated into Lrig1$^{high}$ or Lrig1$^{low}$ T cell subpopulation. Although both CD4$^+$Foxp3$^+$Lrig1$^{high}$ and CD4$^+$Foxp3$^+$Lrig1$^{low}$ cells expressed a similar level of Foxp3, a substantially higher suppressive potential was shown in CD4$^+$Foxp3$^+$Lrig1$^{high}$ T cells than the CD4$^+$Foxp3$^+$Lrig1$^{low}$ T cell cells (Fig. 2c). As shown in Fig. 1b, when the naïve T cells were induced to differentiate into various T cell subsets, the slight induction of Lrig1 expression was also detected on the surface of activated T cells (Th0), Th1, and Th17 cells. To assess whether the expression of Lrig1 can also be functionally required for a suppressive subpopulation in other CD4$^+$ T cell subsets, the Lrig1$^+$ subpopulation was separated from Th0, Th1, or Th17 cells, and their suppressive activity was compared to that of each Lrig1$^-$ subpopulation. Lrig1$^+$ T cells from i$T_{reg}$ cells or Th17 cells, not from Th0 or Th1, showed higher suppressive activity than each Lrig1$^-$ subpopulation (Fig. 2d). Furthermore, when CD25$^+$ T cells or CD25$^-$CCR6$^+$ T cells from the CD4$^+$ T cells in mouse splenocytes were separated into Lrig1$^+$ and Lrig1$^-$ subpopulations, the suppressive potential of Lrig1$^+$ T cells in CD25$^-$CCR6$^+$ and CD25$^+$ T cells was significantly higher than that of CD25$^-$CCR6$^+$Lrig1$^-$ and CD25$^+$Lrig1$^-$ T cells, respectively (Fig. 2e). Consistently, the human LRIG1$^+$ $T_{reg}$ cells and LRIG1$^+$ Th17 cells showed higher immuno-suppressive activity than their LRIG1$^-$ populations (Fig. 2f). Therefore, Lrig1 on the surface of

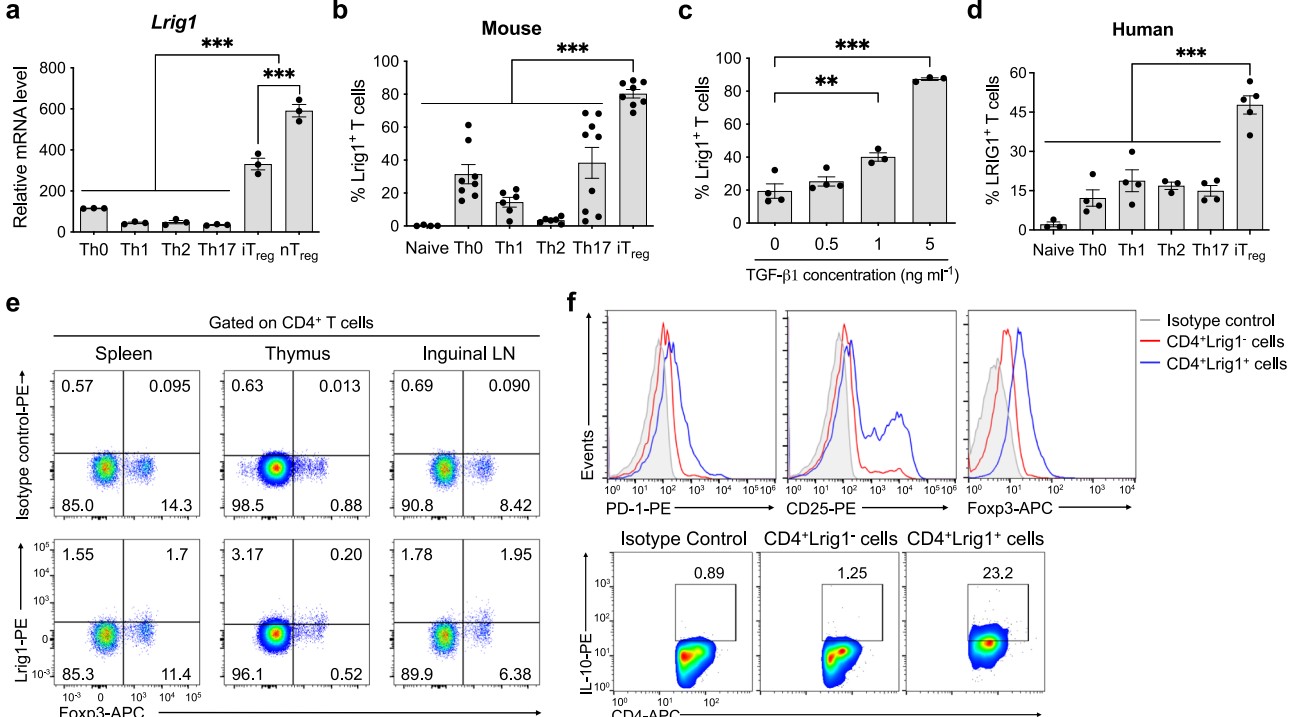

**Fig. 1 | Lrig1 is a new surface protein dominantly expressed in mouse and human T_reg cells. a** Relative mRNA amount of *Lrig1* in T_reg cells or different T cell subsets by qRT-PCR ($n = 3$). Data are expressed as mean ± S.E.M (***$P < 0.0001$). **b** The level of Lrig1 expressing cells among mouse T cell subsets. Naïve: $n = 4$, Th0: $n = 8$, Th1: $n = 6$, Th2: $n = 6$, Th17: $n = 9$, iT_reg: $n = 8$. Data are expressed as mean ± S.E.M (***$P < 0.0001$). **c** The level of Lrig1 expressing cells was analyzed when mouse naïve CD4⁺ T cells were cultured in iT_reg-polarizing condition with the different concentrations (0–5 ng ml⁻¹) of TGF-β1. 0 or 0.5 ng ml⁻¹: $n = 4$, 1, or 5 ng ml⁻¹: $n = 3$. Data are expressed as mean ± S.E.M (**$P = 0.0032$, ***$P < 0.0001$). **d** Examination of

human LRIG1 on the surface of various human T cell subsets. Naïve: $n = 3$, Th0: $n = 4$, Th1: $n = 4$, Th2: $n = 3$, Th17: $n = 4$, iT_reg: $n = 5$. Data are expressed as mean ± S.E.M (***$P < 0.0001$). **e** Representative images of frequencies of Lrig1⁺Foxp3⁺ T cells among mouse CD4⁺ T cells in the spleen, thymus, or inguinal lymph node. **f** The expression level of PD-1, CD25, Foxp3, or IL-10 in mouse splenic CD4⁺Lrig1⁺ or Lrig1⁻ T cells. Data are combined from at least three independent experiments. Statistics were calculated by ordinary one-way ANOVA with Tukey's multiple comparisons test (**a**) or ordinary one-way ANOVA with Dunnett's multiple comparisons test (**b**–**d**). Source data are provided as a Source Data file.

CD4⁺ T cells is an important surface protein for the suppressive population of T_reg cells and Th17 cells.

## Lack of *Lrig1* expression impairs the suppressive function of T_reg cells

To confirm the functional importance of Lrig1 for the suppressive activity of T_reg cells, the surface expression of Lrig1 on T_reg cells was knock-downed by *Lrig1*-targeting siRNA. *Lrig1*-targeting siRNA (si*Lrig1*) treatment effectively reduced the surface expression of Lrig1 on T_reg cells (Fig. 3a, Supplementary Fig. 5a). The considerable decrease in the suppressive potential of T_reg cells was observed by the reduction of Lrig1 expression on the T_reg cell surface (Fig. 3b) even though the Foxp3 level was not affected by the *Lrig1* silence during the iT_reg cell differentiation (Fig. 3c). However, the reduced expression of *Lrig1* did not influence the expression of other known T_reg cell markers, including Foxp3, GITR, or PD-1 (Supplementary Fig. 5b), and it did not alter T_reg cell proliferation by IL-2 (Supplementary Fig. 5c). These results demonstrate that the expression of Lrig1 on the surface of T_reg cells is necessary for their suppressive capacity.

Next, we further investigated the functions of Lrig1 on T_reg cells and other T cell subsets using *Lrig1*-knockout (KO) mice[42]. iT_reg cells from *Lrig1*⁻/⁻ mice showed a considerably lower level of suppressive potential than Lrig1-expressing iT_reg cells from *Lrig1*⁺/⁺ and *Lrig1*⁺/⁻ mice (Fig. 3d). Consistent with the results from the knockdown study, the difference in Foxp3, CD25, GITR, CTLA4, TIGIT, or PD-1 expression between iT_reg cells from *Lrig1*⁺/⁻ and *Lrig1*⁻/⁻ mice was not observed (Supplementary Fig. 6). We next examined whether the lack of *Lrig1* expression affects the differentiation program of naïve T cells into

various T cell subsets. As shown in Fig. 3e, the differentiation capability of *Lrig1*-deficient naïve T cells into Th1, Th2, Th17, or iT_reg cells was unaffected. Moreover, the spleen weight, the spleen's size, and the level of various T cell subsets in the spleen were unchanged (Fig. 3f and Supplementary Fig. 7a). The deficiency of *Lrig1* did not provoke the inflammatory condition in the lung, and the slight increase of CD4⁺CD44⁺, not CD4⁺CD62L⁺ T cells, in the spleen and the lung was observed, but it was not statistically meaningful (Supplementary Fig. 7b). The slight reduction of the dermal skin layer in *Lrig1*⁻/⁻ mice may be due to the previous finding[43] in which T_reg cells in the skin facilitate epithelial stem cell differentiation (Supplementary Fig. 7c). In addition, the populations of thymic T_reg (tT_reg) precursor cells and tT_reg cells were similar in *Lrig1*⁺/⁺ and *Lrig1*⁻/⁻ mice, and the proportion of CD4⁺Foxp3⁺ T cells did not differ between *Lrig1*⁺/⁺ and *Lrig1*⁻/⁻ mice in the spleen, thymus, and inguinal LN, suggesting *Lrig1*-deficiency does not affect thymic development of T_reg cells (Supplementary Fig. 8a, b). These results suggest that Lrig1 plays an important role in the suppressive functions of T_reg cells without influencing the differentiation capacity and functions of other T cell subsets in vivo.

## Adoptive transfer of CD4⁺Lrig1⁺ T cells into IBD or lupus animal model shows significant therapeutic potential

To confirm the therapeutic potential of CD4⁺Lrig1⁺ T cells, CD4⁺Foxp3⁺, CD4⁺Lrig1⁺ or CD4⁺Lrig1⁻ T cells were isolated from Foxp3-IRES-GFP or C57BL/6 mice (CD45.2⁺) and co-transferred with CD4⁺CD45RB^high T cells (CD45.1⁺) into *Rag1*⁻/⁻ mice. As shown in Fig. 4a–d, the adoptive transfer of CD4⁺Foxp3⁺ or CD4⁺Lrig1⁺ T cells markedly alleviated the IBD symptoms such as body weight loss, inflammation in the colon, splenomegaly, and histological score, while these IBD symptoms were

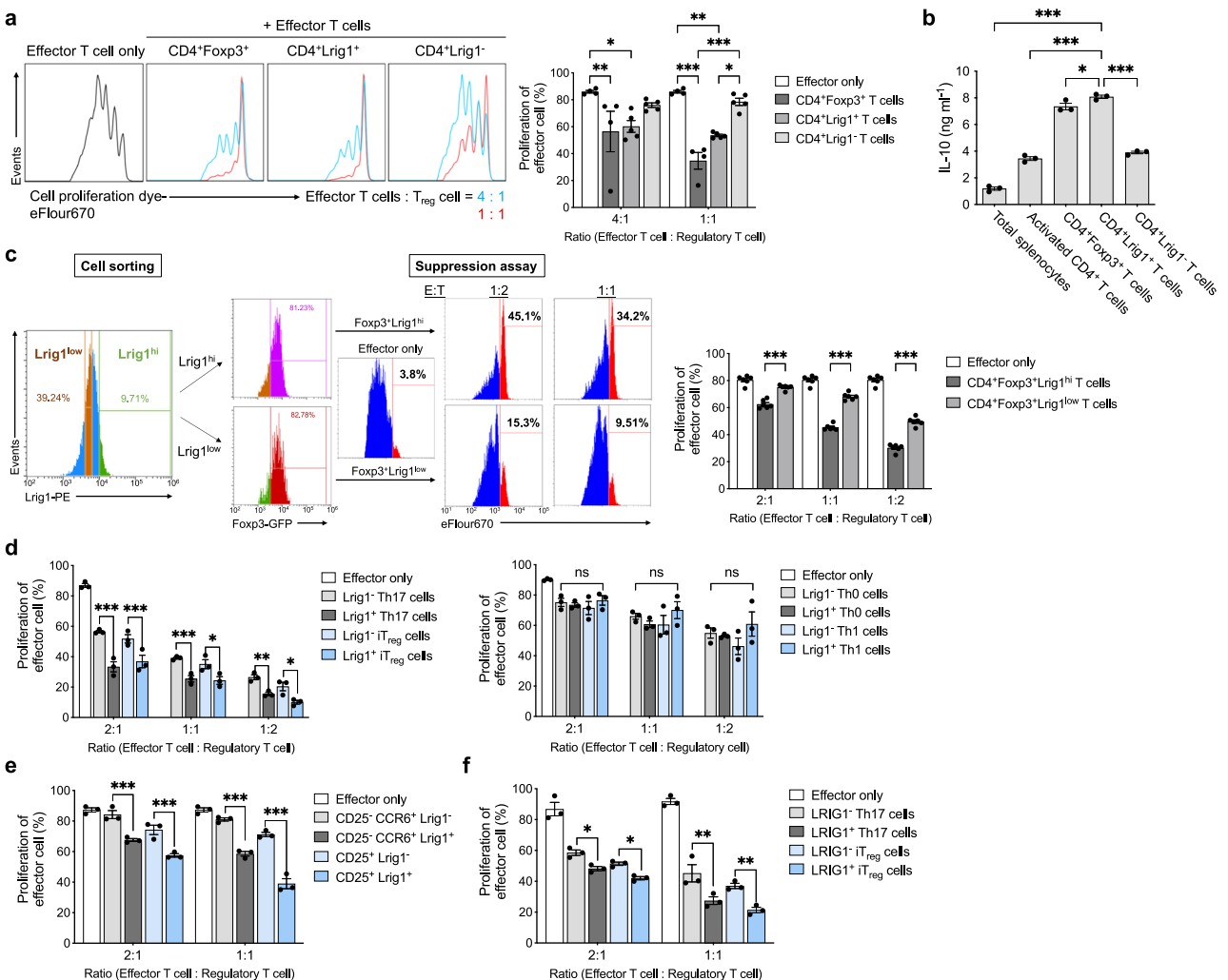

**Fig. 2 | Lrig1 is required for the suppressive population of T_reg and Th17 cells.**
**a** Suppressive potential of CD4⁺Lrig1⁺ cells, CD4⁺Lrig1⁻ cells, or CD4⁺GFP⁺ cells (Foxp3⁺) toward eFlour 670-labeled effector T cells with a different ratio. Effector only or CD4⁺Foxp3⁺ T cells: $n = 4$, CD4⁺Lrig1⁺ T cells and CD4⁺Lrig1⁻ T cells: $n = 5$. Data are expressed as mean ± S.E.M (in ratio 4:1 *$P = 0.02$, **$P = 0.007$; in ratio 1:1 *$P = 0.01$, **$P = 0.002$, ***$P < 0.001$). **b** The level of IL-10 in the culture medium of total splenocytes, activated CD4⁺ T cells, CD4⁺Foxp3⁺ T cells, or CD4⁺Lrig1⁺ or Lrig1⁻ T cells in mouse spleen ($n = 3$). Data are expressed as mean ± S.E.M (*$P = 0.03$, ***$P < 0.001$). **c** Representative images (left) and quantification (right) of suppression activity of CD4⁺Foxp3⁺Lrig1ʰⁱ or CD4⁺Foxp3⁺Lrig1ˡᵒʷ T cells expressing a similar level of Foxp3 ($n = 6$). Data are expressed as mean ± S.E.M (***$P < 0.001$). **d, e** Comparison of the suppressive activity of Lrig1⁺ and Lrig1⁻ cells from mouse

Th17, iT_reg, Th0, or Th1 cells (**d**) ($n = 3$) or mouse Lrig1⁺ and Lrig1⁻ cells from CD25⁻CCR6⁺ T cells or CD25⁺ T cells in the splenocytes (**e**) ($n = 3$). Data are expressed as mean ± S.E.M (in **d** 2:1 ratio, ***$P < 0.0001$ Lrig1⁻ Th17 vs Lrig1⁺ Th17, ***$P = 0.0003$ Lrig1⁻ iT_reg vs Lrig1⁺ iT_reg; 1:1 ratio, *$P = 0.012$, ***$P = 0.0007$; 1:2 ratio, *$P = 0.0127$, **$P = 0.0099$; in **e** ***$P < 0.0001$). **f** The suppressive potential of human LRIG1⁺ and LRIG1⁻ cells from human blood differentiated Th17 or iT_reg cells ($n = 3$). Data are expressed as mean ± S.E.M (*$P = 0.0479$ LRIG1⁻ Th17 vs LRIG1⁺ Th17, *$P = 0.0471$ LRIG1⁻ iT_reg vs LRIG1⁺ iT_reg, **$P = 0.0014$ LRIG1⁻ Th17 vs LRIG1⁺ Th17, **$P = 0.0032$ LRIG1⁻ iT_reg vs LRIG1⁺ iT_reg). Ordinary one-way ANOVA with Dunnett's multiple comparisons test (**b**) or two-way ANOVA with Tukey's multiple comparisons test (**a**, **c–f**). All experiments were repeated at least three times. Source data are provided as a Source Data file.

observed in CD4⁺Lrig1⁻ T cell recipients. When the immuno-cellular factors for the improvement of IBD symptoms were investigated, the number of Foxp3⁺ T cells in the CD4⁺CD45.2⁺ population was higher (Fig. 4e), and the level of TNFα or IL-1β in the serum was substantially lower in CD4⁺Foxp3⁺ or CD4⁺Lrig1⁺ T cell recipients than those in CD4⁺Lrig1⁻ T cell recipients (Fig. 4f). Furthermore, to investigate the effect of Lrig1 on the function of T_reg cells, *Lrig1⁺/⁻* or *Lrig1⁻/⁻* iT_reg cells were transferred into *Rag1⁻/⁻* mice. IBD symptoms were alleviated in mice receiving the *Lrig1⁺/⁻* iT_reg cell transfer, whereas not in mice with *Lrig1⁻/⁻* iT_reg cell transfer (Supplementary Fig. 9a–c). *Lrig1⁺/⁻* iT_reg-recipient mice had a higher level of CD4⁺Foxp3⁺ T cells in intraepithelial lymphocytes (IEL) of the colon and mesenteric lymph node (MLN) and decreased proportion of CD4⁺IFNγ⁺ T cells in both CD45.1⁺ and CD45.1⁻ cells (Supplementary Fig. 9d–f). Similar therapeutic efficacy was observed when CD4⁺Lrig1⁺ T cells were adoptively transferred into

lupus-prone (NZB/NZW) F1 mice. The therapeutic activity of CD4⁺Lrig1⁺ T cells was comparable to that of methylprednisolone, a clinically used lupus medication (Supplementary Fig. 10a–c).

We next sought to examine the transcriptional landscape of CD4⁺Lrig1⁺ T cells. Bulk RNA sequencing analysis of CD4⁺Lrig1⁺ or CD4⁺Lrig1⁻ T cells purified from the normal mouse splenocytes revealed that the genes associated with the suppressive functions of T_reg cells, including *Foxp3*, *Il2ra*, and *Ctla4*, were highly expressed in CD4⁺Lrig1⁺ T cells compared to CD4⁺Lrig1⁻ T cells (Fig. 4g). Interestingly, most of the immune-suppressive genes were included in the top 20 differentially expressed genes (DEG) among 468 upregulated DEGs in Lrig1⁺CD4⁺ T cells (Supplementary Fig. 11a). As shown in the Supplementary Fig. 11b, the DEGs expressed by the Lrig1⁺ T cells were involved in the upregulation of cytokine production and negative regulation of the immune

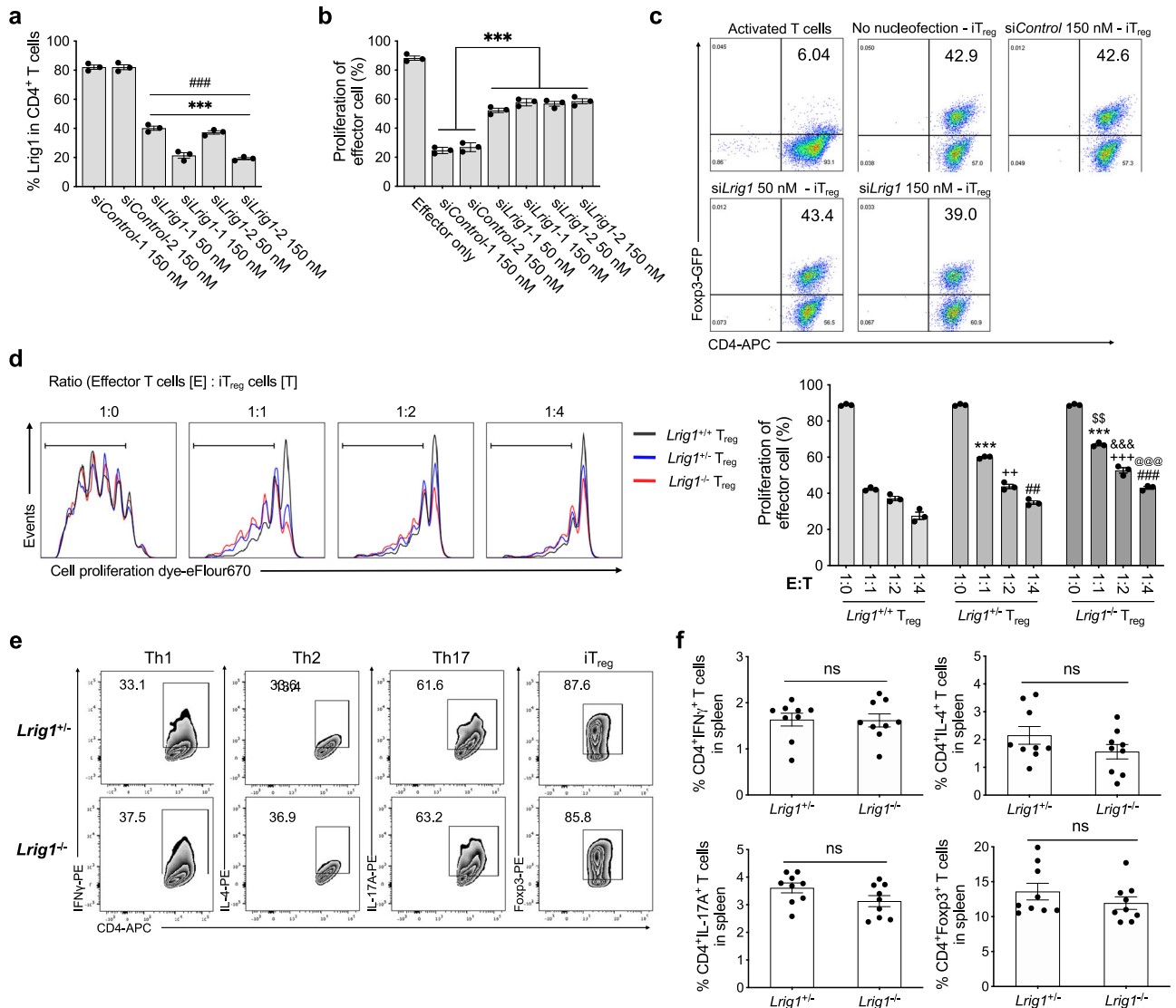

**Fig. 3 | *Lrig1*-deficiency in T_reg cells impairs the suppressive activity. a** The level of Lrig1 expression on the cell surface of iT_reg cells whose *Lrig1* expression was knocked down by 50 nM or 150 nM of 2 different siRNAs targeting *Lrig1* (si*Lrig1*) or 150 nM of scrambled sequences (si*Control*) (*n* = 3). Data are expressed as mean ± S.E.M (***P < 0.001 versus si*Control*-1, ###P < 0.001 versus si*Control*-2).
**b** Quantification of the suppressive activity of *Lrig1*-silenced iT_reg cells (*n* = 3) Data are expressed as mean ± S.E.M (***P < 0.0001). **c** si*Lrig1*- or si*Control*- nucleofected naïve CD4+ T cells from Foxp3-IRES-GFP mice were induced to differentiate into iT_reg cells, and the level of Foxp3 in CD4+Foxp3+ (GFP+) cells was examined.
**d** Representative images (left), and quantification (right) of the suppressive potential of iT_reg cells from *Lrig1*+/+, *Lrig1*+/−, or *Lrig1*−/− mice (*n* = 3). Data are expressed as mean ± S.E.M (***P < 0.001, ++P = 0.005, +++P < 0.001, ##P = 0.002,

###P < 0.001, $$P = 0.002, &&&P < 0.001, @@@P < 0.001). * symbol shows versus 1:1 of *Lrig1*+/+, + symbol shows versus 1:2 of *Lrig1*+/+, # symbol shows versus 1:4 of *Lrig1*+/+, $ symbol shows versus 1:1 of *Lrig1*+/−, & symbol shows versus 1:2 of *Lrig1*+/−, @ symbol shows versus 1:4 of *Lrig1*−/−. **e** Representative dot plots of the level of IFNγ, IL-4, IL-17A, or Foxp3 in each cell type differentiated from *Lrig1*+/− or *Lrig1*−/− mice. **f** Mouse splenocytes were isolated from *Lrig1*+/− or *Lrig1*−/− mice, and the level of IFNγ+, IL-4+, IL-17A+, or Foxp3+ in CD4+ T cells was examined (*n* = 9). Data are expressed as mean ± S.E.M (IFNγ: *P* = 0.9341, IL-4: *P* = 0.1721, IL-17A: *P* = 0.0934, Foxp3: *P* = 0.2866). Ordinary one-way ANOVA with Dunnett's multiple comparisons test (**a**, **b**), two-way ANOVA with Tukey's multiple comparisons test (**d**) or two-tailed unpaired Student's *t* test (**f**). All experiments were repeated at least three times. Source data are provided as a Source Data file.

system process (upper panel), while the upregulated genes in Lrig1− T cells were related to immunoglobulin production or lymphocyte activation (lower panel). Surprisingly, our results showed that the cell type with the gene set most similar to the upregulated genes in Lrig1+ T cell is FoxP3+ T cells (upper panel, Supplementary Fig. 12a). On the other hand, the cell type with the highest relation to the upregulated genes in Lrig1− T cell was B220+ B cells (lower panel), as shown in the Supplementary Fig. 12a. Moreover, using TRRUST[44] enrichment analysis, we demonstrated that the expression of the upregulated genes in Lrig1+ T cell are significantly regulated by Nkfb1, Rela, or Foxp3 which is associated with the expression of suppressive molecules in T_reg cells (Supplementary Fig. 12b). Furthermore, Gene Set

Enrichment Analysis (GSEA) suggested that CD4+Lrig1+ T cells were highly enriched for the signature genes associated with Foxp3 targets and suppressive functions of T cells (Fig. 4h). Therefore, CD4+Lrig1+ T cells are a therapeutically important subpopulation with high immuno-suppressive capacity.

**Antibody-mediated Lrig1 stimulation significantly alleviates the symptoms of EAE by induction of *Foxp3* transcription via phosphorylation of Smad2/3**

To investigate the intracellular signal transduction pathway through Lrig1 in T_reg cells, monoclonal antibody 6F01 specific to Lrig1 was generated, which stained mouse and human iT_reg cells effectively (Supplementary Fig. 13). When the naïve T cells were differentiated

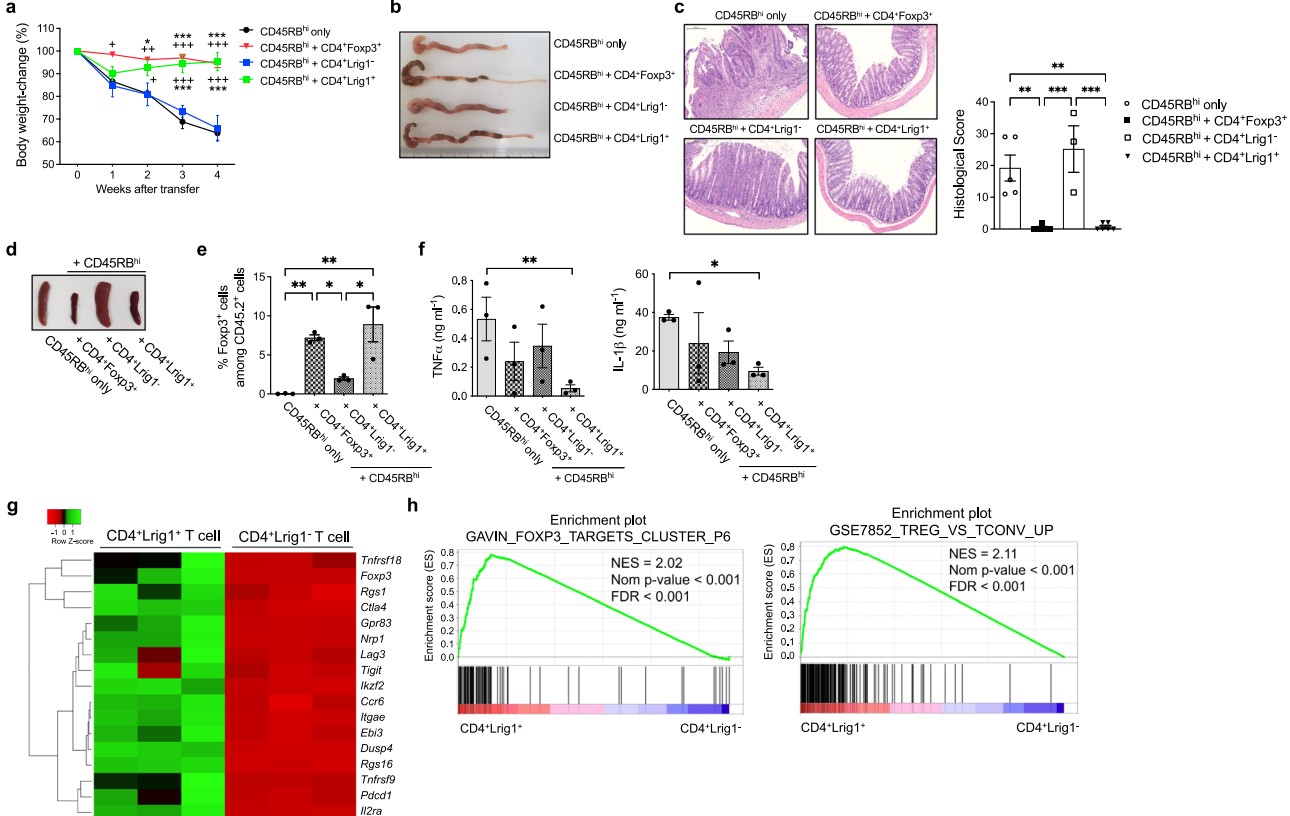

**Fig. 4 | Adoptive transfer of CD4+Lrig1+ T cells into IBD mice significantly alleviates the autoimmune symptoms. a** Body weight change of *Rag1*-deficient mice transferred with CD45.1+CD4+CD45RB^high T cell alone (*n* = 3), or together with CD4+Foxp3+ T cells from Foxp3-IRES-GFP mice (*n* = 5), CD4+Lrig1+ (*n* = 6) or Lrig1⁻ (*n* = 4) T cells from C57BL/6 mice. Data are expressed as mean ± S.E.M (week 1 +*P* = 0.0121; week 2 *P* = 0.0122, +*P* = 0.0336, ++*P* = 0.0042; ***P* < 0.0001, +++*P* < 0.0001). + symbol shows versus CD4+Lrig1⁻ T cells, * symbol shows versus CD4+CD45RB^high T cells only. **b** Representative image of macroscopic changes in the colon from each recipient group. **c** Representative images of hematoxylin and eosin (H&E) staining (left) and the combined histopathological clinical scores (right) of large intestines. Scale bar indicates 200 μm. CD45.1+CD4+CD45RB^high T cell alone (*n* = 5), or together with CD4+Foxp3+ T cells (*n* = 5), CD4+Lrig1+ (*n* = 6), or Lrig1⁻ (*n* = 3) T cells. Data are expressed as mean ± S.E.M (***P* = 0.0022 CD45RB^hi only vs + CD4+Foxp3+, ***P* = 0.0017 CD45RB^hi only vs + CD4+Lrig1+, ****P* = 0.0007 + CD4+Foxp3+ vs + CD4+Lrig1⁻, ***P* = 0.0005 + CD4+Lrig1⁻ vs + CD4+Lrig1+). **d** Representative images of splenomegaly from each recipient group.

**e** The level of CD4+Foxp3+ cells among CD45.2+TCRβ+ T cells in the re-stimulated lymphocytes from the colonic lamina propria of each recipient group (*n* = 3). Data are expressed as mean ± S.E.M (*P* = 0.0456 + CD4+Foxp3+ vs + CD4+Lrig1⁻, *P* = 0.0107 + CD4+Lrig1⁻ vs + CD4+Lrig1+, ***P* = 0.009 CD45RB^hi only vs +CD4+Foxp3+, ***P* = 0.0024 CD45RB^hi only vs +CD4+Lrig1+). **f** Serum concentration of TNFα or IL-1β in each recipient mice (*n* = 3). Data are expressed as mean ± S.E.M (*P* = 0.0347, ****P* = 0.0004). **g** Heatmap showing the expression level of suppressive signature genes in Lrig1+ or Lrig1⁻ cells from mouse splenic CD4+ T cells (*n* = 3). **h** GSEA plots comparing CD4+Lrig1+ and CD4+Lrig1⁻ T cell populations using the gene sets of Foxp3-target clusters and the upregulated genes in T_reg cells compared to conventional T cells (cgp.v.7.4.symbols and ImmuneSigDB.v.7.4.symbols). Data were collected from at least three independent experiments. Statistical significance was measured by two-way ANOVA with Tukey's multiple comparisons test (**a**), one-way ANOVA with Tukey's multiple comparisons test (**c**, **e**), and two-tailed unpaired Student's *t* test (**f**). Source data are provided as a Source Data file.

into iT_reg cells in the presence of 6F01 under the suboptimal concentration of TGF-β1 (0.5 ng ml⁻¹), 6F01 treatment significantly increased the number of Foxp3+ T cells close to the number induced by the optimal concentration of TGF-β1 (5 ng ml⁻¹; Fig. 5a). Interestingly, Lrig1 stimulation by 6F01 during Th17 cell differentiation also increased the number of Foxp3+ T cells and the level of Smad2/3 phosphorylation (Supplementary Fig. 14a, b). 6F01-mediated stimulation of Lrig1-expressing iT_reg cells substantially enhanced the phosphorylation of Smad2/3, leading to the increase of *Foxp3* expression at the transcription level analyzed by qRT-PCR (Fig. 5b, d). When Smad3 phosphorylation inhibitor (SIS3) was included in this experimental condition, the function of 6F01 to increase *Foxp3* expression was completely abolished (Fig. 5c). This result demonstrates that the increase of Smad2/3 phosphorylation accomplishes the increment of the Foxp3 level by 6F01 treatment.

In previous studies, Lrig1 has been known to be involved in EGFR degradation in cancer cells[45]. To further investigate the underlying molecular mechanism of 6F01-mediated induction of Foxp3 expression, we analyzed EGFR downstream signaling pathways in 6F01-treated iT_reg cells. The expression level of EGFR was reduced in 6F01-treated iT_reg cells (Fig. 5e and Supplementary Fig. 15), starting from 30 min after 6F01 stimulation, and the phosphorylation of AKT (p-AKT) or mTOR (p-mTOR), which are downstream molecules of EGFR signaling[46], was also inhibited by 6F01 treatment in a dose-dependent manner (Fig. 5f, g). We observed p-AKT starts to dephosphorylate very quickly by 6F01 treatment within 1 min after stimulation (Supplementary Fig. 16). Although no treatment or isotype IgG also induced dephosphorylation of AKT because the transferred iT_reg cells could not receive the activation signals through T cell receptor stimulated with anti-CD3e/28 antibodies or cytokines, 6F01 treated-iT_reg cells showed the quicker dephosphorylation of AKT than no or isotype treatment. On the other hand, the non-phosphorylated form of AKT or mTOR expression was not affected by 6F01 treatment (Supplementary Fig. 17a, b). FoxO1, a transcription factor known to induce *Foxp3* expression, can be exported from the nucleus by p-AKT-induced phosphorylation[47,48]. The phosphorylation of FoxO1 (p-FoxO1) was also suppressed by 6F01 treatment, which inhibits the phosphorylation of AKT (Fig. 5h and Supplementary Fig. 17c).

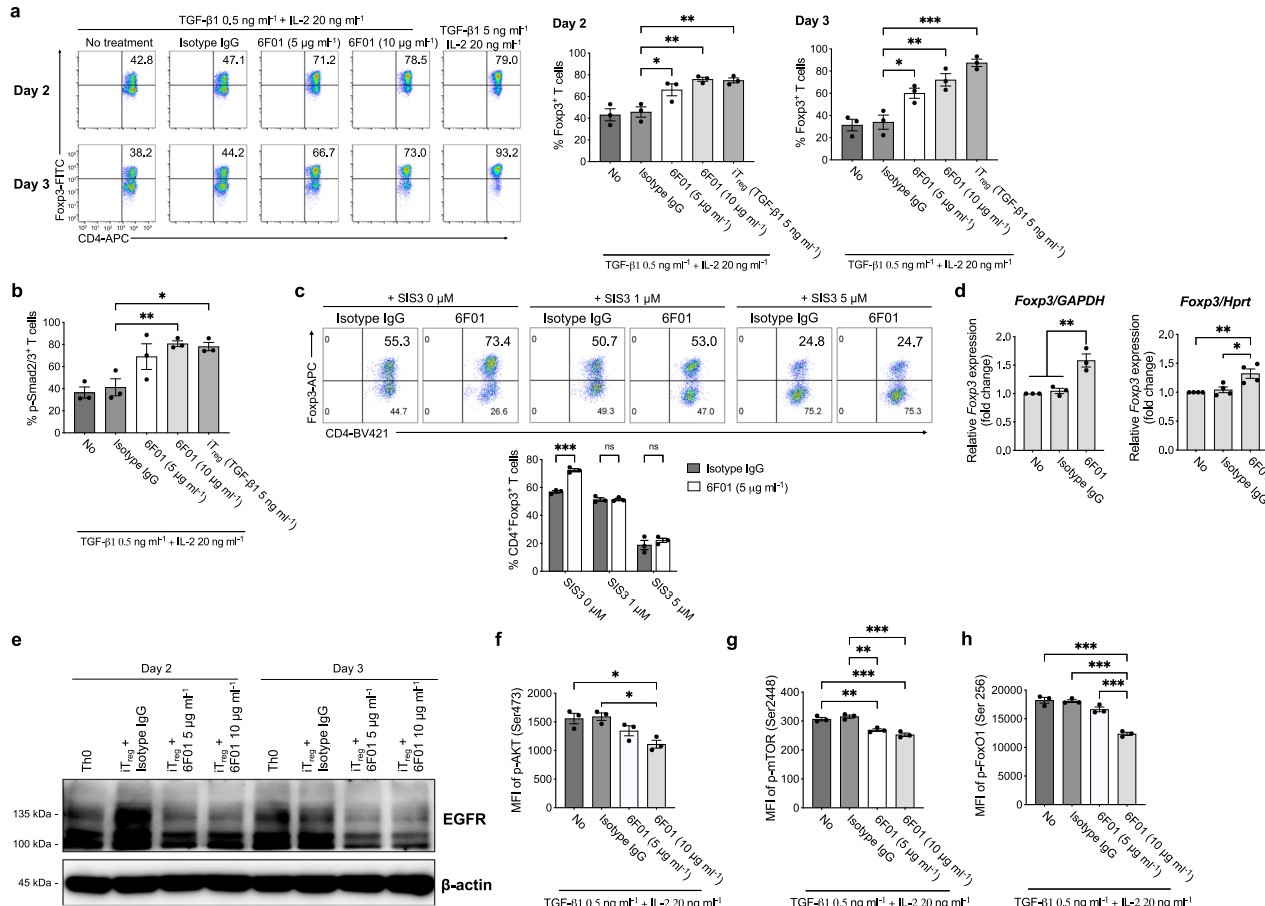

**Fig. 5 | Lrig1 stimulation enhances Foxp3 expression via induction of Smad2/3 phosphorylation. a**, **b** Representative images (left) or the percentage (right) of CD4⁺Foxp3⁺ (**a**) (*n* = 3) or p-Smad2/3⁺ (**b**) (*n* = 3) T cells by 6F01 stimulation in a dose-dependent manner during iT$_{reg}$ differentiation. Treatment of 5 ng ml⁻¹ TGF-β1 was used as a positive control. Data are expressed as mean ± S.E.M (in **a** Day 2 *P = 0.0226, **P = 0.0019 Isotype IgG vs 6F01, **P = 0.0024 Isotype IgG vs iT$_{reg}$; Day3 *P = 0.0156, **P = 0.0012, ***P < 0.001) (in **b** *P = 0.0123, **P = 0.0081).
**c** Representative dot plots (upper) or quantification of the level of CD4⁺Foxp3⁺ T cells in SIS3 treatment with 6F01 or isotype IgG during iT$_{reg}$ differentiation in a dose-dependent manner (*n* = 3). Data are expressed as mean ± S.E.M (***P < 0.001).
**d** The level of *Foxp3* mRNA from mouse iT$_{reg}$ cells stimulated with 6F01 or isotype control normalized with *GAPDH* (*left*) (*n* = 3) or *Hprt* (*right*) (*n* = 4) level. Data are expressed as mean ± S.E.M (in left panel **P = 0.0029 No vs 6F01, **P = 0.0044

Isotype IgG vs 6F01; in right panel *P = 0.0141, **P = 0.0062). **e** The level of EGFR expression of Th0 or iT$_{reg}$ cells in response to isotype IgG or various 6F01 treatment using western blot analysis. β-actin was used as a quality control. The experiment was repeated three times independently with similar results. **f**–**h** Quantification of MFI of phosphorylated AKT (Ser473), mTOR (Ser2448), or FoxO1 (Ser256) by stimulation of 6F01 or isotype control (*n* = 3 for each group). Data are expressed as mean ± S.E.M (p-AKT: *P = 0.0164 No vs 6F01, *P = 0.0112 Isotype IgG vs 6F01; p-mTOR: **P = 0.005 No vs 6F01, **P = 0.001 Isotype IgG vs 6F01, ***P < 0.001; p-FoxO1: ***P < 0.001). One-way ANOVA with Dunnett's multiple comparisons test (**a**, **b**) or one-way ANOVA with Tukey's multiple comparisons test (**d**, **f**–**h**) or two-way ANOVA with Šídák's multiple comparisons test (**c**). Source data are provided as a Source Data file.

To confirm the involvement of EGFR signaling in the Lrig1 pathway in T$_{reg}$ cells, sub-optimally induced T$_{reg}$ cells were treated with different concentrations of EGFR-specific inhibitors (Gefitinib and Erlotinib). The p-AKT level was reduced by 6F01 treatment (Fig. 5f) or EGFR inhibitors. However, the EGFR inhibitors did not increase the population of p-Smad2/3⁺ or Foxp3⁺ T cells, which was consistently increased by the 6F01 treatment (Supplementary Fig. 18). These results suggest that EGFR and p-AKT may not be a primary signal mediator in the Lrig1-mediated intracellular signaling pathway in T$_{reg}$ cells. These findings demonstrate that 6F01-mediated Lrig1 stimulation can induce EGFR degradation and reduce p-AKT/mTOR signaling. However, reduction of p-AKT may be likely irrelevant to the increase in phosphorylation of Smad2/3 and the expression of Foxp3.

To investigate the therapeutic potential of 6F01 mAb in the animal model of autoimmunity, the experimental autoimmune encephalomyelitis (EAE) mice were treated with 6F01 mAb three times a week, and the treatment with anti-IL-17A mAb was used as a positive control. The improvement of the clinical score by 6F01 mAb treatment was comparable to that by anti-IL-17A mAb treatment (Fig. 6a). And the

level of inflammatory T cell infiltration into the spinal cord or demyelination in the spinal cord was substantially reduced by 6F01 mAb treatment (Fig. 6b). When the level of T cell subsets in the lymph node (LN) and spinal cord (SC) of 6F01-treated EAE mice was analyzed, the increases of Lrig1⁺Foxp3⁺ T cells and total CD4⁺Foxp3⁺ T cells were observed in both lymph node and spinal cord (Fig. 6c, d). In addition, we detected the level of CD4⁺IL-17A⁺ or CD4⁺IFNγ⁺ T cells was significantly decreased in lymph nodes and spinal cord from EAE mice treated with 6F01 compared to isotype control-treated mice (Fig. 6e, f). Therefore, targeting Lrig1 with mAb enhances the anti-inflammatory functions of the suppressive T cell population to protect autoimmunity in the EAE animal models.

## Discussion

Onset, progression, and maintenance of autoimmune disease are known to be contributed by many immunological components such as autoreactive T and B cells, autoantibody, infiltrating inflammatory cells, and the secreted inflammatory cytokines and chemokines[49–51]. Current therapeutic reagents for treating autoimmune patients

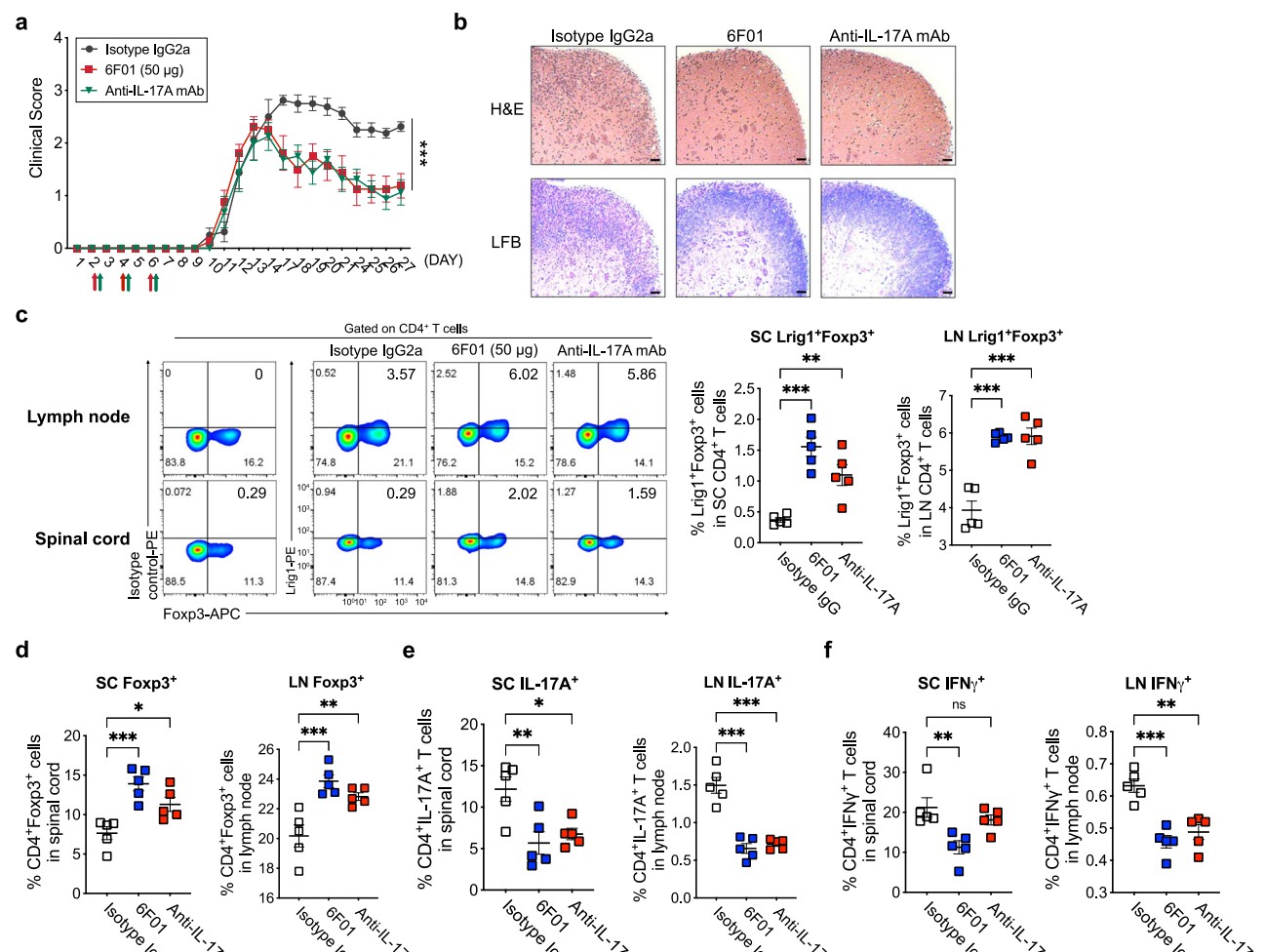

**Fig. 6 | Therapeutic efficacy of Lrig1-specific monoclonal antibody (6F01) in the experimental autoimmune encephalomyelitis (EAE) model. a** Clinical score of EAE symptoms until 27 days after EAE onset. EAE model mice were treated with isotype IgG2a ($n = 8$), 50 μg of 6F01 ($n = 8$), or anti-IL-17A mAb ($n = 8$). Each arrow indicates the injection day of isotype IgG2a, 50 μg of 6F01 or anti-IL-17A monoclonal antibody (mAb). Data are expressed as mean ± S.E.M (***$P < 0.0001$). **b** Representative images showing the level of inflammatory cell infiltration into the spinal cord and the level of spinal cord demyelination by H&E or LFB staining, respectively. Scale bar indicates 50 μm. The experiment was repeated two times independently with similar results. **c**–**f** The level of Lrig1⁺Foxp3⁺ (**c**), total Foxp3⁺ (**d**), IL-17A⁺ (**e**), or IFNγ⁺ cells (**f**) among CD4⁺ T cells in the spinal cord (SC) or lymph node (LN) of the isotype IgG2a-, 6F01-, or anti-IL-17A mAb-treated EAE animals ($n = 5$ for each group). Data are expressed as mean ± S.E.M (in **c**, SC **$P = 0.0064$, ***$P = 0.0001$; LN ***$P < 0.001$) (in **d**, SC *$P = 0.0288$, ***$P = 0.0007$; LN **$P = 0.0089$, ***$P = 0.0008$) (in **e**, SC *$P = 0.022$, **$P = 0.0071$; LN ***$P < 0.001$) (in **f**, SC **$P = 0.0067$; LN **$P = 0.0011$, ***$P = 0.0002$). ns not significant. Two-way ANOVA with Dunnett's multiple comparisons test (**a**), One-way ANOVA with Tukey's multiple comparisons test (**c**–**f**). Source data are provided as a Source Data file.

are mAbs targeting inflammatory cytokines, their receptors or costimulatory molecules, and the small molecules inhibiting the signal mediators involved in the inflammatory cytokine/chemokine receptor-mediated intracellular signaling pathway[52]. However, due to the functional and physiological redundancy among inflammatory cytokines and chemokines, the therapeutic efficacy of current medication for autoimmunity has been challenged. Instead of blocking the function of each inflammatory component in the autoimmune microenvironment, a new immunological entity needs to be developed to modulate the autoimmune microenvironment in a coordinated and systemic manner. In this regard, T$_{reg}$ cells are the best candidate, which play a crucial role in maintaining immune homeostasis and preventing the pathogenesis of autoimmune diseases[3]. However, the previously identified surface proteins known to be specific to T$_{reg}$ cells are also expressed in activated T cells and pro-inflammatory T cell subsets, which is the major limitation for the therapeutic application of T$_{reg}$ cells.

Using next-generation sequencing, microarray analysis, and functional gene network analysis, we identified Lrig1 as a new surface protein specific to T$_{reg}$ cells and the immune-suppressive T cell sub-population from Th17 cells. Previously, Lrig1 has been known to be detected on the surface of several tumor cells or stem cells, functioning as a negative regulator of ErbB receptors or tumor suppressors[53]. However, the immunological functions of Lrig1 are mainly unknown. The level of Lrig1 protein was highly enriched in mouse and human T$_{reg}$ cells, especially in the activated stage. In contrast, its expression in activated T cells, other T cell subsets, or non-T immune cells was substantially low. TGF-β1 induced the expression of Lrig1, and a higher level of PD-1, CD25, IL-10, or Foxp3 was detected in the CD4⁺Lrig1⁺ population than CD4⁺Lrig1⁻ population in vitro and in vivo (Fig. 1). Several studies have reported that activated T$_{reg}$ cells have increased suppressive activity and largely infiltrate into tumor tissues[5,54]. Currently, CD121a/b, CD137, and GARP are the surface markers for the activated T$_{reg}$ cells[55–57]. But these surface proteins are also detected in activated CD4⁺ or CD8⁺ T cells, dendritic cells, or platelets at substantially high levels[58–60]. In this study, Lrig1 can be a new surface protein required for the immune-suppressive functions in the activated stage.

The immuno-suppressive activity of $T_{reg}$ cells was positively correlated with the level of Lrig1 on the surface, and CD4$^+$Foxp3$^+$ T cells expressing the low level of Lrig1 on the surface showed substantially low suppressive potential. During mouse and human CD4$^+$ naïve T cell differentiation into various T cell subsets, the recognizable level of Lrig1 was detected on the surface. Interestingly, the suppressive activity of Lrig1$^+$ Th17 cells was comparable to that of Lrig1$^+$ $T_{reg}$ cells and substantially higher than that of Lrig1$^-$ Th17 cells. The consistent results were obtained when $T_{reg}$ cells or Th17 cells were isolated from the mouse using CD25 or CCR6 as a surface marker for $T_{reg}$ cells and Th17 cells, respectively. However, the Lrig1$^+$ population from the activated T cells (Th0) and Th1 cells neither exhibited the suppressive activity nor the functional difference between Lrig1$^+$ and Lrig1$^-$ populations (Fig. 2). Recently, it was demonstrated that Th17 cells could be functionally converted into the suppressive population in particular immunological conditions in vivo[37,38]. Our results suggest the important finding that Lrig1 may be a new surface protein to represent the immuno-suppressive population from Th17 and $T_{reg}$ cells and can serve as a unique or combinational marker with CD25 for isolating the therapeutically effective immuno-suppressive population from T cells.

The functional importance of Lrig1 in the immuno-suppressive activity of $T_{reg}$ cells was confirmed using siRNA-mediated knockdown of *Lrig1* and *Lrig1* KO mouse. Consistent with the results from Fig. 2, the suppressive potential of $T_{reg}$ cells was significantly decreased by the reduced expression or absence of Lrig1. Surprisingly, the differentiation capacity of CD4$^+$ naïve T cells into various T cell subsets or the number of Th1, Th2, Th17, or $T_{reg}$ cells was not affected in *Lrig1*-deficient mice. Moreover, the level of Foxp3, CD25, GITR, or PD-1 expression on $T_{reg}$ cells and IL-2-mediated $T_{reg}$ cell proliferation were not altered in *Lrig1*-deficient conditions (Fig. 3). These results are physiologically compatible because no immunological abnormality was observed in the *Lrig1* KO mouse phenotype although most abnormalities shown in *Lrig1*$^{-/-}$ mice may be related to the uncontrolled proliferation of stem cells in intestinal or epithelial areas or cancer cells[42,61–64]. Previous studies have reported that skin-resident $T_{reg}$ cells play an essential role in hair follicle regeneration by inducing the proliferation and differentiation of hair follicle stem cells[43]. Also, *Lrig1* KO mice show psoriasis-like epidermal hyperplasia with the over-proliferative ability of keratinocytes, and further loss of *Lrig1* causes chronic inflammation through the STAT3-dependent pathway[42,61]. These data strongly support our finding that Lrig1 is necessary for the functions of the immuno-suppressive population of T cells.

The therapeutic efficacy of CD4$^+$Lrig1$^+$ T cells was confirmed in chronic IBD and lupus genetic-animal model through the adoptive transfer, which is comparable to that of CD4$^+$Foxp3$^+$ T cells in the IBD model and CD4$^+$CD25$^+$ T cells or methylprednisolone in the lupus-prone model. These results suggest that a combination therapy of CD4$^+$Lrig1$^+$ T cells and methylprednisolone may provide a more effective treatment for lupus patients with less toxicity. Consistent with the high therapeutic potential of CD4$^+$Lrig1$^+$ T cells in IBD and lupus animal models, the higher expression of the genes involved in functions of $T_{reg}$ cells or induced by Foxp3 was observed in CD4$^+$Lrig1$^+$ T cells than CD4$^+$Lrig1$^-$ T cells (Fig. 4).

Agonistic $T_{reg}$-ness activation of Lrig1 by 6F01 mAb induces the phosphorylation of Smad2/3, followed by the enhanced *Foxp3* transcription. Lrig1-mediated inductions of Smad2/3 phosphorylation and Foxp3 expression by 6F01 treatment were also observed in Th17 cell-skewing differentiation (Fig. 5). Additionally, we used the EGFR inhibitors to confirm the involvement of EGFR signaling in the Lrig1 pathway in $T_{reg}$ cells. As shown in Supplementary Fig. 18, when sub-optimally induced $T_{reg}$ cells were treated with different concentrations of EGFR-specific inhibitors (Gefitinib and Erlotinib) for 1 day, the p-AKT level was reduced by the EGFR inhibitors as similar with 6F01 treatment (included in Fig. 5f). However, the treatment of EGFR inhibitors did not increase the population of p-Smad2/3$^+$ or Foxp3$^+$ T cells, which

was consistently increased by the 6F01 treatment. These results suggest that EGFR may not be a primary signal mediator in the Lrig1-mediated intracellular signaling pathway in $T_{reg}$ cells. The possibility can be explained by the fact that the level of EGFR is very low on the surface of normal $T_{reg}$ cells. Alternatively, intracellular signaling context and transcription factor functional network constellation might be quite different between cancer cells and $T_{reg}$ cells. It may be possible that other or unidentified members of the receptor tyrosine kinase (RTK) family, such as c-MET, AXL, MERTK, or VEGFR, or an unknown surface protein associated with tyrosine kinase, can be involved in the Lrig1-mediated intracellular signaling pathway in $T_{reg}$ cells[65–67]. In the EAE animal model, 6F01 mAb showed the equivalent level of therapeutic potential to anti-IL-17A mAb in a dose-dependent manner. In line with our mechanism of action study, 6F01 mAb treatment significantly reduced the level of demyelination or infiltration of inflammatory cells via the increase of Foxp3$^+$ T cells and decrease of Th17 and Th1 cells in the spinal cord (Fig. 6).

Previous studies suggest that Lrig1 shows the tumor suppressor activity via functioning as a negative regulator of the ErbB RTK family in the tumor microenvironment[53]. In the immunological environment, CD4$^+$ T cells express a low level of EGFR, one of the ErbB family, and the expression level of EGFR is higher on $T_{reg}$ cells than that on effector CD4$^+$ or CD8$^+$ T cells in tumor or inflammatory microenvironment[68,69]. AKT, a downstream signal mediator of EGFR or RTKs, is also known to play an essential role in T cell development or activation. The results from our study demonstrate that Lrig1 stimulation in $T_{reg}$ cells induced the degradation of EGFR and reduction of p-AKT. However, EGFR-specific inhibitors reduced the p-AKT level, but the population of p-Smad2/3$^+$ or Foxp3$^+$ T cells was not increased. Therefore, the reduction of p-AKT by agonistic stimulation of Lrig1 in $T_{reg}$ cells may be likely irrelevant to the Lrig1-mediated functional enhancement of $T_{reg}$ cells. Accordingly, the intracellular signaling pathway for the increase of p-Smad2/3 and Foxp3 by Lrig1 stimulation in $T_{reg}$ cells needs to be further elucidated.

Although this study revealed that the expression and function of Lrig1 are regulated by TGF-β1, we propose that there is a possibility that bone morphogenetic protein (BMP) signaling, one of the TGF-β superfamily members, may also be involved in Lrig1 signaling in $T_{reg}$ cells. The previous report has shown that LRIG proteins, including LRIG1, regulate lipid metabolism via BMP signaling through the phosphorylation of Smad1/5 in mouse embryonic fibroblasts, suggesting LRIG proteins may play an essential role as BMP sensitizers[70]. Furthermore, several studies have found that BMP signaling through BMPR1α controls $T_{reg}$ lineage and stability via Smad1/5/8 phosphorylation[71,72], and BMPR1α is highly expressed in activated T cells and $T_{reg}$ cells compared to naïve CD4$^+$ T cells[72,73]. In addition, TGF-β and BMP signals have a synergistic effect on the induction of i$T_{reg}$ cells even though the TGF-β signaling pathway is a key factor for the development of Foxp3$^+$ $T_{reg}$ cells[74]. Thus, we hypothesize that Lrig1 may play a role in the expression of suppressive markers through both TGF-β and BMP signaling to induce the phosphorylation of Smad1/5/8 from BMP and Smad2/3 from TGF-β. The molecular basis for the association of Lrig1 and BMP signaling remains to be elucidated.

The major therapeutic barrier of the mAb specific to the surface protein is the cross-reactivity to the protein on the non-target cells. There were no physiological abnormalities in *Lrig1* knockout mice except minor bone mineral density and coat texture abnormality. Any abnormal immunological features such as splenomegaly or inflammatory CD4$^+$ T cell population in lymph nodes and lungs were undetected. It is noteworthy that the fact that *Lrig1*$^{-/-}$ mice did not show any significant physiological abnormality may hold the promise of the minimal cytotoxicity of mAb specific to Lrig1 upon therapeutic application. Also, we repeatedly confirmed Lrig1 mAb 6F01 doesn't bind to *Lrig1*-null i$T_{reg}$ cells with a similar level of isotype control. In addition, other Lrig family proteins, such as Lrig2 and Lrig3, are rarely expressed

in CD4$^+$ T cells, including T$_{reg}$ cells, suggesting 6F01 specifically binds to the Lrig1 protein and functions as T$_{reg}$-ness stimulatory mAb. From our preliminary results, 6F01 mAb did not show any abnormality in the single-dose and repeat-dose toxicity test, which is consistent with the results from the *Lrig1* KO mouse. Currently, the humanized form of 6F01 mAb was generated, and its large-scale production in the GMP facility is being scheduled for the GLP toxicity and clinical study.

Although we have tested the isotype specificity and non-toxicity of the anti-Lrig1 monoclonal antibody, there would be possible side effects of the LRIG1 antibody treatment in humans based on the current knowledge of LRIG1 expression and function in other cell types. (1) Previously, Lrig1 has been well known as a tumor suppressor[75–77]. When the T$_{reg}$ cell-stimulating mAb specific to Lrig1 is administered to patients with autoimmune disease, there is a possibility that the antibody binds to LRIG1$^+$ cells, such as epithelial or intestinal stem cells in other tissues and may affect the proliferation status of the cells. Also, this Lrig1 mAb may not be beneficial for anti-cancer therapy because the function of T$_{reg}$ cell is an immunological barrier in the tumor microenvironment. (2) Previous report has shown that *LRIG1* gene variants have a risk and association with type 2 diabetes[78]. Because the functional importance of Lrig1 in various metabolic diseases such as type 2 diabetes has been explored very recently, it remains to be seen what therapeutic efficacy of Lrig1 mAb stimulating T$_{reg}$ cell functions would show to the patient with type 2 diabetes. (3) In many previous studies, it has been known that Lrig1 regulates various stem cell quiescence such as epidermal interfollicular stem cells, intestinal stem cells, or neural stem cells[79]. Although this mechanism has not yet been fully elucidated, one possibility is that T$_{reg}$-ness stimulating mAb specific to Lrig1 may enhance the sensitization of BMP signal in stem cells for regulating their quiescence status.

Our study is the first report about the immunological functions of Lrig1 and its therapeutic potential for autoimmune disease using mAb therapy and autologous adoptive T$_{reg}$ cell therapy. The results of our study demonstrate the important immunological findings of Lrig1 in that (1) Lrig1 is a new surface protein important for the immuno-suppressive function of T$_{reg}$ cells as well as Th17 cells, (2) agonistic T$_{reg}$-ness stimulation of Lrig1 by 6F01 mAb or adoptive transfer of Lrig1$^+$ CD4$^+$ T or T$_{reg}$ cells significantly alleviates the autoimmune symptoms in various autoimmune animal models, (3) intracellular signaling pathway through stimulation of Lrig1 is the induction of Smad2/3 phosphorylation and thereby the increase of *Foxp3* transcription. Therefore, the reagents targeting Lrig1 protein on the surface of the immuno-suppressive T cell population may provide a new therapeutic regime for treating autoimmunity.

## Methods

### Animals
All mice were maintained under specific pathogen-free (SPF) conditions and all procedures were performed in accordance with protocols approved by the Institutional Animal Care and Use Committee (IACUC) in Yonsei University (IACUC-A-202102-1210-03). All mice had a 12 h/12 h day/night cycle and were provided with Rodent diet (JA Bio, South Korea) and a GLP mini mouse house for enrichment. All mice were maintained at $22 \pm 2\,°C$ and $50 \pm 5\%$ humidity. C57BL/6 J (000664, Jackson Laboratory) and Balb/c (000651, Jackson Laboratory) female mice at 7-8 weeks of age were purchased from Orient Bio (Republic of Korea). Foxp3-IRES-RFP mice (C57BL/6-Foxp3$^{tm/Flv/J}$), Foxp3-IRES-GFP (B6.Cg-FoxP3$^{tm2Tch/J}$) mice, and *Rag1$^{−/−}$* (B6.129S7-Rag1$^{tm1Mom}$/J) mice were provided by Dr. Rho Hyun Seong at Seoul National University. Also, *Lrig1* heterozygous and knockout (*Lrig1*$^{tm1a(EUCOMM)Wtsi/N}$) mice were generated (strain origin: C57BL/6 N) and characterized as previously described[42]. When used, the *Lrig1*-null mice did not have any abnormalities and were age-sex matched with *Lrig1$^{+/−}$* mice. (NZB/ NZW) F1 (100008, Jackson Laboratory) female mice at the age of 23 weeks were purchased from Central Lab Animal, Inc. (Republic of Korea). Age and sex-matched mice were used for each study and both male and female mice were used at an age of 7–10 weeks. Experimental/control animals were bred separately. Sex was not selected for each experiment. Mice were euthanized by $CO_2$ asphyxiation under controlled conditions.

### Cell culture
Jurkat T cells (Clone E6-1, ATCC #TIB-152) were maintained in Roswell Park Memorial Institute 1640 (RPMI 1640, Lonza) with 10% heat-inactivated Fetal Bovine Serum (FBS, GIBCO), 2 mM L-glutamine (Lonza) and 100 units mL$^{-1}$ of streptom and 100 µg mL$^{-1}$ of streptomycin (Lonza). Mouse primary lymphocytes were maintained in RPMI 1640 with 7.5% heat-inactivated FBS, 2 mM L-glutamine, 100 µg ml$^{-1}$ streptom/streptomycin, and 50 µM β-mercaptoethanol (Sigma-Aldrich). Human primary lymphocytes were maintained in X-VIVO TM 15 with 5% human serum (Sigma-Aldrich) and 100 µg ml$^{-1}$ streptom/ streptomycin. All cells were cultured in 5% $CO_2$ at 37 °C.

### In vitro T cell activation and differentiation of naïve T cells into each T cell subset
Naïve CD4$^+$ T cells from lymph nodes or spleen were purified by CD4$^+$CD62L$^+$ T cell isolation MACS kit (Miltenyi Biotech) over 95% purity. Naïve T cells were activated for 72 h with plate-bound 1 µg ml$^{-1}$ of anti-CD3e and -CD28 antibody (BD Biosciences) in a 96-well flat-bottom plate. For the differentiation to each subset of T helper (Th) cells, the following condition was applied for each Th subset: Th1 cells for IL-12 (10 ng ml$^{-1}$; Peprotech), anti-IL-4 antibody (1 µg ml$^{-1}$; BioLegend), Th2 cells for IL-4 (20 ng ml$^{-1}$; Peprotech), anti-IFNγ antibody (1 µg ml$^{-1}$; BioLegend), Th17 cells for TGF-β1 (1 ng ml$^{-1}$; Peprotech), IL-6 (30 ng ml$^{-1}$; Peprotech), anti-IFNγ and IL-4 antibody (1 µg ml$^{-1}$) and iT$_{reg}$ cells for TGF-β1 (0.5, 1 or 5 ng ml$^{-1}$), and IL-2 (20 ng ml$^{-1}$; Peprotech).

### RNA extraction and qRT-PCR
Mouse naïve CD4$^+$ T cells were differentiated into each T cell subsets. Cells were harvested and washed using PBS and then resuspended by Trizol. RNA concentration was quantified by nano-drop. The extracted RNA was considered pure if the absorbance ratio at 260 and 280 nm was between 1.7 and 2.2, and the RIN value was above 9.9. cDNA was synthesized by the SMARTer Pico PCR cDNA synthesis kit (Takara Bio). Quantitative Real-time PCR (qRT-PCR) was performed with Real-time Bioanalyzer (CFX Connect Real-Time PCR Detection System) (Bio-Rad) using SYBR Green (Molecular Probes). The relative *Lrig1* gene expression was calculated by the ΔCT method. The mRNA expression was normalized to *Hprt*. 40 cycles of PCR were performed at 95 °C for 10 seconds, 61 °C for 15 seconds, and 72 °C for 30 seconds. For iT$_{reg}$ cells stimulated with 6F01, 3 days-differentiated iT$_{reg}$ cells were stimulated for 24 h in anti-CD3e antibody (1 µg ml$^{-1}$) and 6F01 (5 µg ml$^{-1}$) coated wells along with anti-CD28 antibody (1 µg ml$^{-1}$). The cells were harvested and washed with PBS and then resuspended by Trizol. The extracted RNA was synthesized to cDNA, and qRT-PCR was performed using SYBR Green by a Real-time Bioanalyzer. 30 cycles of PCR were performed at 95 °C for 10 s, 55 °C for 15 s, and 72 °C for 30 s. The relative *Foxp3* gene expression was calculated using the ΔCT method with the normalization of *GAPDH* and *Hprt* mRNA expression. The primer sequences were used as follows: Lrig1 forward, 5′- CCT CTA TCC AAG CAA CCA TGA CAG-3′ and reverse, 5′- GGG CTT CAG TAG ATA TGG CGT CTT-3′, Hprt forward, 5′- CTG GTG AAA AGG ACC TCT CGA AG-3′ and reverse, 5′- CCA GTT TCA CTA ATG ACA CAA ACG-3′, Foxp3 forward, 5′-CCT GGT TGT GAG AAG GTC TTC G-3′ and reverse, 5′-TGC TCC AGA GAC TGC ACC ACT T-3′, GAPDH forward, 5′-AAA TGG TGA AGG TCG GTG TG-3′ and reverse, 5′-TGA AGG GGT CGT TGA TGG-3′.

### Flow cytometric analysis
The fluorescence-conjugated antibodies to stain surface or intracellular antigens were followed (all antibodies were diluted 1:250 for

staining, otherwise the dilution factor was noted).: anti-CD3 (17A2; BioLegend) APC-cy7; anti-CD4 (RM 4-5; Thermo Fisher Scientific) FITC, APC; anti-CD4 (RM4-4; BioLegend) BV421; anti-CD8a (53-6.7; Thermo Fisher Scientific) FITC; anti-CD19 (eBio1D3(1D3), Thermo Fisher Scientific) FITC; anti-Foxp3 (FJK-16s; Thermo Fisher Scientific) FITC, PE, APC; anti-IFNγ (XMG1.2; Thermo Fisher Scientific) FITC, PE, APC; anti-IL-4 (11B11; Thermo Fisher Scientific) PE; anti-IL-17A (eBio17B7; Thermo Fisher Scientific) PE; anti-IL-10 (JES5-16E3; Thermo Fisher Scientific) PE, APC (1:200); anti-CD25 (PC61.5; Thermo Fisher Scientific) FITC, PE; anti-PD-1 (J43; Thermo Fisher Scientific) PE; anti-GITR (DTA-1; Thermo Fisher Scientific) PE; anti-TIGIT (GIGD7; Thermo Fisher Scientific) PE; anti-CTLA-4 (UC10-4B9; Thermo Fisher Scientific) PE; anti-CD62L (MEL-14; Thermo Fisher Scientific) FITC; anti-CD44 (IM7; Thermo Fisher Scientific) PE, PE-cy7; anti-CD45.1 (A20; BD Biosciences) APC-cy7; anti-CD45.2 (104; Thermo Fisher Scientific) FITC; anti-Lrig1 (polyclonal; R&D systems) AF488, PE (1:100); anti-LRIG1 (789211; R&D systems) AF488, PE (1:100); anti-Smad2 (pS465/pS467)/Smad3 (pS423/pS425) (072-670; BD Biosciences) PE (1:100); anti-AKT (C67E7; Cell Signaling Technology); anti-p-AKT1 (Ser473) (SDRNR; Thermo Fisher Scientific) APC; anti-mTOR (7C10; Cell Signaling Technology) (1:400); anti-p-mTOR (Ser2448) (MRRBY; Thermo Fisher Scientific) PE-cy7; anti-FoxO1 (C29H4; Cell Signaling Technology) (1:200); anti-p-FoxO1 (Ser256) (E1F7T; Cell Signaling Technology). For staining of surface molecules, cells were stained with PBS containing relevant antibodies for 30 min at 4 °C. For cell restimulation, an eBioscience cell restimulation cocktail with protein transport inhibitors (Thermo Fisher Scientific) was treated to cells for 6 hr at 37 °C. Intracellular staining was carried out using the Foxp3 transcription factor buffer set (Thermo Fisher Scientific). Stained cells were detected with FACSCalibur, FACSCanto II, LSRFortessa (Becton Dickinson) using BD FACSDiva (v9.0), or Spectral cell analyzer (SA3800; Sony Biotechnology) using SA3800 software (FCS Express Software 6.0) and analyzed by Flowjo software (v10.5.3). For sorting of natural $T_{reg}$ cells, CD4$^+$ T cells were purified from the spleen of Foxp3-IRES-RFP or Foxp3-IRES-GFP mice by MACS and sorted by FACSAria II cell sorter using RFP or GFP marker. All antibody information is shown in Supplementary Table 1.

## Isolation and differentiation of human T cells

Blood from human participants was obtained from the LRS chamber by Korean Redcross Blood Services. This study was approved by the Institutional Review Board at the Bioethics Committee of Korean Redcross Blood Services (21-1-Jung-1). Korean Redcross Blood Services did not select donors based on sex or age. They randomly provided the blood from donors and does not provide participant's information. Sex or gender was not considered in each study because we focused on the LRIG1 expression level in healthy individuals. Peripheral blood mononuclear cells (PBMCs) were isolated from the whole blood of healthy donors by density gradient centrifugation using Ficoll Histopaque (Sigma-Aldrich). Isolated PBMCs were stained by APC-conjugated anti-CD4 antibody and PE-conjugated anti-CD25 antibody (Thermo Fisher Scientific) for 30 min. Naive CD4$^+$CD25$^-$ T cells were sorted by Sony cell sorter (SH800S; Sony Biotechnology). Purified naïve CD4$^+$ T cells were activated by Dynabead human T activator CD3/28 (Thermo Fisher Scientific) and differentiated into each T cell subset using Th cell differentiation conditions; Th1 cells for IL-12 (2.5 ng ml$^{-1}$, Peprotech), anti-IL-4 antibody (2 μg ml$^{-1}$, BioLegend), Th2 cells for IL-4 (12.5 ng ml$^{-1}$, Peprotech), anti-IFNγ antibody (2 μg ml$^{-1}$, BioLegend), Th17 cells for IL-1β (20 ng ml$^{-1}$, Peprotech), IL-6 (30 ng ml$^{-1}$, Peprotech), IL-23 (30 ng ml$^{-1}$, Peprotech), TGF-β1 (2.25 ng ml$^{-1}$, R&D), iT$_{reg}$ cells for TGF-β1 (5 ng ml$^{-1}$, R&D systems), and IL-2 (100 U ml$^{-1}$, Peprotech). After 72 h of differentiation, the cells were washed with PBS and stained with anti-CD4 antibody and anti-LRIG1 antibody (R&D Systems). Stained cells were analyzed with Sony spectral analyzer and Flowjo software.

## Western blot analysis

The expression of LRIG1 protein was measured in human-activated CD4$^+$ T cells, iT$_{reg}$ cells with or without Rapamycin, or Everolimus. Naïve CD4$^+$ T cells isolated from PBMCs were activated by Dynabeads human T activator CD3/28 with or without TGF-β1 and IL-2 to differentiate into iT$_{reg}$ cells. Under iT$_{reg}$ cell differentiation condition, Rapamycin (100 nM, Calbiochem EMD Millipore) or Everolimus (100 nM, Sigma-Aldrich) was treated additionally to induce FOXP3 expression and iT$_{reg}$ cell expansion. Jurkat T cells and anti-CD3e/28-activated Jurkat T cells were used as a control. After 72 h, the cells were harvested and lysed with RIPA buffer (Thermo Fisher Scientific) with protease inhibitor cocktail tablet (Roche) for 30 min on ice. After the centrifugation, the supernatant was quantified with a BCA protein assay (Thermo Fisher Scientific), and equal amounts of the supernatant were separated on 10% SDS polyacrylamide gels and transferred onto polyvinylidene difluoride (PVDF) transfer membrane (EMD Millipore). The membrane was blocked by 4% Bovine Serum Albumin (Sigma-Aldrich) and incubated with anti-LRIG1 antibody (R&D systems) followed by the manufacturer's protocol. After washing, anti-mouse IgG-conjugated horseradish peroxidase (Sigma-Aldrich) was bound in a dilution of 1:10,000. Anti-β-actin antibody (Cell Signaling) was used as a control. The signals were visualized by chemiluminescence using WEST-ZOL plus (iNtRON Biotechnology). LRIG1 expression was normalized to the expression of β-actin from the same sample. The expression of EGFR protein was measured in mouse-activated T cells, iT$_{reg}$ cells, and 6F01-stimulated iT$_{reg}$ cells. Mouse naïve T cells were differentiated for 2 or 3 days in wells coated with anti-CD3e antibody (1 μg ml$^{-1}$) and 6F01 (5 or 10 μg ml$^{-1}$) along with anti-CD28 antibody (1 μg ml$^{-1}$) under suboptimal iT$_{reg}$ skewing condition. The cells were harvested and lysed by RIPA buffer (Thermo Fisher Scientific). The lysate was quantified with BCA protein assay, and equal amounts of the proteins were separated on 7.5% SDS polyacrylamide gels and transferred onto a PVDF transfer membrane (EMD Millipore). The membrane was blocked by 4% Bovine Serum Albumin (Sigma-Aldrich) and incubated with an anti-EGFR antibody (Invitrogen) followed by the manufacturer's protocol. After washing, anti-mouse IgG-conjugated horseradish peroxidase (Abcam) was bound in a dilution of 1: 10,000. Anti-β-actin antibody (Cell Signaling Technology) was used as a control. The signals were visualized by chemiluminescence. Uncropped scans are provided as a Source Data file or at the end of Supplementary Information.

## In vitro suppression assay

For the isolation of T$_{reg}$ cells, CD4$^+$Foxp3$^+$ cells, CD4$^+$Lrig1$^+$ cells, CD4$^+$Lrig1$^-$ cells, CD4$^+$Lrig1$^+$Foxp3$^+$ cells, CD4$^+$Lrig1$^+$Foxp3$^-$ cells, and CD4$^+$Foxp3$^+$Lrig1$^-$ cells from Foxp3-IRES-GFP mice or CD4$^+$CD25$^+$ cells from *Lrig1* WT or KO mice were isolated by FACSAria II or SH800S cell sorter (Sony Biotechnology). For the isolation of splenic CD4$^+$ T cells, CD4$^+$CD25$^-$CCR6$^+$Lrig1$^-$ cells, CD4$^+$CD25$^-$CCR6$^+$Lrig1$^+$ cells, CD4$^+$CD25$^+$Lrig1$^-$ cells, and CD4$^+$CD25$^+$Lrig1$^+$ cells from C57BL/6 mice were isolated by SH800S cell sorter. In all, $5 \times 10^4$ naïve CD4$^+$ T cells (effector T cells) were stained with cell proliferation dye eFluor 670 (Thermo Fisher Scientific). Then, T$_{reg}$ cells and stained effector cells with different ratios were mixed in a 96 round-bottom well plate with Bruff media containing RPMI 1640, streptom/streptomycin, 7.5% heat-inactivated FBS, and β-mercaptoethanol. Co-cultured cells were stimulated by the soluble anti-CD3e antibody (BD Biosciences) and mitomycin-C treated-antigen-presenting cells (APCs) for the co-stimulatory signal at 37 °C and 5% CO$_2$. After 3 or 4 days of co-culture, the proliferation of effector cells was analyzed by Sony Spectral analyzer or LSRFortessa. For suppression assay using differentiated CD4$^+$ T cells, naïve CD4$^+$ T cells isolated from C57BL/6 mice were differentiated into each T cell subset in each Th0, Th1, Th17, or iT$_{reg}$-polarized culture condition and were sorted to Lrig1$^-$ or Lrig1$^+$ cells before mixing with effector cells. And the rest

of the experimental procedure was the same as above. For suppression assay using human CD4$^+$ T cells, naïve CD4$^+$ T cells were isolated from human PBMCs and were differentiated into each T cell subset such as Th0, Th17, and iT$_{reg}$. After 4 days of differentiation, the cells were sorted into LRIG1$^-$ or LRIG1$^+$ cells, and each cell was co-cultured with cell proliferation dye-stained naïve CD4$^+$ T cells. The cells were stimulated by Dynabeads human T activator CD3/28 in a 1:1 ratio with cells for 4 days and were analyzed through Flow cytometry. The suppressive activity was analyzed by Flowjo V10 or Sony SH800S software and calculated by the percentage of proliferating cells gating from division 1 to the final stage of division in cell proliferation dye eFlour 670-stained cells.

### siRNA nucleofection to iT$_{reg}$ cells

Silencer Select siRNAs targeting mouse *Lrig1* (AM16708, 4390771) and negative control (AM4611, AM4613) were purchased from Thermo Fisher Scientific. In all, $1 \times 10^6$ cells of iT$_{reg}$ cells per nucleofection reaction were resuspended in Mouse T cells Nucleofector Solution (100 µl per nucleofection) (Lonza) at room temperature. In all, 50, 150 nM of *Lrig1* siRNA (si*Lrig1*) or 150 nM of control siRNA (si*Control*) was combined with 100 µl of the cell suspension and transferred into the nucleocuvettes of the 4D-Nucleofector X unit (Lonza). The Nucleocuvettes were placed into the 4D-Nucleofector X unit and pulsed with the Nucleofector program DN-100. The nucleocuvettes were incubated for 10 min at 37 °C, 5% CO$_2$, and then resuspended with RPMI 1640 medium. After 24 h post nucleofection, siRNA-treated iT$_{reg}$ cells were stained with relevant antibodies to confirm the expression level of markers. The cells were co-cultured with eFlour 670-stained effector T cells for the suppression assay, followed by flow cytometric analysis. To measure the cell proliferation capacity, siRNA-treated iT$_{reg}$ cells were cultured with IL-2 (20 ng ml$^{-1}$) for 72 h. Cell proliferation of each group was quantified with the gate by Flowjo V10.

### ELISA

After the culture of activated total splenocytes, activated CD4$^+$ T cells, CD4$^+$Foxp3$^+$ T cells, CD4$^+$Lrig1$^-$ T cells, and CD4$^+$Lrig1$^+$ T cells for 72 h, the supernatants from each sample were collected. And the level of IL-10, an anti-inflammatory cytokine, was measured using a mouse IL-10 ELISA kit (Thermo Fisher Scientific) following the manufacturer's instruction.

### Isolation and analysis of lymphocytes in lung and skin of *Lrig1* KO mouse

Lung tissue was chopped into small pieces and digested with Collagenase II (300 U ml$^{-1}$, Worthington) and DNase 1 (10 mg ml$^{-1}$, Roche) for 1 h at 37 °C with shaking. A single-cell suspension was obtained by a 40 µm strainer. The cells were treated with RBC lysis buffer (BioLegend) and washed with RPMI media and the cells were incubated with anti-CD4, CD44, or CD62L antibodies. Flow cytometry analysis was performed by LSRFortessa. For hematoxylin and eosin (H&E) staining of lung and skin from mice, the lung and skin tissues were fixed in 10% formalin and were paraffin-embedded. The sections were stained with H&E by the conventional method. Images were visualized by a Hamamatsu slide scanner and analyzed with NanoZommer software (Hamamatsu).

### T cell transfer colitis model

To induce a colitis model, $5 \times 10^5$ FACS-sorted CD4$^+$CD25$^-$CD45RB$^{high}$ cells from CD45.1 mouse were injected into *Rag1*$^{-/-}$ mice intravenously with or without $2 \times 10^5$ CD4$^+$GFP$^+$ T$_{reg}$ cells from Foxp3-IRES-GFP mice or $2 \times 10^5$ CD4$^+$Lrig1$^{+/-}$ cells from C57BL/6 mice. Mice were monitored weekly for weight loss and clinical signs and then sacrificed. The colons were fixed with 10% formalin and 4% formaldehyde, and then Paraffin-embedded sections were stained with H&E. Also, the spleen was removed to deduce the severity of inflammation by comparing the size. The colon was cut into 1 cm$^{-2}$ size and incubated with RPMI 1640 containing EDTA. Cut colon tissue was dissociated with collagenase buffer. Lymphocytes in the colon were isolated by Percoll (Sigma-Aldrich)-gradient centrifugation. The isolated lymphocytes were re-stimulated in RPMI 1640 media with 10% heat-inactivated FBS, PMA 10 ng ml$^{-1}$, Ionomycin 500 ng ml$^{-1}$, and brefeldin A (Sigma-Aldrich). Restimulated lymphocytes were stained with the anti-Foxp3-APC antibody. Stained samples were analyzed by FACS Canto II (Becton Dickinson). Blood serum was collected at the sacrifice point, and the level of pro-inflammatory cytokines (IL-1β, TNFα) was measured by each ELISA kit (Thermo Fisher Scientific). For *Lrig1*$^{-/-}$ or *Lrig1*$^{+/-}$ iT$_{reg}$ cell transfer, $2 \times 10^5$ *Lrig1*$^{-/-}$ or *Lrig1*$^{+/-}$ iT$_{reg}$ cells were injected into CD4$^+$CD25$^-$CD45RB$^{high}$ transferred *Rag1*$^{-/-}$ mice. The weight of the mice was measured weekly. After sacrificed, the colons were fixed with 10% formalin and 4% formaldehyde, and then Paraffin-embedded sections were stained with H&E. Lymphocytes in the lymph node and colon were isolated and stained with anti-Foxp3 antibody-APC or anti-IFNγ antibody-PE and were analyzed by Spectral cell analyzer (SA3800; Sony Biotechnology).

### RNA sequencing analysis

CD4$^+$Lrig1$^{+/-}$ T cells were isolated from the spleen of 7 weeks of age female C57BL/6 mice. Total RNA was extracted using the RNeasy Plus Micro Kit (QIAGEN) and subjected to library construction and $2 \times 50$ paired-end sequencing on a NovaSeq 6000 (Illumina) at the Macrogen (Republic of Korea). The quality of sequencing reads was checked through FastQC (v0.11.7) and the reads were trimmed through Trimmomatic (v0.38), mapped through HISAT2 (v2.1.0), and were assembled through StringTie (v2.1.3b). Differentially expressed genes (DEGs) were determined by EdgeR (v3.14.0). Pathway analyses and upstream regulator analyses were performed using Ingenuity Pathway Analysis (Qiagen). For the analysis of DEGs, pathway and process enrichment analysis, including GO Biological Processes, KEGG Pathway, Reactome Gene Sets, CORUM, WikiPathways and PANTHER Pathway, or PaGenBase[80]/ TRRUST-based enrichment analysis was performed using Metascape[81] which is a user-friendly and web-based analysis tool. Heatmaps of the Top 20 DEGs or T$_{reg}$ cell marker genes were visualized by Heatmapper.

### Lupus nephritis model

Female (NZB/NZW) F1 mice at 21 weeks of age were purchased and maintained in a specific pathogen-free barrier facility. In all, 7 mice were assigned to each group. Adoptive transfer of 10$^6$ cells/mouse was proceeded by intravenous injection at 25 and 27 weeks. Untreated mice were intraperitoneally treated with Phosphate-buffered saline (PBS), and methylprednisolone (140 µg/mouse) was intraperitoneally injected at the same schedule. Proteinuria in urine collected from each mouse was detected with albumin reagent strips (Yongdong Pharmaceutical Co.). Proteinuria was quantified: 0, none or trace; 1+, ≤100 mg/dL; 2+, ≤300 mg/dL; 3+, ≤1000 mg/dL; and 4+, >1000 mg/dL.

### Development of Lrig1-targeting antibody 6F01

Anti-Lrig1 monoclonal antibodies were generated using the antibody phage display and the panning method using mouse *Lrig1* gene transfected-L cell as an antigen. After validating the antibody with high binding affinity among the generated anti-Lrig1 monoclonal antibodies through flow cytometric analysis, affinity maturation was performed through light-chain shuffling using the mouse Lrig1-ectodomain. Finally, 6F01 with high binding affinity to both mouse and human Lrig1 was selected through ELISA and FACS analysis. The cDNA clone of 6F01 was transfected into HEK293-F cells, and the 6F01 mAb was purified by Y-Biologics (South Korea). All unique materials used in this study are only available under a Material Transfer Agreement with the authors.

## Investigation of the function of 6F01 in iT$_{reg}$ cells using the phospho-flow assay

Mouse naive CD4$^+$ T cells ($2 \times 10^5$ cells/well) were differentiated under suboptimal iT$_{reg}$-skewing condition (TGF-β1 0.5 ng ml$^{-1}$ and IL-2 20 ng ml$^{-1}$) in each 96 flat-bottom well coated with 5 μg ml$^{-1}$ of mouse IgG isotype control (Invitrogen), 5 μg ml$^{-1}$ or 10 μg ml$^{-1}$ of 6F01 along with 1 μg ml$^{-1}$ of anti-CD3e antibody. In all, 1 μg ml$^{-1}$ of anti-CD28 antibody was added into the culture medium. After 2 or 3 days of differentiation, the cells were harvested and were fixed/permeabilized using FIX & PERM™ Cell Permeabilization Kit (Thermo Fisher Scientific). The fixed cells were stained with anti-CD4-BV421 or APC, anti-Foxp3-FITC or APC, anti-p-Smad2/3-PE, anti-AKT antibody (Cell Signaling Technology), anti-p-AKT-APC, anti-mTOR antibody (Cell Signaling Technology), anti-p-mTOR-PE-cy7, anti-FoxO1 antibody (Cell Signaling Technology), and anti-p-FoxO1 antibody (Cell Signaling Technology). Anti-Rabbit IgG-PE secondary antibody (Invitrogen) was used for unconjugated antibodies. The stained cells were measured by Flow cytometry. iT$_{reg}$ cells differentiated under optimal iT$_{reg}$-skewing condition (TGF-β1 5 ng ml$^{-1}$ and IL-2 20 ng ml$^{-1}$) were used as a positive control.

## Short-term kinetics of p-AKT and EGFR degradation by 6F01 treatment

To analyze the shorter kinetics of p-AKT and EGFR, 3 days differentiated iT$_{reg}$ cells were transferred into isotype IgG (5 μg ml$^{-1}$) or 6F01 (5 μg ml$^{-1}$) coated wells and were spun down for receiving signals from mAbs. The cells were cultured for the indicated times and were harvested. To examine the level of p-AKT, the cells were fixed/permeabilized using FIX & PERM™ Cell Permeabilization Kit (Thermo Fisher Scientific). The fixed cells were stained with anti-p-AKT-APC and were analyzed by Flow cytometry. To examine the level of EGFR, the cells were lysed with RIPA buffer (Thermo Fisher Scientific). Lysates were quantified with a BCA protein assay (Thermo Fisher Scientific), and equal amounts of the proteins were separated on 7.5% SDS polyacrylamide gels and transferred onto a PVDF transfer membrane (EMD Millipore). The membrane was blocked by 4% Bovine Serum Albumin (Sigma-Aldrich) and incubated with an anti-EGFR antibody (Invitrogen) followed by the manufacturer's protocol. After washing, anti-mouse IgG-conjugated horseradish peroxidase (Abcam) was bound in a dilution of 1:10,000. Anti-β-actin antibody (Cell Signaling Technology) was used as a control. The signals were visualized by chemiluminescence.

## Smad3 inhibitor assay

Mouse naïve CD4$^+$ T cells were differentiated under the suboptimal iT$_{reg}$-skewing condition in mouse isotype control IgG or 6F01 (5 μg ml$^{-1}$) coated wells. After 24 h differentiation, 1 or 5 μM of Smad3 inhibitor (SIS3) (Sigma-Aldrich) was added to the culture medium, and cells were cultured for an additional 24 h. The cells were harvested and stained with anti-CD4 antibody-BV421 and anti-Foxp3 antibody-APC and were analyzed by Flow cytometry.

## EGFR inhibitor assay

To investigate the function of EGFR inhibitor in iT$_{reg}$ cells, mouse CD4$^+$ naïve T cells were differentiated under sub-optimal iT$_{reg}$-skewing conditions. After 24 h of differentiation, each EGFR inhibitor Gefitinib (Cayman Chemical) or Erlotinib (Sigma-Aldrich) was treated to the cells dose-dependently. After an additional 24 h of incubation, the cells were harvested and fixed/permeabilized. The cells were stained with anti-p-AKT (Ser473)-APC, anti-p-Smad2/3-PE, and anti-Foxp3-APC antibodies, and the stained cells were analyzed through flow cytometry.

## EAE Induction and assessment

Female C57BL/6 mice were purchased, and all experiments in the EAE animal model were approved by the IACUC of Yonsei Laboratory Animal Research Center (IACUC-202006-1080-02). In all, 10-weeks-old female mice ($n = 6$ or 8/group) were immunized with the emulsion of 200 μg MOG$_{35-55}$ (MEVGWYRSPFSRVVHLYR-NGK) and Complete Freund's Adjuvant by subcutaneous injection, followed by intraperitoneal injection of 200 ng pertussis toxin (PTX) as following the manufacturer's instructions (Hooke Laboratories). After 24 h, the same amount of PTX was intraperitoneally injected again. Clinical score was daily evaluated on a scale 0 to 5 (0: normal, 1: weakness of tail, 2: limp tail and weakness of hind legs, 3: complete paralysis of hind legs, 4: partial front leg paralysis, 5: dead).

## Isolation and analysis of the lymph node or spinal cord-infiltrating lymphocytes

To isolate lymphocytes from the spinal cord, EAE-induced mice were perfused with PBS. Then, the spinal cords were excised and ground, followed by digested in 1 mg ml$^{-1}$ Collagenase D (Roche) and 50 μg ml$^{-1}$ DNase I (Roche) solution. The digested spinal cords were filtered with cell strainers, and lymphocytes were isolated by 40%/60%/90% Percoll (Cytiva) density gradient centrifugation. The isolated cells from the lymph node or spinal cord were stained with anti-CD4-BV421, anti-Lrig1-PE, anti-Foxp3-APC, anti-IL-17A-APC, and anti-IFNγ-PE antibodies and analyzed by Flow cytometry.

## Histologic studies of the EAE model

On day 27 or 30 of post-immunization, the spinal cords from EAE mice were fixed in 4% paraformaldehyde (PFA) for 24 h, paraffin-embedded, and sectioned. The slices were stained with either Hematoxylin & Eosin to analyze cell infiltration or Luxol Fast Blue (LFB) to evaluate demyelination. An optical microscope observed these stained samples.

## Statistics and reproducibility

No statistical method was used to predetermine sample size. All mice were randomly assigned to each experimental groups in each mouse model. All the clinical assessment experiments were conducted with a blind group allocation during data collection and analysis. As indicated in the figure legends, all quantitative data are presented with biologically independent experiments or samples. All data are presented as a mean ± the standard error of the mean (S.E.M). Data were analyzed by GraphPad Prism 9. Statistical analysis was determined with an unpaired Student's $t$ test, one-way ANOVA with Tukey's multiple comparison test or Dunnett's multiple comparison test, or two-way ANOVA with Šídák's multiple comparisons test, Tukey's multiple comparison test or Dunnett's multiple comparison test. *$P < 0.05$, **$P < 0.01$ and ***$P < 0.001$ were considered statistically significant.

## Reporting summary

Further information on research design is available in the Nature Portfolio Reporting Summary linked to this article.

## Data availability

The mRNA sequencing data generated in this study have been deposited in the Gene Expression Omnibus (GEO) database under accession code GSE236988. Source data are provided with this paper.

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

## Acknowledgements

We thank S. J. Ha for providing Foxp3-RFP- or Foxp3-GFP-KI mice. We also thank J. M. Koo and all other Professor S. K. Lee's laboratory members and Good T Cells, Inc. for suggestions, technical assistance, and insightful discussions. This research was supported by Global Research Laboratory (GRL) Program through the National Research Foundation of Korea (NRF), funded by the Ministry of Science and ICT (No. 2016K1A1A2912755-S.K.L.) and the National Research Foundation of Korea (NRF) grant funded by the Korean government (MSIT) (No. 2017R1A2A1A17069807-S.K.L.), and Advanced Medical Technology Development Program funded by Ministry of Health and Welfare (No. HI18C2010).

## Author contributions

S.K.L. conceived the project, and J.S.M., J.H.P., and S.K.L. designed experiments; J.S.M. and C.C.H. performed most of the experiments and analyzed data; B.Y.S., J.S.S., J.W.N., and E.S.L. performed Lrig1 mAb treatment using EAE model; J.S.M., K.P., S.H.L., and R.H.S. designed and carried out IBD model experiments; Y.K. and J.P. supported the IBD experiments; C.C.H., J.H.K., B.S.K., J.Y.S., Y.L., S.H.S., and E.C. performed Lrig1 mAb development and analysis; I.S. and F.M.W. kindly provided Lrig1 knockout and heterogenous mice and tissue histology of skin and lung of mice; J.S.M., C.H.M., and S.W.L. designed and performed adoptive transfer of T cells in the lupus-prone mouse model; J.S.S. provided the technical assistance with suppression assay; H.C and J.W.O performed qRT-PCR; T.M. performed western blot analysis; J.S.M., C.C.H., and S.K.L. wrote the manuscript; S.K.L. supervised the study.

## Competing interests

An invention disclosure describing Lrig1 and mAb has been filed based on the data generated in this study and is owned by Good T Cells, Inc. S.K.L. as a founder of Good T Cells, Inc. The remaining authors declare no competing interests.
