## [Peer Review File · Nature Communications]

Lrig1-expression confers suppressive function to CD4⁺ cells and is essential for averting autoimmunity via the Smad2/3/Foxp3 axisREVIEWER COMMENTS

Reviewer #1 (Remarks to the Author):

The manuscript entitled "LRIG1 essential for alleviating autoimmunity defines the suppressive population of CD4⁺T cells through signaling via the Smad2/3/Foxp3 axis" investigated the functional role of a new marker, LRIG1, in regulatory T cell (Treg) function. The authors showed that LRIG1 is mainly expressed on activated Treg cells and is involved in Treg cell function. The authors also raised a potential therapeutic role of LRIG1 in EAE. Given the pivotal role of Treg cells in maintaining immune homeostasis and in several diseases, this study can be an important finding and can contribute to a better understanding of the fundamental aspects of Treg cell biology. However, the following issues need to be addressed:

Major comments:

- The authors claim that targeting LRIG1 presents a therapeutic potential. However, the authors didn't treat the mice with anti-LIRG1 and anti-IL-17A in the same manner. The authors should use the same therapeutic strategy for 6F01 and anti-IL17A to analyze the therapeutic potential of modulating LRIG1.
- The authors propose that activation of LRIG1 induces activation of p-Smad2/3 and increases Foxp3 mRNA expression through an Akt/mTOR/Foxo mechanism. The authors didn't prove that and should analyze more carefully the signaling following LRIG1 activation and the outcome of this stimulation.
- The CD45RB^{High} colitis model was performed using total CD4⁺ sufficient or deficient for LRIG1. This model should be performed using LIRG1⁺ or LIRG1⁻ Tregs cells instead of total CD4⁺ cells. The conclusion of the current experiment is not clear and doesn't show that LIRG1⁺ Treg cells present a better suppressive capacity.
- Fig2A: The quantification of the in vitro suppressive assay is missing. A careful analysis of the CD4⁺ LIRG1⁺ cells is necessary to be able to interpret these results: percentage and MFI of Foxp3, CTLA4, CD25 etc.

- Fig2 C: the panel's results are not clear. Can you please quantitate those results?
- Fig3 C: WT Tregs should be added in this experiment, the level of suppression of the LRIG1+/- is really low
- Fig4: a more careful analysis of the RNAs-seq results should be added in the figure: volcano plot or heatmap of the Top 20 differentially expressed genes, KEGG/Go pathway analysis etc.
- Based on the impact of LRIG1 on the Foxp3 mRNA, the authors should analyze the development of the natural thymic Treg cells at an early and late stage.

Minor Comments:

- Statistical analysis is missing in different figures 1, 3,4,5,6, ext2,
- Different fonts on different figures.
- Fig 1e: the authors should add a representative isotype control
- Fig1f: the IL-10 should be represented IL-10 (y-axis) vs CD4(x-axis)
- Fig2B: not the same group between fig 2a and 2b
- Fig3: the level of LRIG1 should be analyzed and add to the figure
- Fig6 c is not convincing the authors should represent an isotype control and dot plot with better quality and the same gating
- Extended figure 1B: isotype control is needed
- Extended figure 1c: quantification is needed
- Extended figure 4: other markers are needed, CTLA4, PD1, Cd39, Cd73, etc
- Extended figure 6b, Survival curve should be added in the figure

Reviewer #2 (Remarks to the Author):

This is an interesting but preliminary study. The authors present what looks like an interesting discovery; that is, they show evidence suggesting that the transmembrane protein Lrig1 both marks and regulates regulatory T cells (Tregs). Furthermore, they generated an anti-Lrig-1 antibody, AF01, that promoted Treg function in vitro and in vivo and suppressed experimental autoimmune diseases. The strength of the paper is the novelty of the Treg-Lrig1 finding and the possible therapeutic potential of the developed Lrig1 antibody. However, on the downside, the paper contains overinterpretation of the data, it is mechanistically shallow, important controls are missing, and there are a series of questionmarks regarding the 'therapeutic' antibody.

Specific points regarding the biological discovery:

1. The authors analyze cytokine-stimulated T cells that they named Th0, Th1, Th2, and Th17 without confirming their respective characteristics. The authors should confirm their subtype identities through transcriptional profiling, cytokine production, or functionality or rename the cell populations according to their treatment rather than their believed phenotypes.
2. The authors claim in the Title and elsewhere that Lrig1 acts through Smad2/3 signaling, which is not firmly supported by the data presented. The only data pertaining to Smad2/3 that is presented is an experiment showing that treatment with the non-validated anti-Lrig1-antibody 6F01 for 2 or 3 days during iTreg differentiation results in enhanced pSmad2/3 levels (Fig. 5b). There could be numerous different explanations for this result. If the authors wish to investigate the role of Lrig1 and TGF-beta signaling in these T cell populations, they should perform proper signal transduction experiments, including shorter kinetics, specific agonists and inhibitors, and/or genetic (e.g., RNAi) experiments.
3. Similarly, in Supplementary Figure 8, the authors propose that LRIG1 functions through EGFR-mediated negative regulation of Smad2/3 signaling. This is too speculative given that no supportive data is provided. To this end, it is suggested that the authors address the role of EGF signaling in their experimental system.
4. The anti-LRIG1 antibody from R&D used in FACS is a polyclonal antibody raised against the entirety of the extracellular/luminal part of LRIG1. Therefore, it is likely that the

antibody crossreacts with other proteins. And the antibody do not seem to be well validated. Although the siRNA experiment in Fig. 3a looks good, it would be desirable to confirm the antibody's specificity via side-by-side experiments including cells both from wild-type and Lrig1 knockout mice.

Specific points regarding the 6F01 antibody:

5. The generation of 6F01 is not adequately described. The detailed methods should be included, such as information about the epitope used for its selection and the validation of its specificity.

6. Regarding the possible effects of 6F01 on Lrig1, it should be noted that the function of Lrig1 in the system under study is not known. Therefore, at this stage, it cannot be concluded whether 6F01 "activates" or "inactivates" Lrig1, both scenarios are possible.

7. 6F01 showed promising activities in vivo, however, important controls were missing. For example, it should be important to establish whether the antibody acts through Lrig1 or via other mechanisms. To this end, the authors could take advantage of their wild-type and Lrig1 knockout mice and investigate if 6F01 has the same effect in both genotypes or if it is effective only in the Lrig1-expressing mice.

8. Regarding the possible clinical utility of Lrig1-directed therapies, the possible side effects are central. To this end, the authors state that "...the fact that Lrig1^{-/-} mice did not show any significant physiological abnormality may hold the promise of the minimal cytotoxicity of mAb specific to Lrig1 upon therapeutic application". However, the authors should also cite other's work, showing for example that Lrig1^{-/-} mice suffer from psoriasis-like lesions, intestinal hyperplasia, intestinal cancer, deficits in social behavior, and blindness.

Other general and specific points:

1. Some of the FACS results are anecdotal in the sense that no quantifications or experimental statistics are provided. It is desirable to quantify also FACS results and to provide statistics on their experimental variability.

2. Regarding the Lrig1 knockout mouse strain, it is only stated that it was obtained from Fiona Watt. The mouse strain needs to be clearly described via more information or through a reference to such a thorough description.
3. In the Methods section, it appears strange that one of the authors “kindly provided” mice to the study. Respective author’s contribution is better listed in the Authors contribution section.
4. The pSmad2/3 assay is not clearly described and should be described in more detail.
5. Regarding the role of TGF-beta, LRIG1 was recently shown to promote signaling induced by the TGF-beta superfamily members BMP4 and BMP6 ,but not by TGF-beta 1 itself [Herdenberg et al., 2021]. This information seems highly relevant to the present study and should be included and discussed in the manuscript.
6. The qRT-PCR experiments suffer from methodological issues. First, it is problematic that only one reference gene is used in each experiment. How does the authors know that Lrig1 goes up rather than the reference gene is going down, or vice-versa? Second, why is a different reference gene used for Foxp3?

Title: Lrig1 essential for alleviating autoimmunity defines the suppressive population of CD4⁺ T cells through signaling via the Smad2/3/Foxp3 axis

We appreciate your excellent review of our manuscript. Your and two reviewers' valuable comments helped us to make a better manuscript. The significant changes were made or added in the revised manuscript to answer the reviewers' comments. Please find our explanations and answers that may hopefully resolve the issues raised by the reviewers.

REVIEWER COMMENTS

Reviewer #1 (Remarks to the Author):

The manuscript entitled "LRIG1 essential for alleviating autoimmunity defines the suppressive population of CD4⁺T cells through signaling via the Smad2/3/Foxp3 axis" investigated the functional role of a new marker, LRIG1, in regulatory T cell (Treg) function. The authors showed that LRIG1 is mainly expressed on activated Treg cells and is involved in Treg cell function. The authors also raised a potential therapeutic role of LRIG1 in EAE. Given the pivotal role of Treg cells in maintaining immune homeostasis and in several diseases, this study can be an important finding and can contribute to a better understanding of the fundamental aspects of Treg cell biology. However, the following issues need to be addressed:

Major comments:

(Comment 1.1) The authors claim that targeting LRIG1 presents a therapeutic potential. However, the authors didn't treat the mice with anti-LRIG1 and anti-IL-17A in the same manner. The authors should use the same therapeutic strategy for 6F01 and anti-IL17A to analyze the therapeutic potential of modulating LRIG1.

Response 1.1 =>

We appreciate the Reviewer's comment on the therapeutic effect of 6F01. As the Reviewer suggested, we have repeated EAE *in vivo* efficacy test of the anti-Lrig1 antibody (6F01) using the same treatment strategy with anti-IL-17A monoclonal antibody (mAb) to address this issue. After induction of the EAE in the animal model, we treated EAE animal models with 50 µg of isotype control IgG2a, 6F01, or anti-IL-17A mAb three times on days 2, 4, and 6 of EAE onset. The clinical score of EAE disease was assessed daily for 27 days after EAE induction. We also examined the population of Lrig1⁺Foxp3⁺, total Foxp3⁺, IFNγ⁺, or IL-17A⁺ in CD4⁺ T cells in the lymph node or spinal cord of the treated mice using flow cytometry.

As shown in the revised Figure 6, we demonstrated that the therapeutic potential of 6F01 mAb treatment was comparable to that of anti-IL-17A mAb treatment in the EAE animal model. Also, we confirmed that the level of the suppressive population such as Lrig1⁺Foxp3⁺ or total Foxp3⁺ T cells was significantly increased in response to 6F01 treatment. And, the percentage of inflammatory populations was decreased by the treatment of 6F01 similar to IL-17A blockade. In addition, the level of spinal cord-infiltrated T cells and demyelination in the spinal cord was reduced by the treatment of 6F01. These results suggest **that** Lrig1-targeting strategy in autoimmune disease has a strong therapeutic potential similar to anti-IL-17A mAb treatment.

Revised Fig. 6

Amendments in the manuscript: In Results section, Page 12, Line 220-233

(Comment 1.2) The authors propose that activation of LRIG1 induces activation of p-Smad2/3 and increases Foxp3 mRNA expression through an Akt/mTOR/Foxo mechanism. The authors didn't prove that and should analyze more carefully the signaling following LRIG1 activation and the outcome of this stimulation.

Response 1.2 =>

We appreciate the Reviewer's comment on the proposed molecular mechanism of Lrig1 signaling. It has been known that Lrig1 interacts with EGFR and induces the degradation of EGFR by recruiting E3 ligase c-Cbl to the cytoplasmic domain of EGFR in the cancer cells¹. Also, EGFR has an immunological role in inflammatory settings mediated by its downstream signaling molecules such as AKT or mTOR². Phosphorylated AKT induced by EGFR stimulation inhibits Smad3 phosphorylation, suppressing TGF-β-mediated Foxp3 expression and also inhibit the functions of FoxO1 protein essential for Foxp3 expression and activates mTORC1 pathway to suppress Foxp3 expression in T_{reg} cells³⁻⁶. Therefore, we hypothesized that in T_{reg} cells, Lrig1 stimulation induces EGFR degradation by c-Cbl recruitment and further inhibits AKT and mTOR phosphorylation, thereby increasing Foxp3 expression by

lifting the inhibition of AKT-mediated suppression of Smad3 or FoxO1 signaling. We have carefully investigated this hypothesis and included the data in the revised Figure 5e-h.

First, we examined the level of EGFR on the surface of T_{reg} cells using flow cytometry, and EGFR expression in T_{reg} cells was analyzed by western blot. As shown in the revised Fig. 5e-h, EGFR expression in iT_{reg} cells was reduced by stimulating Lrig1 with Lrig1-specific monoclonal antibody 6F01. In addition, the treatment of CD4⁺Foxp3⁺ T cells with 6F01 antibody significantly inhibited the phosphorylation of AKT, mTOR, and FoxO1 in a concentration-dependent manner. We assume that the down-regulation of EGFR expression on the surface of T_{reg} cells by Lrig1 stimulation mediates the induction of Foxp3 expression via suppressing the inhibitory signals for Foxp3 expression.

Revised Fig. 5e-h

Amendments in the manuscript: In Results section, Page 12, Line 205-219, In Discussion section, Page 17, Line 322-335.

(Comment 1.3) The CD45RB^{High} colitis model was performed using total CD4⁺ sufficient or deficient for LIRG1. This model should be performed using LIRG1⁺ or LIRG1⁻ Tregs cells instead of total CD4⁺ cells. The conclusion of the current experiment is not clear and doesn't show that LIRG1⁺ Treg cells present a better suppressive capacity.

Response 1.3 =>

We thank the Reviewer for the valuable comment. Our report demonstrated that Lrig1-expressing CD4⁺ T cells have a strong suppressive activity by highly expressing anti-inflammatory molecules such as Foxp3, IL-10, or PD-1, as presented in Figures 1, 2, and 4. Therefore, we performed the CD45RB^{hi} cell-mediated colitis model using Lrig1⁺ or Lrig1⁻ cells in total CD4⁺ T cells to see whether the Lrig1⁺ population in CD4⁺ T cells can inhibit autoimmune symptoms *in vivo* model, and the results were included in Fig. 4a-f.

We have performed the new experiment by transferring *Lrig1*-expressing or *Lrig1*-knockout (KO) iT_{reg} cells into the CD45RB^{hi}-mediated inflammatory bowel disease (IBD) model, as the Reviewer suggested. In the revised Supplementary Fig. 8, we showed that the adoptive transfer of iT_{reg} cells from *Lrig1*-knockout mice didn't alleviate the IBD symptoms well and suppress the colonic inflammation induced by CD45RB^{hi} T cell transfer when compared to that iT_{reg} cells from *Lrig1*^{+/-} mice. Further, we observed that *Lrig1*^{+/-} iT_{reg} cell-recipient mice showed decreased histological scores of colon inflammation compared to the recipients of CD45RB^{hi} cell only or *Lrig1* KO iT_{reg} cells. Consistent with CD4⁺Lrig1^{high} or Lrig1^{low} cell transfer results, *Lrig1*-expressing iT_{reg} cell-recipient had a higher percentage of Foxp3⁺ T cells in intraepithelial lymphocytes (IEL) in the colon and also mesenteric lymph node (mLN) than *Lrig1*-knockout iT_{reg} cell recipient. Furthermore, the proportion of CD4⁺IFN γ ⁺ T cells was significantly decreased by *Lrig1*^{+/-} iT_{reg} cell transfer in both CD45.1⁺ and CD45.1⁻ cells in IEL and mLN compared to *Lrig1*-deficient iT_{reg} cell transfer. These results demonstrate that *Lrig1*-expressing iT_{reg} cells have a higher suppressive activity to reduce inflammatory symptoms in the colitis model than *Lrig1*-deficient iT_{reg} cells.

Updated Supplementary Fig. 8

Amendments in the manuscript: In Results section, Page 10, Line 157-166.

(Comment 1.4) Fig2A: The quantification of the in vitro suppressive assay is missing. A careful analysis of the CD4⁺ LIRG1⁺ cells is necessary to be able to interpret these results: percentage and MFi of Foxp3, CTLA4, CD25 etc.

Response 1.4 =>

We appreciate the Reviewer's point and now provide the quantification of effector cell proliferation activity *in vitro* suppression assay. As shown in updated Fig. 2A (see below), Lrig1-expressing CD4⁺ and CD4⁺Foxp3⁺ T cells have a significantly higher suppressive activity than CD4⁺Lrig1⁻ T cells. This result suggests that the Lrig1⁺ population in CD4⁺ T cells represents a strong suppressive subpopulation of CD4⁺ T cells like conventional T_{reg} cells. In addition to this quantification, we analyzed the expression of T_{reg} cell markers such as CD25, PD-1, CTLA-4, or Foxp3 in CD4⁺Lrig1⁻, Lrig1⁺, or Foxp3⁺ T cells to investigate the suppressive function of CD4⁺Lrig1⁺ T cells. Interestingly, we found that CD4⁺Lrig1⁺ T cells show a higher expression level of these T_{reg} cell markers than CD4⁺Lrig1⁻ T cells, and the expression level of these markers in CD4⁺Lrig1⁺ T cells was similar to that in CD4⁺Foxp3⁺ T cells except CD25 expression (Revised Supplementary Fig. 3). Based on these results (Fig. 1f, 2b, 4g and Supplementary Fig. 3), we found that Lrig1⁺ T cells had a suppressive potential comparable to Foxp3⁺ T cells by highly expressing T_{reg} cell-associated regulatory markers.

Updated Fig. 2a

Revised Supplementary Fig. 3

Amendments in the manuscript : In Results section, Page 7, Line 84-91.

(Comment 1.5) Fig2 C: the panel's results are not clear. Can you please quantitate those results?

Response 1.5 =>

We thank the Reviewer for this suggestion. We included the quantification graph for the Fig. 2c data. Compared to $Lrig1^{low}$ cells, $Lrig1^{hi}$ cells among $CD4^{+}Foxp3^{+}$ T cells exhibit a higher suppressive activity even though all cells express a similar level of Foxp3. Based on this result, we demonstrate that $Lrig1$ is a representative marker of a strong suppressive subpopulation in T_{reg} cells.

Updated Fig. 2c

(Comment 1.6) Fig3 C: WT Tregs should be added in this experiment, the level of suppression of the $LRI G1^{+/-}$ is really low

Response 1.6 =>

We appreciate the Reviewer's suggestion. We repeated the *in vitro* suppression assay and now included the data for the suppression level of WT T_{reg} cells ($Lrig1^{+/+}$) in the updated Fig. 3d with the representative flow cytometry data and graph (See updated Fig. 3d). There was a significant difference between $Lrig1^{+/+}$ and $Lrig1^{+/-}$ or $Lrig1^{-/-}$ iT_{reg} cells in each effector: T_{reg} ratio. WT T_{reg} cells showed the highest suppressive function compared to $Lrig1^{+/-}$ and $Lrig1$ -deficient T_{reg} cells.

Updated Fig. 3d

(Comment 1.7) Fig4: a more careful analysis of the RNAs-seq results should be added in the figure: volcano plot or heatmap of the Top 20 differentially expressed genes, KEGG/Go pathway analysis etc.

Response 1.7 =>

We appreciate the Reviewer's suggestion. We carefully analyzed the RNA sequencing data to compare the gene expression profile of $Lrig1^{+}$ with that of $Lrig1^{-}$ in mouse $CD4^{+}$ T cells. Now the heatmap of the top 20 differentially

expressed genes (DEGs), pathway and process enrichment analysis or enrichment analysis using PaGenBase⁷ or TRRUST⁸ with upregulated DEGs in Lrig1⁺ or Lrig1⁻CD4⁺ T cells is provided in the revised Supplementary Fig. 10 and 11.

We identified 468 upregulated DEGs in Lrig1⁺CD4⁺ T cells compared to Lrig1⁻CD4⁺ T cells and 309 upregulated DEGs in Lrig1⁻CD4⁺ T cells compared to Lrig1⁺CD4⁺ T cells. Interestingly, the top 20 DEGs contained various T_{reg} cell markers such as *Foxp3*, *Il2ra*, and *Ctla4*. The expression of these genes was higher in Lrig1⁺ T cells than in Lrig1⁻ T cells in all three samples (Revised Supplementary Fig. 10a). Next, we used the Metascape⁹ to do the enrichment analysis in each gene set. The Metascape carries out pathway and process enrichment analysis with the following ontology sources: GO Biological Processes, KEGG Pathway, Reactome Gene Sets, CORUM, WikiPathways and PANTHER Pathway. In addition to the pathway and process enrichment analysis, we also performed a more careful analysis to investigate which cell type is similar to Lrig1⁺ T cells or Lrig1⁻ T cells in terms of gene expression profile and which transcription factor regulates the expression of DEGs in Lrig1⁺CD4⁺ T cells.

As shown in revised Supplementary Fig. 10b, we demonstrated that the DEGs expressed by the Lrig1⁺ T cells are involved in the upregulation of cytokine production and negative regulation of the immune system process (*Upper panel*). In contrast, the upregulated pathways in Lrig1⁻ T cells are related to immunoglobulin production or lymphocyte activation (*Lower panel*). Surprisingly, our results showed that the cell type with the gene set most similar to the upregulated genes in Lrig1⁺ T cell is FoxP3⁺ T cells (*Upper panel*). And the cell type with the highest relation to the upregulated genes in Lrig1⁻ T cell is B220⁺ B cells (*Lower panel*) as shown in the revised Supplementary Fig. 11a. Moreover, using TRRUST enrichment analysis, we demonstrated that the expression of the upregulated genes in Lrig1⁺ T cell are regulated by Nkfb1, Rela or Foxp3 which is associated with the expression of suppressive molecules in T_{reg} cells (Revised Supplementary Fig. 11b).

Revised Supplementary Fig. 10

Revised Supplementary Fig. 11

Methods:

“For the analysis of DEGs, pathway and process enrichment analysis, including GO Biological Processes, KEGG Pathway, Reactome Gene Sets, CORUM, WikiPathways and PANTHER Pathway, or PaGenBase/ TRRUST-based enrichment analysis, was performed using Metascape, which is a user-friendly and web-based analysis tool. Heatmaps of the Top 20 DEGs or T_{reg} cell marker genes were visualized by Heatmapper.”

Amendments in the manuscript : In Results section, Page 10, Line 171-183. In Methods section, Page 28, Line 576-581.

(Comment 1.8) Based on the impact of *LRI1* on the *Foxp3* mRNA, the authors should analyze the development of the natural thymic T_{reg} cells at an early and late stage.

Response 1.8 =>

We appreciate the Reviewer’s suggestion, and we have shown the functional association between *Lrig1* and the natural thymic T_{reg} cells (tT_{reg}) development using *Lrig1*^{-/-} mice in revised Supplementary Fig. 7. It is known that the tT_{reg} precursor (CD4⁺CD25⁺Foxp3⁻) is developed into tT_{reg} (CD4⁺CD25⁺Foxp3⁺) through the tT_{reg} development process in Thymus^{10,11}. Therefore, we analyzed the populations of tT_{reg} precursor (early stage) and tT_{reg} (late stage) in the thymus of *Lrig1*^{+/+} and *Lrig1*^{-/-} mice by flow cytometry.

As shown in the revised Supplementary Fig. 7a, similar levels of tT_{reg} precursor and the tT_{reg} populations were detected in *Lrig1*^{+/+} and *Lrig1*^{-/-} mice. In addition, the difference in the level of CD4⁺Foxp3⁺ T cell population in the spleen, thymus, and inguinal LNs of *Lrig1*^{+/+} and *Lrig1*^{-/-} mice was not observed (Revised Supplementary Fig. 7b). These results suggest that the absence of *Lrig1* did not affect the development of tT_{reg} cells.

Revised Supplementary Fig. 7

Amendments in the manuscript : In Results section, Page 9, Line 136-144.

Minor Comments:

(Comment 1.9) Statistical analysis is missing in different figures 1, 3,4,5,6, ext2.

Response 1.9 =>

We appreciate the Reviewer's point. We now added the statistical analysis in every figure.

(Comment 1.10) Different fonts on different figures.

Response 1.10 =>

We now updated and unified the font size on each figure.

(Comment 1.11) Fig 1e: the authors should add a representative isotype control

Response 1.11 =>

We appreciate the Reviewer's comment. We now included the isotype control for each group.

Updated Fig. 1e

(Comment 1.12) Fig1f: the IL-10 should be represented IL-10 (y-axis) vs CD4 (x-axis)

Response 1.12 =>

We thank the Reviewer for the suggestion. As the Reviewer suggested, we revised the format of the representative dot plot – IL-10-PE (y-axis) vs. CD4-APC (x-axis), including isotype control-PE antibody in Fig. 1f. IL-10 expression in CD4⁺Lrig1⁺ T cells was significantly higher compared to CD4⁺Lrig1⁻ T cells in the spleen.

Updated Fig. 1f

(Comment 1.13) Fig2B: not the same group between fig 2a and 2b

Response 1.13 =>

We appreciate the Reviewer's suggestion. We repeated the IL-10 ELISA assay using the same group in Fig. 2a and the updated group in the updated Fig. 2b. The level of IL-10 secretion from CD4⁺Lrig1⁺ T cells was significantly higher than that of CD4⁺Lrig1⁻ T cells. This result supports that Lrig1-expressing CD4⁺ T cells have a higher suppressive function than CD4⁺Lrig1⁻ T cells.

Updated Fig. 2b

Amendments in the manuscript : In Results section, Page 7, Line 89-91.

(Comment 1.14) Fig3: the level of LRIG1 should be analyzed and add to the figure

Response 1.14 =>

According to the Reviewer's comment, we analyzed the Lrig1 level on the surface of CD4⁺ T cells after treating the cells with control or the different doses of *Lrig1*-targeting siRNA. The expression level of Lrig1 protein on the surface was significantly reduced by si*Lrig1* treatment in a concentration-dependent manner compared to scrambled siRNA (Revised Fig. 3a).

Revised Fig. 3a

(Comment 1.15) Fig 6 c is not convincing the authors should represent an isotype control and dot plot with better quality and the same gating

Response 1.15 =>

We appreciate the Reviewer's comment. As shown in Comment 1.1, we repeated the EAE animal model using a Lrig1-targeting antibody and analyzed the ratio of Lrig1⁺Foxp3⁺, total Foxp3⁺, IFN γ ⁺ or IL-17A⁺ in CD4⁺ T cells in a lymph node, or spinal cord of the treated mice. In the representative dot plots (see below figures), the increase of Lrig1⁺Foxp3⁺ or total Foxp3⁺ population in lymph node and spinal cord CD4⁺ T cells was observed by the 6F01 treatment or anti-IL-17A mAb treatment compared to isotype IgG2a injection. According to the Reviewer's comment, we added the representative images of the isotype control-PE for the Lrig1-PE antibody using the same gating strategy for all samples in the updated Fig. 6c or data now shown in the manuscript below.

Updated Fig. 6c

Data now shown in the manuscript

(Comment 1.16) Extended figure 1B: isotype control is needed

Response 1.16 =>

We appreciate that isotype control for Lrig1-PE is needed. In the updated Supplementary Fig. 1b, we added the isotype control goat IgG-PE-stained samples. Compared to isotype control-PE, we examined Lrig1 expression on each T cell population using Lrig1-PE antibody.

Updated Supplementary Fig. 1b

(Comment 1.17) Extended figure 1c: quantification is needed

Response 1.17 =>

We appreciate the Reviewer's suggestion. We now provide the quantification of LRIG1 expression in western blot analysis using Image J software. As shown in the revised Supplementary Fig. 1c, the relative expression level of

LRIG1 normalized by β -actin was increased in iT_{reg} cells compared to activated T cells by anti-CD3/28 antibodies, and further Everolimus or Rapamycin stimulation significantly induced the level of LRIG1 in the expanded iT_{reg} cells. On the other hand, the Jurkat T cells did not express LRIG1, and the activation of Jurkat T cells using CD3/28 stimulation also didn't induce the LRIG1 expression.

Revised Supplementary Fig. 1c

(Comment 1.18) Extended figure 4: other markers are needed, CTLA4, PD1, Cd39, Cd73, etc

Response 1.18 =>

As the Reviewer suggested, we examined other T_{reg} cell markers such as TIGIT, CTLA4, and PD-1, including Lrig1 on the surface of *Lrig1*^{+/+} or *Lrig1*^{-/-} iT_{reg} cells. As shown in the updated Supplementary Fig. 5, the expression level of each T_{reg} surface marker was similar in both iT_{reg} cells.

Updated Supplementary Fig. 5

Amendments in the manuscript : In Results section, Page 9, Line 127-129.

(Comment 1.19) Extended figure 6b, Survival curve should be added in the figure

Response 1.19 =>

We thank the Reviewer for this suggestion. We added the survival curve of lupus-prone mice in Supplementary Fig. 9b. It was observed that 20% of the untreated and CD4⁺Lrig1⁻ T cell-transferred mice died at 28 weeks of age.

Updated Supplementary Fig. 9b

Reviewer #2 (Remarks to the Author):

This is an interesting but preliminary study. The authors present what looks like an interesting discovery; that is, they show evidence suggesting that the transmembrane protein Lrig1 both marks and regulates regulatory T cells (Tregs). Furthermore, they generated an anti-Lrig-1 antibody, AF01, that promoted Treg function in vitro and in vivo and suppressed experimental autoimmune diseases. The strength of the paper is the novelty of the Treg-Lrig1 finding and the possible therapeutic potential of the developed Lrig1 antibody. However, on the downside, the paper contains overinterpretation of the data, it is mechanistically shallow, important controls are missing, and there are a series of questionmarks regarding the ‘therapeutic’ antibody.

Specific points regarding the biological discovery:

(Comment 2.1) The authors analyze cytokine-stimulated T cells that they named Th0, Th1, Th2, and Th17 without confirming their respective characteristics. The authors should confirm their subtype identities through transcriptional profiling, cytokine production, or functionality or rename the cell populations according to their treatment rather than their believed phenotypes.

Response 2.1 =>

We appreciate the Reviewer’s point. Whenever we induce the differentiation of naïve T cells into each T cell subset (Th0=activated T cells, Th1, Th2, Th17 or T_{reg} cells) under each T cell subset-specific polarizing culture condition, which was published in numerous previous papers by other groups and our group, the characteristics of each T cell subset including transcription factor expression, cytokine production or functionality is always confirmed before performing the subsequent experiments.

In this report, whenever the level of Lrig1 is measured at the RNA or protein level in each T cell subset, we examined the specific markers, including transcription factors and cytokines of each T cell subset, after inducing the differentiation of each T cell subset; Th0 (activated T cells)= CD69⁺CD25⁺, Th1= T-bet⁺IFN γ ⁺, Th2= GATA3⁺IL-4⁺, Th17= ROR γ t⁺IL-17A⁺, iT_{reg}=Foxp3⁺IL-10⁺ (See the representative analysis below). After the differentiation induction of naïve T cells into each T cell subset, we gated the T cell subset-positive population and analyzed the level of Lrig1 protein on the surface of the highly differentiated T cell subset population to measure the protein levels correctly.

(Comment 2.2) The authors claim in the Title and elsewhere that Lrig1 acts through Smad2/3 signaling, which is not firmly supported by the data presented. The only data pertaining to Smad2/3 that is presented is an experiment showing that treatment with the non-validated anti-Lrig1-antibody 6F01 for 2 or 3 days during iTreg differentiation results in enhanced pSmad2/3 levels (Fig. 5b). There could be numerous different explanations for this result. If the authors wish to investigate the role of Lrig1 and TGF-beta signaling in these T cell populations, they should perform proper signal transduction experiments, including shorter kinetics, specific agonists and inhibitors, and/or genetic (e.g., RNAi) experiments.

Response 2.2 =>

The Reviewer asks an essential question about the mechanism of action of Lrig1 through Smad2/3 signaling in T_{reg} cells. As the Reviewer commented, we demonstrated that Lrig1-targeting monoclonal antibody (6F01) treatment during iT_{reg} cell differentiation induced the expression of Foxp3 (RNA and protein levels) via the phosphorylation of Smad2/3 in the revised Fig. 5a, 5b and 5d. In addition to iT_{reg} cells, we presented that Lrig1 stimulation enhanced the Foxp3⁺ population through the increase of p-Smad2/3 in Th17 differentiation in Supplementary Fig. 13a and b.

We also agree with the Reviewer's point that more supporting data would be needed to explain this mechanism. To address this point, we performed 1) an inhibitor assay using SIS3, which is a potent inhibitor of Smad3 phosphorylation, and 2) a phospho-flow assay to find if 6F01 treatment inhibits AKT phosphorylation which is an upstream suppressor of Smad3 phosphorylation with shorter kinetics.

First, naïve T cells were induced to differentiate into T_{reg} cells under the suboptimal iT_{reg} differentiation condition in the presence of isotype control IgG2a or 6F01 (5 µg ml⁻¹) and were treated with different concentrations of Smad3 phosphorylation inhibitor, SIS3. Then the level of CD4⁺Foxp3⁺ T cell population was examined using flow cytometry. As described in the revised Fig. 5c, the level of CD4⁺Foxp3⁺ T cells with 6F01 treatment was similar to that with isotype control treatment when SIS3 1 or 5 µM was used. But, SIS3-untreated cells showed an increased Foxp3⁺ T cells with 6F01 treatment compared to those with isotype control treatment. This result suggests that the increase of Smad2/3 phosphorylation accomplishes the increment of the Foxp3 level by 6F01 treatment, and this effect was completely abrogated by the blockage of Smad2/3 phosphorylation.

Second, to investigate whether this stimulatory effect of 6F01 is mediated by the down-regulation of inhibitory signaling by phospho-AKT, we analyzed the level of phosphorylated AKT in iT_{reg} cells with shorter kinetics using flow cytometry. Within 1 min to 30 min, the level of AKT phosphorylation in Ser473 was considerably diminished by 6F01 stimulation compared to isotype control or no stimulation, as shown in the revised Supplementary Fig. 15. Therefore, it is confirmed that Lrig1 stimulation using 6F01 inhibits EGFR-AKT signaling in a short time and could induce Smad2/3 phosphorylation to mediate the increase of *Foxp3* expression (Fig. 5).

Revised Fig. 5c

Revised Supplementary Fig. 15

Amendments in the manuscript : In Results section, Page 12, Line 201-204 and 216-217.

(Comment 2.3) Similarly, in Supplementary Figure 8, the authors propose that LRIG1 functions through EGFR-mediated negative regulation of Smad2/3 signaling. This is too speculative given that no supportive data is provided. To this end, it is suggested that the authors address the role of EGF signaling in their experimental system.

Response 2.3 =>

We appreciate the Reviewer's comment on the proposed molecular mechanism of Lrig1 signaling. It has been known that Lrig1 interacts with EGFR and induces the degradation of EGFR by recruiting E3 ligase c-Cbl to the cytoplasmic domain of EGFR in the cancer cells¹. Also, EGFR has an immunological role in inflammatory settings mediated by its downstream signaling molecules such as AKT or mTOR². Phosphorylated AKT induced by EGFR stimulation inhibits Smad3 phosphorylation, suppressing TGF-β-mediated Foxp3 expression and also inhibit the

functions of FoxO1 protein essential for Foxp3 expression and activates mTORC1 pathway to suppress Foxp3 expression in T_{reg} cells³⁻⁶. Therefore, we hypothesized that in T_{reg} cells, Lrig1 stimulation induces EGFR degradation by c-Cbl recruitment and further inhibits AKT and mTOR phosphorylation, thereby increasing Foxp3 expression by lifting the inhibition of AKT-mediated suppression of Smad3 or FoxO1 signaling. We have carefully investigated this hypothesis and included the data in the revised Figure 5e-h.

First, we examined the level of EGFR on the surface of T_{reg} cells using flow cytometry, and EGFR expression in T_{reg} cells was analyzed by western blot. As shown in the revised Fig. 5e-h, EGFR expression in iT_{reg} cells was reduced by stimulating Lrig1 with Lrig1-specific monoclonal antibody 6F01. In addition, the treatment of CD4⁺Foxp3⁺ T cells with 6F01 antibody significantly inhibited the phosphorylation of AKT, mTOR, and FoxO1 in a concentration-dependent manner. We assume that the down-regulation of EGFR expression on the surface of T_{reg} cells by Lrig1 stimulation mediates the induction of Foxp3 expression via suppressing the inhibitory signals for *Foxp3* expression.

Revised Fig. 5e-h

Amendments in the manuscript : In Results section, Page 12, Line 205-219, In Discussion section, Page 17, Line 322-335.

To further demonstrate the relationship between Lrig1 and EGFR signaling in T_{reg} cells, we used the recombinant EGF to activate EGFR signaling pathway. However, when recombinant EGF was included during iT_{reg} differentiation induction, there was no change in the level of Foxp3⁺ T cells by the different concentrations of EGF treatment (see the figure below). This could be due to the low level of EGFR expression in normal T_{reg} cells or that other ligands could be used for EGFR signaling in T_{reg} cells. Alternatively, it may be possible that other or unidentified members of the receptor tyrosine kinase (RTK) family, which can activate the AKT-mTOR signaling pathway, can be involved in the Lrig1-mediated intracellular signaling pathway.

Data not shown in the manuscript

*(Comment 2.4) The anti-LRIG1 antibody from R&D used in FACS is a polyclonal antibody raised against the entirety of the extracellular/luminal part of LRIG1. Therefore, it is likely that the antibody crossreacts with other proteins. And the antibody do not seem to be well validated. Although the siRNA experiment in Fig. 3a looks good, it would be desirable to confirm the antibody's specificity via side-by-side experiments including cells both from wild-type and *Lrig1* knockout mice.*

Response 2.4 =>

We appreciate the Reviewer's point. As the Reviewer suggested, we confirmed the antibody specificity with iT_{reg} cells isolated from *Lrig1* wild-type and knockout mice. Consistent with the *Lrig1* siRNA experiment, the anti-Lrig1 polyclonal antibody specifically detects Lrig1 expression on the surface of iT_{reg} cells from *Lrig1* WT mice. The Lrig1 antibody doesn't bind to *Lrig1*-deficient iT_{reg} cells with a similar level of isotype control (See the Figure below). Also, we now provide the results about Lrig1 expression level in *Lrig1*^{+/-} and *Lrig1*^{-/-} mice in updated Supplementary Fig. 5, including isotype control-PE or other T_{reg} cell marker staining. Anti-Lrig1-PE polyclonal antibody only stained *Lrig1* WT and *Lrig1*^{+/-} T_{reg} cells.

Data not shown in the manuscript

Updated Supplementary Fig. 5

(Comment 2.5) The generation of 6F01 is not adequately described. The detailed methods should be included, such as information about the epitope used for its selection and the validation of its specificity.

Response 2.5 =>

As the Reviewer commented, we now add the detailed methods, including an antigen used for the immunization and the selection and the validation process to check if this antibody binds specifically to Lrig1 in the Method section. Also, the flow cytometry data is provided to evaluate the binding affinity of affinity-maturation clones, showing the 6F01 antibody has the highest binding affinity to mouse iT_{reg} cells among affinity-matured clones.

The amendments were made in Methods section of the manuscript:

Methods:

“Anti-Lrig1 monoclonal antibodies were generated using the antibody phage display and the panning method using mouse *Lrig1* gene transfected-L cell as an antigen. After validating the antibody with high binding affinity among the generated anti-Lrig1 monoclonal antibodies through flow cytometric analysis, affinity maturation was performed

through light-chain shuffling using the mouse Lrig1-ectodomain. Finally, 6F01 with high binding affinity to both mouse and human Lrig1 was selected through ELISA and FACS analysis. The cDNA clone of 6F01 was transfected into HEK293-F cells, and the 6F01 mAb was purified by Y-Biologics (South Korea).”

Amendments in the manuscript : In Methods section, Page 29, Line 591-599.

(Comment 2.6) Regarding the possible effects of 6F01 on Lrig1, it should be noted that the function of Lrig1 in the system under study is not known. Therefore, at this stage, it cannot be concluded whether 6F01 “activates” or “inactivates” Lrig1, both scenarios are possible.

Response 2.6 =>

We appreciate the Reviewer’s point. To answer the Reviewer’s Comment 2.3, we investigated the effect of 6F01 on the EGFR signaling pathway. As described in Comment 2.3, it is confirmed that 6F01 treatment could induce the degradation of EGFR in T_{reg} cells and inhibit the phosphorylation of AKT, mTOR, and FoxO1. Based on these results, we suggest that 6F01 may stimulate Lrig1 on the surface and enhance the functions of the Lrig1-mediated intracellular signaling pathway, leading to the inhibition of EGFR signaling and phosphorylation of AKT, mTOR, and FoxO1 and subsequently inducing *Foxp3* expression.

(Comment 2.7) 6F01 showed promising activities in vivo, however, important controls were missing. For example, it should be important to establish whether the antibody acts through Lrig1 or via other mechanisms. To this end, the authors could take advantage of their wild-type and Lrig1 knockout mice and investigate if 6F01 has the same effect in both genotypes or if it is effective only in the Lrig1-expressing mice.

Response 2.7 =>

We appreciate the Reviewer’s point about the specificity of 6F01. As the Reviewer mentioned in the comment, we used *Lrig1*^{-/-} mice to investigate whether 6F01 has a function through Lrig1 binding. To confirm whether 6F01 has the specificity for Lrig1, naïve T cells obtained from *Lrig1* WT or KO mice were differentiated into iT_{reg} cells. Then the differentiated iT_{reg} cells were treated with 6F01, and the level of binding percent was analyzed by flow cytometry. As a result, 6F01 only bound to *Lrig1*^{+/+} iT_{reg} cells expressing Lrig1 (See below figure). In addition, we investigated the function of 6F01 in both *Lrig1*^{+/+} and *Lrig1*^{-/-} iT_{reg} cells. When *Lrig1*^{+/+} iT_{reg} cells were treated with 6F01, the Foxp3⁺ T cell population increased, and phosphorylation of AKT or mTOR was inhibited. However, 6F01 did not show the inhibitory functions when treating *Lrig1*^{-/-} iT_{reg} cells (See below figure).

Data not shown in the manuscript

(Comment 2.8) Regarding the possible clinical utility of Lrig1-directed therapies, the possible side effects are central. To this end, the authors state that “...the fact that Lrig1^{-/-} mice did not show any significant physiological abnormality may hold the promise of the minimal cytotoxicity of mAb specific to Lrig1 upon therapeutic application”. However, the authors should also cite other’s work, showing for example that Lrig1^{-/-} mice suffer from psoriasis-like lesions, intestinal hyperplasia, intestinal cancer, deficits in social behavior, and blindness.

Response 2.8 =>

We agree with the Reviewer’s comment that references and descriptions of previous findings related to some defects in Lrig1 knockout mice need to be mentioned in our manuscript. Earlier studies on Lrig1-deficient mice have mainly investigated the defects induced by non-immune cell populations like intestinal or epithelial stem cells or neuronal cells.

In previous studies, Lrig1 is known to be expressed in intestinal or epithelial stem cells and some cancer cells: The reports from Nakamura et al. (Corneal stem cell – corneal blindness¹²), Alsina et al. (Neuron - defects in social interaction¹³), Wong et al. (Intestinal stem cell – enlarged crypt¹⁴), Powell et al. (Intestinal stem cell - intestinal cancer¹⁵) and Suzuki et al. (Epithelial stem cell - psoriasisiform epidermal hyperplasia¹⁶) are currently cited in our manuscript to show the examples of Lrig1^{-/-} mice defects.

Our study is the first case to investigate the functionality of Lrig1 in immune cells, especially in T_{reg} cells, *in vitro* and *in vivo*, which other studies might neglect. Importantly, Lrig1 is highly specific to T_{reg} cells, and the level of Lrig1 is higher in the activated T_{reg} cells than in the resting T_{reg} cells. These results suggest that 6F01 mAb showed significant therapeutic potential in the autoimmune setting.

Most abnormalities shown in *Lrig1*^{-/-} mice may be related to the uncontrolled proliferation of stem cells in intestinal or epithelial areas or cancer cells. In contrast to the *Lrig1*-deficient condition, 6F01 enhances the *Lrig1*-mediated functions. And, as with other surface proteins expressed in both immune cells and non-immune cells, the function of *Lrig1* in T cells might be unique and different from that in other non-immune cells.

In addition to the fact that *Lrig1*^{-/-} mice did not exhibit any significant immune-related abnormality, our results showing that 6F01 mAb did not show any single dose- and repeated dose-toxicity may hold the careful promise of minimal toxicity upon therapeutic application to autoimmunity. Currently, the standard toxicity study in rodents and monkeys is being undertaken with 6F01 mAb produced by the GMP facility.

Other general and specific points:

(Comment 2.9) Some of the FACS results are anecdotal in the sense that no quantifications or experimental statistics are provided. It is desirable to quantify also FACS results and to provide statistics on their experimental variability.

Response 2.9 =>

We appreciate the Reviewer's point. Some results in the previous manuscript version didn't have a statistical analysis by mistake. We now add the experimental statistics in every graph and the quantification data.

(Comment 2.10) Regarding the Lrig1 knockout mouse strain, it is only stated that it was obtained from Fiona Watt. The mouse strain needs to be clearly described via more information or through a reference to such a thorough description.

Response 2.10 =>

According to the Reviewer's comment, we included the reference to describe the *Lrig1* knockout mouse strain. The strain used in this study was provided by Prof. Fiona Watt and is the same strain used in previous publications.

Methods:

"*Lrig1* heterozygous and knockout (B6.129-*Lrig1*^{tm1.1Hhed}) mice were generated and characterized as previously described. When used, the *Lrig1*-null mice did not have any abnormalities and were age-sex matched with *Lrig1*^{+/-} mice."

Amendments in the manuscript : In Methods section, Page 20, Line 374-376.

(Comment 2.11) In the Methods section, it appears strange that one of the authors "kindly provided" mice to the study. Respective author's contribution is better listed in the Authors contribution section.

Response 2.11 =>

We thank the Reviewer and agree that the sentence should be listed in the Authors contribution section. As described in Response 2.10, we added the mouse information in the Methods section, and the sentence was moved to the Authors contribution section.

(Comment 2.12) The pSmad2/3 assay is not clearly described and should be described in more detail.

Response 2.12 =>

We now described the pSmad2/3 assay in more detail in the Method section.

Methods:

“Mouse naive CD4⁺ T cells (2 x 10⁵ cells/well) were differentiated under suboptimal iTreg-skewing condition (TGF-β1 0.5 ng ml⁻¹ and IL-2 20 ng ml⁻¹) in each 96 flat-bottom well coated with 5 μg ml⁻¹ of mouse IgG isotype control (Invitrogen), 5 μg ml⁻¹ or 10 μg ml⁻¹ of 6F01 along with 1 μg ml⁻¹ of anti-CD3e antibody. 1 μg ml⁻¹ of anti-CD28 antibody was added into the culture medium. After 2 or 3 days of differentiation, the cells were harvested and were fixed/permeabilized using FIX & PERM™ Cell Permeabilization Kit (Thermo Fisher Scientific). The fixed cells were stained with anti-CD4-BV421 or APC, anti-Foxp3-FITC or APC, anti-p-Smad2/3-PE, anti-AKT antibody (Cell Signaling Technology), anti-p-AKT-APC, anti-mTOR antibody (Cell Signaling Technology), anti-p-mTOR-PE-cy7, anti-FoxO1 antibody (Cell Signaling Technology), anti-p-FoxO1 antibody (Cell Signaling Technology). Anti-Rabbit IgG-PE secondary antibody (Invitrogen) was used for unconjugated antibodies. The stained cells were measured by Flow cytometry. iT_{reg} cells differentiated under optimal iT_{reg}-skewing condition (TGF-β1 5 ng ml⁻¹ and IL-2 20 ng ml⁻¹) were used as a positive control. To analyze p-AKT shorter kinetics, 3 days differentiated iT_{reg} cells were cultured for the indicated time in isotype IgG (5 μg ml⁻¹) or 6F01 (5 μg ml⁻¹) coated wells. The cells were harvested and were fixed/permeabilized using FIX & PERM™ Cell Permeabilization Kit (Thermo Fisher Scientific). The fixed cells were stained with anti-p-AKT-APC and were analyzed by Flow cytometry.”

Amendments in the manuscript : In Methods section, Page 29, Line 600-618.

(Comment 2.13) Regarding the role of TGF-beta, LRIG1 was recently shown to promote signaling induced by the TGF-beta superfamily members BMP4 and BMP6 ,but not by TGF-beta 1 itself [Herdenberg et al., 2021]. This information seems highly relevant to the present study and should be included and discussed in the manuscript.

Response 2.13 =>

We agree with the Reviewer’s comment that we need to include the reference and descriptions of previous findings related to the association of LRIG1 and BMP signaling in lipid metabolism.

The report from Herdenberg *et al.*¹⁷ is cited and discussed in our manuscript.

Herdenberg *et al.* describe the function of LRIG proteins, including LRIG1 as regulators of lipid metabolism in adipocyte differentiation of fibroblast. This study generated *Lrig*-null mouse embryonic fibroblasts (MEF) and found that the role of LRIG proteins promotes adipogenesis when treated with an adipogenic cocktail. Further, they showed that *Lrig*-deficient MEFs had a lower sensitivity of BMP4 and BMP6 compared to wild type. The BMP sensitivity was recovered by ectopic LRIG1 and LRIG3 expression by analyzing pSmad1/5 phosphorylation with immunocytochemistry. On the other hand, the phosphorylation level of Smad3 in response to TGF-β1 was not changed in *Lrig*-null MEFs, suggesting that LRIG proteins, especially LRIG1, are associated with adipogenesis via BMP signaling, not required for TGF-β or receptor tyrosine kinase signaling.

This study is highly relevant to our paper, and the amendments were made in the **Discussion section:**

“This study revealed that the expression and function of *Lrig1* are dependent on TGF-β1. And we propose that there is a possibility that bone morphogenetic protein (BMP) signaling, one of the TGF-β superfamily members, may also be involved in *Lrig1* signaling in T_{reg} cells. The previous report has shown that LRIG proteins, including LRIG1,

regulate lipid metabolism via BMP signaling through the phosphorylation of Smad1/5 in mouse embryonic fibroblasts, suggesting LRIG proteins may play an important role as BMP sensitizers. Furthermore, several studies have found that BMP signaling through BMPRI α controls T_{reg} lineage and stability via Smad1/5/8 phosphorylation, and BMPRI α is highly expressed in activated T cells and T_{reg} cells compared to naïve CD4⁺ T cells. In addition, TGF- β and BMP signals have a synergistic effect on the induction of iT_{reg} cells even though the TGF- β signaling pathway is a key factor for the development of Foxp3⁺ T_{reg} cells. Thus, we hypothesize that Lrig1 may play a role in the expression of suppressive markers through both TGF- β and BMP signaling to induce the phosphorylation of Smad1/5/8 from BMP and Smad2/3 from TGF- β . The molecular basis for the association of Lrig1 and BMP signaling remains to be elucidated.”

Amendments in the manuscript : In Discussion section, Page 18-19, Line 344-357.

(Comment 2.14) The qRT-PCR experiments suffer from methodological issues. First, it is problematic that only one reference gene is used in each experiment. How does the authors know that Lrig1 goes up rather than the reference gene is going down, or vice-versa? Second, why is a different reference gene used for Foxp3?

Response 2.14 =>

We appreciate the Reviewer’s point.

For the first part, we showed in the graph below that the expression of the reference gene, *Hprt*, was not changed, and there was no significant difference for each sample. Therefore, we could confirm that the expression of *Lrig1* was different for each cell type, not by the change of the reference gene expression (*Hprt*) in Fig. 1a.

For the second question, we agree with the reviewer’s concerns. To resolve this issue, we re-verified the relative expression of *Foxp3* using *Hprt* as a first reference gene used for the comparative analysis of *Lrig1* expression and *GAPDH* as a second reference gene. As shown in the updated Fig. 5d, it was confirmed that the relative expression level of *Foxp3* was increased in response to the Lrig1-targeting antibody (6F01) compared to isotype IgG treatment regardless of the reference genes.

Data not shown in the manuscript

Updated Fig. 5d

Reference

- 1 Wang, Y., Poulin, E. & Coffey, R. LRIG1 is a triple threat: ERBB negative regulator, intestinal stem cell marker and tumour suppressor. *British journal of cancer* **108**, 1765-1770 (2013).
- 2 Hennessy, B. T., Smith, D. L., Ram, P. T., Lu, Y. & Mills, G. B. Exploiting the PI3K/AKT pathway for cancer drug discovery. *Nature reviews Drug discovery* **4**, 988-1004 (2005).
- 3 Ohkura, N. & Sakaguchi, S. Foxo1 and Foxo3 help Foxp3. *Immunity* **33**, 835-837 (2010).
- 4 Conery, A. R. *et al.* Akt interacts directly with Smad3 to regulate the sensitivity to TGF- β -induced apoptosis. *Nature cell biology* **6**, 366-372 (2004).
- 5 Kasper, I. R., Apostolidis, S. A., Sharabi, A. & Tsokos, G. C. Empowering regulatory T cells in autoimmunity. *Trends in molecular medicine* **22**, 784-797 (2016).
- 6 Haxhinasto, S., Mathis, D. & Benoist, C. The AKT-mTOR axis regulates de novo differentiation of CD4+ Foxp3+ cells. *The Journal of experimental medicine* **205**, 565-574 (2008).
- 7 Pan, J.-B. *et al.* PaGenBase: a pattern gene database for the global and dynamic understanding of gene function. *PloS one* **8**, e80747 (2013).
- 8 Han, H. *et al.* TRRUST v2: an expanded reference database of human and mouse transcriptional regulatory interactions. *Nucleic acids research* **46**, D380-D386 (2018).
- 9 Zhou, Y. *et al.* Metascape provides a biologist-oriented resource for the analysis of systems-level datasets. *Nature communications* **10**, 1-10 (2019).
- 10 Goldstein, J. D. *et al.* Role of cytokines in thymus-versus peripherally derived-regulatory T cell differentiation and function. *Frontiers in immunology* **4**, 155 (2013).
- 11 Nie, J., Li, Y. Y., Zheng, S. G., Tsun, A. & Li, B. FOXP3+ Treg cells and gender bias in autoimmune diseases. *Frontiers in immunology* **6**, 493 (2015).
- 12 Nakamura, T. *et al.* LRIG1 inhibits STAT3-dependent inflammation to maintain corneal homeostasis. *The Journal of clinical investigation* **124** (2013).
- 13 Alsina, F. C. *et al.* Lrig1 is a cell-intrinsic modulator of hippocampal dendrite complexity and BDNF signaling. *EMBO reports* **17**, 601-616 (2016).
- 14 Wong, V. W. *et al.* Lrig1 controls intestinal stem-cell homeostasis by negative regulation of ErbB signalling. *Nature cell biology* **14**, 401-408 (2012).
- 15 Powell, A. E. *et al.* The pan-ErbB negative regulator Lrig1 is an intestinal stem cell marker that functions as a tumor suppressor. *Cell* **149**, 146-158 (2012).
- 16 Suzuki, Y. *et al.* Targeted disruption of LIG-1 gene results in psoriasiform epidermal hyperplasia. *FEBS letters* **521**, 67-71 (2002).
- 17 Herdenberg, C. *et al.* LRIG proteins regulate lipid metabolism via BMP signaling and affect the risk of type 2 diabetes. *Communications biology* **4**, 1-15 (2021).

REVIEWER COMMENTS

Reviewer #1 (Remarks to the Author):

The authors have satisfactorily addressed the issues I have raised in my review.

Reviewer #2 (Remarks to the Author):

This remains an interesting study. Many of the concerns previously raised by me have been addressed adequately; however, some major and minor issues remain to be addressed.

MAJOR AND MINOR COMMENTS:

1. Fig. 1b, c, and elsewhere, the authors discuss Lrig1 levels, although the data rather show the fractions of Lrig1-positive cells. Percentage of Lrig1-positive cells should not be confused with Lrig levels.
2. The authors claim that Lrig1 “defines”, is “specific” for, or “essential” for Treg cells. These claims are confusing given that Lrig1 is expressed by many other cell types in addition to the Tregs, and not all Tregs express Lrig1. Hence, it is suggested that less confusing expressions are used, indicating that enrichment and levels are the defining features, rather than suggesting absolute specificity or dependency.
3. Fig. 2c, e, f. What does “Effector only” at the different effector:regulatory ratios mean? Most of these “Effector onlys” show 80-90% proliferation. 80-90% in relation to what – what does 100% proliferation represent?
4. Fig. 3a, b. The Lrig1 siRNA reduced the fraction of Lrig1-positive CD4+ T cells and the proliferation of the effector cells. However, siRNAs are known to have prominent off-target effects. To account for the possible off-target effects, at least two different siRNAs against the target mRNA should be used. I apologize for not having noticed this shortcoming in my original review.
5. It is claimed that “the body weight... [was] ...unchanged (Fig. 3f, Supplementary Fig. 6a)” (lines 132-133). However, I do not think any body weight data are presented in these figures or elsewhere.
6. It is claimed that “...the level of various T cell subsets in the spleen were unchanged (Fig. 3f, Supplementary Fig. 6a)” (lines 132-133). Here, the authors analyzed IFN γ , IL-4, IL-17A, and Foxp3 in the CD4+ splenocytes in three Lrig1 $^{+/+}$ mice and three Lrig1 $^{-/-}$ mice. According to Fig. 3f, the Lrig1 $^{-/-}$ mice show a downregulation of IL-4+ cells by approximately 70% and of IL-17A+ cells by approximately 25%. Nevertheless, based on a “two-way ANOVA with Tukey’s multiple comparisons test”, the authors conclude that there was no difference between the genotypes. However, in my opinion, this is not an appropriate statistical test for comparing means of two different populations (Lrig1 $^{+/+}$ vs. Lrig1 $^{-/-}$). Hence, with

appropriate statistical testing, e.g., t-tests, it would not be surprising if the Lrig1^{-/-} mice actually show significant changes in the T cell subset populations.

7. The mechanistic claim that Lrig1 should induce Tregs via the downregulation of EGFR, resulting in reduced pAKT signaling, which in turn would enhance TGFb-pSmad2/3 signaling that drives the differentiation of the Tregs remains speculative and is not firmly supported by the data.

(i) First, the whole model seems to be mostly based on data obtained by cultivating the T cells on a surface coated with the Lrig1 antibody 6F01. There is no evidence suggesting that this stimulus corresponds to a physiological signal. This follows from the fact that there is no known physiological ligand for Lrig1 and in fact very little is known about the actual molecular function of Lrig1. Hence, plastic surface-coated antibodies might indeed stimulate the physiological function of Lrig1, but it may equally well inhibit its function or do something completely different. Hence, it cannot be concluded that this artificial stimulus “stimulates the function of Lrig1”. It also follows that one cannot assume that 6F01 is agonistic, because it might equally well be antagonistic or something else.

(ii) Second, regarding the downregulation of EGFR after 2-3 days of cultivation, this represents a correlation rather than implying causation. The fact that EGF did not affect the differentiation of the Tregs may rather suggest (but not prove) that regulation of EGFR is not involved. The same argument holds for the pAKT and pmTOR results. These are also only correlations seen after 2-3 days in culture, which could possibly result from the differentiation of the cells or because of other secondary causes.

(iii) Furthermore, the idea that EGFR signaling should suppress Treg differentiation or functionality seems to be at odds with previous prominent publications showing that EGFR signaling promotes rather than suppresses Treg function [e.g., Zaiss et al., 2013, Immunity (256 citations on November 28, 2022) and Wang et al., 2016, JBC (74 citations on November 28, 2022)]. This apparent discrepancy with published work needs to be addressed, in the Discussion, at least.

(iv) The acute effect of 6F01 on pAKT levels shown in Supplementary Fig. 15 is difficult to understand. First, what happened at time 0 which made the pAKT levels go down by more than 50% during the subsequent 60 minutes, irrespective of the treatment given? Second, as I understood it, the treatment here was the transfer of the differentiated cells into wells coated with 6F01. If so, one would imagine that the response to 6F01 would require that the cells first contacted the bottom or sides of the wells. How then can a strong response be recorded already after 1 minute? For most cell types, it takes minutes or hours for the cells to sediment to the bottoms of the wells. Furthermore, except for the rapid dip of 6F01-treated cells at 1 minute, the response curves appear more or less identical, or superimposable on each other. How should that be understood?

(v) It is stated that 6F01 induces the phosphorylation of Smad2/3 followed by the enhanced Foxp3 transcription (lines 306-307). However, no time-resolved data showing this sequence of events is provided.

(vi) As pointed out by the authors, LRIG1 has been shown to enhance BMP signaling, but to not affect TGFb signaling. This is now discussed by the authors but it also warrants further experimental clarification, not the least because the sera used for T cell cultivation

presumably contain physiologically relevant BMP levels. For example, is the antibody used to evaluate pSmad2/pSmad3 levels specific for pSmad2/3 or does it crossreact with BMP-regulated pSmad1/5? Such crossreactivity between pSmad2/3 and pSmad1/5 is common among “specific” pSmad antibodies and this reviewer was not able to find any information for the used antibody in this regard.

8. The Lrig1 mouse strain used remains ambiguously defined. In M&M (lines 374-375) it is stated that B6.129-Lrig1^{tm1.1Hhed} mice were generated as described by Mao et al. (Mao et al., 2018). However, in the Results section (line 125), it is stated that the Lrig1 mice used were the mice generated by Suzuki et al. (Suzuki et al., 2002). Hence, it remains unclear which Lrig1 knockout mouse strain was used.

9. Furthermore, in addition to the identity of the Lrig1 knockout allele used it would be important to know the genetic background of the mice. That is, was the Lrig1 knockout allele back-crossed onto a genetically clean mouse strain and, if so, for how many generations? Or were the mice of a mixed genetic background, and if so, what was the mix?

10. Also in vivo, the mechanism of action for 6F01 remains obscure. For example, it is unclear to what degree 6F01 could (i) function by affecting other cells than the suspected Tregs, or (ii) function by binding to other proteins than Lrig1 (e.g., Lrig2, Lrig3, or other). These issues deserve a discussion or could be resolved by treating mice after adoptive transfers of Lrig1^{+/+} or Lrig1^{-/-} cells into Lrig1^{+/+} or Lrig1^{-/-} mice.

11. The apparent absence of observed 6F01 side effects is promising, indeed. Nevertheless, possible side effects based on previous observations in knockout mice should be discussed to give a more complete picture. Furthermore, the statement that “Lrig1-null mice did not have any abnormalities” (lines 375-376) has to be specified – what were the features that were analyzed and found not to be abnormal?

12. It is stated that the cells were cultured in 100 ug/ml penicillin/streptomycin (lines 383 and 387). This information is ambiguous because penicillin and streptomycin are two different compounds. Were they both at 100 ug/ml or was it 100 ug/ml in total, and if so, how were the proportions between the two compounds?

13. I was unable to find information about the BDsprint cDNA synthesis kit online and did not understand why SYBR green was included during cDNA synthesis (line 403)?

Title: Lrig1 enriched in the suppressive population of CD4⁺ cells is essential for alleviating autoimmunity via the Smad2/3/Foxp3 axis

We appreciate your excellent review of our manuscript. Your and reviewer #2's valuable comments helped us to make a better manuscript. Significant changes were made or added in the revised manuscript to answer reviewer #2's comments. Please find our explanations and answers that may hopefully resolve the issues raised by reviewer #2.

REVIEWER COMMENTS

Reviewer #2 (Remarks to the Author):

This remains an interesting study. Many of the concerns previously raised by me have been addressed adequately; however, some major and minor issues remain to be addressed.

MAJOR AND MINOR COMMENTS:

(Comment 2.1) Fig. 1b, c, and elsewhere, the authors discuss Lrig1 levels, although the data rather show the fractions of Lrig1-positive cells. Percentage of Lrig1-positive cells should not be confused with Lrig levels.

Response 2.1 =>

We appreciate the Reviewer's comment on the confusion of the unit definition in the figure and manuscript. As the Reviewer suggested, we changed "Lrig1 protein levels on the cell surface" into "the percentage of Lrig1⁺ or Lrig1⁻ population" in the figure and manuscript.

Amendments in the manuscript:

In Introduction section, Page 5, Line 38

In Results section, Page 7, line 58-62, 65-67, 70-71, 76-77

In Figures section, Page 42, line 901, 902

In Supplementary Figures section, Page 54, line 1003, 1004; Page 55, line 1016

(Comment 2.2) The authors claim that Lrig1 "defines", is "specific" for, or "essential" for Treg cells. These claims are confusing given that Lrig1 is expressed by many other cell types in addition to the Tregs, and not all Tregs express Lrig1. Hence, it is suggested that less confusing expressions are used, indicating that enrichment and levels are the defining features, rather than suggesting absolute specificity or dependency.

Response 2.2 =>

We thank the Reviewer's valuable comment on Lrig1 specificity and expression in the manuscript. As you understand, we found that Lrig1-expressing T_{reg} cells had a higher immunosuppressive function than Lrig1-non-expressing T_{reg} cells. This evidence was also confirmed through mRNA sequencing, and Lrig1-expressing CD4⁺ T cells expressed higher levels of immunosuppressive markers too. Therefore, we suggested that Lrig1 is a surface marker that defines the suppressive population of CD4⁺ cells. However, we also agree with the Reviewer's comment that the expression 'defines' or 'specific' could be confused because Lrig1 is also expressed in the other cell types. We now use less confusing representations of the specificity of Lrig1⁺ CD4⁺ T cells and change the title of this manuscript to "Lrig1 enriched in the suppressive population of CD4⁺ cells is essential for alleviating autoimmunity via the Smad2/3/Foxp3 axis".

Amendments in the manuscript:

In Abstract section, Page 3

In Introduction section, Page 6, line 51

In Results section, Page 7, line 56, 58; Page 8, line 80, 83, 100; Page 9, line 110

In Discussion section, Page 15, line 257; Page 16, line 266, 279; Page 20, line 374

In Figures section, Page 42, line 899; Page 44, line 914

(Comment 2.3) Fig. 2c, e, f. What does “Effector only” at the different effector:regulatory ratios mean? Most of these “Effector onlys” show 80-90% proliferation. 80-90% in relation to what – what does 100% proliferation represent?

Response 2.3 =>

We appreciate the Reviewer’s comment. As you understand from the Methods section protocol, i) “Effector only” means that only effector T cells were added in the tissue culture well, providing a positive control for proliferating T cells. The well with different Effector T cell : Regulatory T cell ratio contains the same number of effector T cells plus the different numbers of regulatory T cells. ii) Before co-culture of effector T cells with the different numbers of T_{reg} cells, we first labeled the effector cells using eBioscience Cell Proliferation Dye eFluor 670 (Cat. 65-0840-85). Because the dye is evenly diluted into both daughter cells during the cell division, each division round can be monitored individually over time as an independent peak. Therefore, as the effector cell proliferates, the proliferation dye dilutes, resulting in several low-intensity picks during FACS analysis.

The first peak with the highest intensity is considered to be the population without proliferation, and the subsequent lower intensity peaks are judged as proliferating cell population with cell divisions. In this way, we quantified the level of cell proliferation, adding up the total intensity peaks except for the first highest intensity peak (non-proliferating population) by flow cytometry analysis. Therefore, “100% proliferation” means “the sum of all intensity peaks including the first highest peak, and “the proliferation % of effector only (80-90%)” is “100% proliferation minus the first highest peak % (non-proliferating population)”.

(Comment 2.4) Fig. 3a, b. The Lrig1 siRNA reduced the fraction of Lrig1-positive CD4+ T cells and the proliferation of the effector cells. However, siRNAs are known to have prominent off-target effects. To account for the possible off-target effects, at least two different siRNAs against the target mRNA should be used. I apologize for not having noticed this shortcoming in my original review.

Response 2.4 =>

We appreciate the Reviewer’s comment. We also agree with the comment to include at least two different siRNAs against the target mRNA to avoid possible off-target effects. In the revised Fig. 3a and 3b, the results using the different and second siRNA targeting Lrig1 or a scrambled siRNA (Cat No: AM4611, AM4613 for scrambled siRNA and AM16708, 4390771 for Lrig1 siRNA from Thermo Fisher) were added. Using two different siRNA, we confirmed that there is no off-target effect by the siRNAs targeting Lrig1, and these siRNAs show similar knock-down effects for Lrig1 protein expression. Also, the decreased level of Lrig1 expression on the CD4⁺ T cell surface by two different siRNAs causes lower suppressive activity.

Revised Fig. 3a, b

Amendments in the manuscript:

In Methods section, Page 28, line 534-535

In Figures section, Page 46, line 931

(Comment 2.5) It is claimed that “the body weight... [was] ...unchanged (Fig. 3f, Supplementary Fig. 6a)” (lines 132-133). However, I do not think any body weight data are presented in these figures or elsewhere.

Response 2.5 =>

We appreciate the Reviewer’s point. The “spleen weight” was misused by mistake by “the body weight” in the figures. We changed “the body weight” to “the spleen weight” in the manuscript.

Amendments in the manuscript:

In Results section, Page 10, line 133

(Comment 2.6) It is claimed that “...the level of various T cell subsets in the spleen were unchanged (Fig. 3f, Supplementary Fig. 6a)” (lines 132-133). Here, the authors analyzed IFN γ , IL-4, IL-17A, and Foxp3 in the CD4+ splenocytes in three Lrig1+/+ mice and three Lrig1-/- mice. According to Fig. 3f, the Lrig1-/- mice show a downregulation of IL-4+ cells by approximately 70% and of IL-17A+ cells by approximately 25%. Nevertheless, based on a “two-way ANOVA with Tukey’s multiple comparisons test”, the authors conclude that there was no difference between the genotypes. However, in my opinion, this is not an appropriate statistical test for comparing means of two different populations (Lrig1+/- vs. Lrig1-/-). Hence, with appropriate statistical testing, e.g., t-tests, it would not be surprising if the Lrig1-/- mice actually show significant changes in the T cell subset populations.

Response 2.6 =>

We appreciate the Reviewer’s comment. As the Reviewer suggested, we have performed the two-tailed unpaired Student t-test to compare the results from two different samples in Fig. 3f. We concluded that there is no significant difference among various CD4+ T cell populations in Lrig1^{+/-} and Lrig1^{-/-} mice. In the Figure 3 legend, we provided the statistical information for Fig. 3f. The exact p-values are; IFN γ ⁺ - p= 0.7209, IL-4⁺ - p=0.132, IL-17A⁺ - p= 0.1007, Foxp3⁺ - p= 0.1329. In addition to Fig. 3f result, we have shown that the differentiation potential of naïve CD4+ T cells from Lrig1 WT or KO mouse to various T cell subsets is similar in Fig. 3e. Taking these results together, Lrig1 deficiency doesn’t affect T cell subset polarization and balance.

(Comment 2.7) The mechanistic claim that Lrig1 should induce Tregs via the downregulation of EGFR, resulting in reduced pAKT signaling, which in turn would enhance TGF β -pSmad2/3 signaling that drives the differentiation of the Tregs remains speculative and is not firmly supported by the data.

(i) First, the whole model seems to be mostly based on data obtained by cultivating the T cells on a surface coated with the Lrig1 antibody 6F01. There is no evidence suggesting that this stimulus corresponds to a physiological signal. This follows from the fact that there is no known physiological ligand for Lrig1 and in fact very little is known about the actual molecular function of Lrig1. Hence, plastic surface-coated antibodies might indeed stimulate the physiological function of Lrig1, but it may equally well inhibit its function or do something completely different. Hence, it cannot be concluded that this artificial stimulus “stimulates the function of Lrig1”. It also follows that one cannot assume that 6F01 is agonistic, because it might equally well be antagonistic or something else.

Response 2.7 (i) =>

We appreciate the Reviewer’s comment. Previous studies have found that Lrig1 binds to EGFR in several cancer cell types and mediates EGFR degradation by recruiting c-Cbl to the EGFR cytosolic domain¹⁻³. Based on these previous findings about Lrig1/EGFR/c-Cbl, our manuscript initially focused on the new functions of Lrig1 in T_{reg} cells. We demonstrated that Lrig1-targeting monoclonal antibody (6F01) treatment increases the degree of EGFR degradation compared to isotype IgG control in T_{reg} cells, thereby increasing pSmad2/3 level and Foxp3 expression in vitro condition. The experimental strategy using plastic surface-coated antibody is a well-established method in immunological conditions to induce the oligomerization of the target surface protein. In addition to in vitro results, we also showed that 6F01 treatment alleviates EAE symptoms by elevating the population of Lrig1⁺ Foxp3⁺ T_{reg} cells in Fig. 6.

However, as the Reviewer mentioned, there is no known physiological ligand, and very little is known about the actual molecular function of Lrig1 in Treg cells, and it cannot be concluded that 6F01-mediated stimulation might not be agonistic and might be artificial.

Among the mAb specific to Lrig1 we generated, there is another agonist-like mAb (1C07), which may recognize the similar epitope in Lrig1, and two antagonist-like mAbs (K12 and K46) which may recognize the different epitope in Lrig1. In our new and unpublished data below, two agonist-like mAbs (6F01 and 1C07) induced EGFR degradation as compared to isotype IgG control in iT_{reg} cells differentiated from naïve T cells. At the same time, two other antagonist-like mAbs (K12 and K46) inhibited the EGFR degradation, increasing EGFR level in iT_{reg} cells. In addition to the level of EGFR change, we confirmed the reverse correlation of EGFR level and Foxp3/p-Smad2/3 levels in T_{reg} cells between these mAbs (See below data). The stimulation of T_{reg} cells by agonist-like mAbs (6F01/1C07) showed a lower level of EGFR and an increased level of Foxp3 and phosphorylated Smad2/3 as compared to isotype IgG control. On the other hand, stimulation of T_{reg} cells with antagonist-like K12 or K46 mAb increased EGFR protein and showed a similar or decreased levels of Foxp3 and p-Smad2/3 compared to isotype IgG treatment. The antagonistic function of K46 mAb was also tested in a mouse tumor model (unpublished data) in that the tumor size of K46-treated mice was reduced, and the population of CD4⁺Foxp3⁺ T_{reg} cells decreased among tumor-infiltrated lymphocytes. Taking these results together, we assume that 6F01 is specific to Lrig1 functions as an agonistic antibody toward the increase of T_{reg}-ness by induction of EGFR degradation.

Data not shown in the manuscript

(ii) Second, regarding the downregulation of EGFR after 2-3 days of cultivation, this represents a correlation rather than implying causation. The fact that EGF did not affect the differentiation of the Tregs may rather suggest (but not prove) that regulation of EGFR is not involved. The same argument holds for the pAKT and pmTOR results. These are also only correlations seen after 2-3 days in culture, which could possibly result from the differentiation of the cells or because of other secondary causes.

Response 2.7 (ii) =>

We appreciate the Reviewer's comment. In our experimental design, the induced level of Foxp3 and Lrig1 is highest at 2-3 days after differentiation induction from naïve T cells to iT_{reg} cells. This is why we measured EGFR level and AKT/mTOR phosphorylation on day 2 or 3 after differentiation into T_{reg} cells. Also, it has been well-known from previous studies that EGFR transmits a signal through the AKT/mTOR pathway by phosphorylation^{4,5}.

As shown in Response 2.7 (i) as the new results above, among Lrig1-specific antibodies, 6F01/1C07 are agonistic mAbs that increase T_{reg} cell function by inducing EGFR degradation, and K12/K46 are antagonistic mAbs that inhibit EGFR degradation and decrease T_{reg} cell function *in vitro* and *in vivo*. Therefore, 6F01 induces degradation of EGFR during T_{reg} cell differentiation from naïve T cells, regulates the p-AKT/p-mTOR signaling pathway, and increases Foxp3 expression.

In the publications the Reviewer mentioned in Response 2.7 (iii), it was suggested that EGFR levels in T_{reg} cells increased in tumor or inflammatory microenvironments. However, the level of EGFR in T_{reg} cells is very low in normal condition, and the functional contribution of the signaling through EGFR during T_{reg} cell differentiation is still controversial in other publications. Therefore, we cannot rule out the possibility that other members of receptor tyrosine kinases (RTK) and their ligands functionally similar to EGF/AREG-EGFR may play more dominant roles in the differentiation process of T_{reg} cells.

(iii) Furthermore, the idea that EGFR signaling should suppress Treg differentiation or functionality seems to be at odds with previous prominent publications showing that EGFR signaling promotes rather than suppresses Treg function [e.g., Zaiss et al., 2013, Immunity (256 citations on November 28, 2022) and Wang et al., 2016, JBC (74 citations on November 28, 2022)]. This apparent discrepancy with published work needs to be addressed, in the Discussion, at least.

Response 2.7 (iii) =>

We appreciate the Reviewer's comment that the discrepancy between their works and this issue would be addressed in the Discussion of the revised manuscript.

The reports from Zaiss *et al.*⁶ and Wang *et al.*⁷ are cited and discussed in this manuscript.

Zaiss *et al.* describe EGFR and amphiregulin (AREG) as a component of local regulatory immune responses. This study used *AREG*-deficient mice and found that the *AREG* KO mouse lacks immune regulation. Also, they observed that EGFR is expressed on the T_{reg} cells upon activation, and AREG stimulation enhances T_{reg} suppressive function *in vitro* and *in vivo*. They showed that the AREG-EGFR signal promotes ERK phosphorylation and the transfer of *EGFR*-deficient natural T_{reg} cells into *RAG1* KO mouse is less capable of suppressive function to colitis development compared to the transfer of wild-type T_{reg} cells. However, this report didn't show the phosphorylation of AKT and mTOR. There are controversial reports about the function of phosphorylated ERK and mTOR in T_{reg} cells. It was reported that p-ERK and p-mTOR inhibit the differentiation of T_{reg} cells⁸⁻¹⁰.

It was suggested that EGFR expression was increased in the tumor microenvironment (mouse tumor-infiltrating lymphocytes). Many different types of cells, including T_{reg} cells, expressed multi-functional growth factors and their receptors in this microenvironment. However, it has been known that the splenic T_{reg} cells in normal conditions did not express EGFR.

Wang *et al.* describe that AREG expression and the level of T_{reg} cells increased in lung and gastric cancer patients. The authors showed that AREG is required to maintain T_{reg} cell suppressive activity via EGFR/GSK-3 β /Foxp3 axis in T_{reg} cells known to mediate Foxp3 protein degradation. This report also observed most of the data with T_{reg} cells isolated from tumor tissue-infiltrated lymphocytes. They didn't show the increased level of Foxp3 protein upon AREG stimulation in healthy control T_{reg} cells. Moreover, the Foxp3 protein level is increased by inhibiting the degradation of Foxp3 protein through AREG/EGFR/GSK-3 β signaling in this study.

Our study demonstrated that Lrig1-targeting monoclonal antibody directly stimulates EGFR degradation on T_{reg} cells in normal conditions. And Lrig1-mediated EGFR degradation induces *Foxp3* mRNA level via p-AKT/p-mTOR/p-Foxo1.

In summary, while previous studies provide an advanced understanding of AREG-EGFR signaling in T_{reg} cells in inflammatory or tumor settings, our study provides key functional insights of Lrig1 in T_{reg} cells during T_{reg} cell-differentiation and activation processes through Lrig1/EGFR/Smad2/3/Foxp3 axis. As mentioned in Response 2.7 (i) and (ii), we confirmed that each of the agonist and antagonist mAbs specific to Lrig1 could upregulate or downregulate the differentiation of T_{reg} cells through the opposite functions of EGFR degradation.

The amendments were made in the Discussion section:

"In the immunological environment, CD4⁺ T cells express EGFR, one of the ErbB family, and the expression level of EGFR is higher on T_{reg} cells than that on effector CD4⁺ or CD8⁺ T cells in tumor or inflammatory microenvironment^{6,7}. These findings have shown that EGFR stimulation with amphiregulin (AREG) enhances T_{reg}

suppressive activity via ERK phosphorylation and EGFR/GSK-3 β /Foxp3 axis to inhibit Foxp3 protein degradation in tumor settings. Contrary to previous studies, our study mainly demonstrates another downstream pathway of EGFR in T_{reg} cells during T_{reg} cell differentiation and activation through the Lrig1/EGFR/Smad2/3/Foxp3 axis.

*Amendments in the manuscript:
In Discussion section, Page 18, line 323-330*

(iv) The acute effect of 6F01 on pAKT levels shown in Supplementary Fig. 15 is difficult to understand. First, what happened at time 0 which made the pAKT levels go down by more than 50% during the subsequent 60 minutes, irrespective of the treatment given? Second, as I understood it, the treatment here was the transfer of the differentiated cells into wells coated with 6F01. If so, one would imagine that the response to 6F01 would require that the cells first contacted the bottom or sides of the wells. How then can a strong response be recorded already after 1 minute? For most cell types, it takes minutes or hours for the cells to sediment to the bottoms of the wells. Furthermore, except for the rapid dip of 6F01-treated cells at 1 minute, the response curves appear more or less identical, or superimposable on each other. How should that be understood?

Response 2.7 (iv) =>

We appreciate the Reviewer's comment. i) As you are already familiar with T cell physiology, phosphorylation induction of intracellular signal mediators proximal to the T cell receptor complex is very rapid¹¹. Upon T cell receptor stimulation by ligand or mAb, phosphorylation of these signal mediators peaks within 3-10 minutes after stimulation returning to the basal level rapidly. These responses are much faster in the primary T cells than in the established T cell line or transformed T cell. Akt can also be rapidly dephosphorylated if no activation signal stimulates T cell receptors or EGFR signaling^{12,13}.

As we demonstrated in the manuscript, the differentiated iT_{reg} cells at day 3 were transferred into another well plate coated with isotype IgG or 6F01, and the transferred iT_{reg} cells cannot receive the activation signals through TcR stimulated with anti-CD3/28 antibodies or cytokines, and Akt starts to dephosphorylate very quickly. ii) Technically, we agree with the Reviewer's comment that the cells need to be sedimented to the bottom of the wells to receive a signal from 6F01. To facilitate this process, we usually spin down the cells into the bottom of the coated wells with weak centrifugation at 300 g for 3 min and start to count the incubation time. This step was added to the protocol in the Methods section.

*Amendments in the manuscript:
In Methods section, Page 32, line 628-633*

(v) It is stated that 6F01 induces the phosphorylation of Smad2/3 followed by the enhanced Foxp3 transcription (lines 306-307). However, no time-resolved data showing this sequence of events is provided.

Response 2.7 (v) =>

We appreciate the Reviewer's comment. It is well known that Foxp3 transcription is induced by the phosphorylation of Smad2/3^{14,15}. In addition, we showed that Foxp3 expression did not increase when Smad2/3 phosphorylation was inhibited by Smad2/3 inhibitor (SIS3) when Treg cells were treated with 6F01 (Fig. 5c). Therefore, together with the previous studies, our results reveal the sequence of events of pSmad2/3 and Foxp3.

(vi) As pointed out by the authors, LRIG1 has been shown to enhance BMP signaling, but to not affect TGF β signaling. This is now discussed by the authors but it also warrants further experimental clarification, not the least because the sera used for T cell cultivation presumably contain physiologically relevant BMP levels. For example, is the antibody used to evaluate pSmad2/pSmad3 levels specific for pSmad2/3 or does it crossreact with

BMP-regulated pSmad1/5? Such crossreactivity between pSmad2/3 and pSmad1/5 is common among “specific” pSmad antibodies and this reviewer was not able to find any information for the used antibody in this regard.

Response 2.7 (vi) =>

We appreciate the Reviewer’s point. We have investigated whether 6F01-mediated Lrig1 stimulation could induce the Smad1/5 phosphorylation known to be triggered by Lrig1-BMP stimulation¹⁶. As shown in the figure below, 6F01 treatment didn’t induce the phosphorylation of Smad1/5. However, the level of pSmad2/3 was increased by Lrig1 stimulation through 6F01 mAb treatment (Fig. 6b). Also, we confirmed that TGF- β signaling doesn’t mediate the phosphorylation of Smad1/5 in T_{reg} cells. In addition to this result, we confirmed with the manufacturer that the pSmad2/3 antibody used in our experiments does not have a cross-reactivity with pSmad1/5.

Data not shown in the manuscript

(Comment 2.8) The Lrig1 mouse strain used remains ambiguously defined. In M&M (lines 374-375) it is stated that B6.129-Lrig1tm1.1Hhed mice were generated as described by Mao et al. (Mao et al., 2018). However, in the Results section (line 125), it is stated that the Lrig1 mice used were the mice generated by Suzuki et al. (Suzuki et al., 2002). Hence, it remains unclear which Lrig1 knockout mouse strain was used.

Response 2.8 =>

Thank the Reviewer for checking this confusing information. The mouse strain we used in our study is the same mouse strain described in Suzuki et al. 2002¹⁷. This information is included In the Results and Methods section.

Amendments in the manuscript:
In Methods section, Page 22, line 386

(Comment 2.9) Furthermore, in addition to the identity of the Lrig1 knockout allele used it would be important to know the genetic background of the mice. That is, was the Lrig1 knockout allele back-crossed onto a genetically clean mouse strain and, if so, for how many generations? Or were the mice of a mixed genetic background, and if so, what was the mix?

Response 2.9 =>

Thank the Reviewer for the comment that we have to provide the genetic background of the Lrig1 knockout mouse. The Lrig1 KO mouse we used in our study was back-crossed onto a genetically clean C57BL/6 mouse strain. It does not have any mixed genetic background. Also, we crossed Lrig1 heterozygous mice and used the mice in our study which do not express the Lrig1 gene through gene screening. Whenever a new generation of Lrig1 KO mice is born,

the *Lrig1* KO status is confirmed through gene screening and FACS analysis, but the accurate information about the number of generations still needs to be correctly monitored.

(Comment 2.10) Also in vivo, the mechanism of action for 6F01 remains obscure. For example, it is unclear to what degree 6F01 could (i) function by affecting other cells than the suspected Tregs, or (ii) function by binding to other proteins than Lrig1 (e.g., Lrig2, Lrig3, or other). These issues deserves a discussion or could be resolved by treating mice after adoptive transfers of Lrig1^{+/+} or Lrig1^{-/-} cells into Lrig1^{+/+} or Lrig1^{-/-} mice.

Response 2.10 =>

We appreciate the Reviewer's comment. The sentences about these issues were newly added in the discussion section. We repeatedly confirmed Lrig1 mAb 6F01 doesn't bind to *Lrig1*-null iT_{reg} cells with a similar level of isotype control (See below figure). Also, other Lrig family proteins, such as Lrig2 and Lrig3, are rarely expressed in CD4⁺ T cells, including T_{reg} cells (See below). Therefore, we think 6F01 specifically binds to the Lrig1 protein and functions as T_{reg}-ness agonistic antibody to regulate autoimmune responses.

Amendments in the manuscript:
In Discussion section, Page 19, line 348

Data not shown in the manuscript

(Comment 2.11) The apparent absence of observed 6F01 side effects is promising, indeed. Nevertheless, possible side effects based on previous observations in knockout mice should be discussed to give a more complete picture. Furthermore, the statement that “Lrig1-null mice did not have any abnormalities” (lines 375-376) has to be specified – what were the features that were analyzed and found not to be abnormal?

Response 2.11 =>

We appreciate the Reviewer's comment and agree with the Reviewer that we need to demonstrate a possible side effect of Lrig1-targeting monoclonal antibody (mAb) and what abnormalities we observed in *Lrig1* KO mice. The critical physiological changes and immunological abnormalities have been examined in *Lrig1* knockout mice. There were no physiological abnormalities in *Lrig1* knockout mice except minor bone mineral density and coat texture abnormality. Any abnormal immunological features such as splenomegaly or inflammatory CD4⁺ T cell population in lymph nodes and lungs were not detected. In addition to the fact that *Lrig1*^{-/-} mice did not exhibit any significant immune-related abnormality, our results demonstrated that 6F01 mAb did not show any single dose- and repeated dose-toxicity, which may hold the careful promise of minimal toxicity upon therapeutic application to autoimmunity. Currently, the standard toxicity study in rodents and monkeys is being undertaken with 6F01 mAb produced by the GMP facility.

Amendments in the manuscript:
In Discussion section, Page 19, line 343

(Comment 2.12) It is stated that the cells were cultured in 100 ug/ml penicillin/streptomycin (lines 383 and 387). This information is ambiguous because penicillin and streptomycin are two different compounds. Were they both at 100 ug/ml or was it 100 ug/ml in total, and if so, how were the proportions between the two compounds?

Response 2.12 =>

As the Reviewer commented, we now provide the exact concentrations of penicillin and streptomycin in the solution. This solution contains 10,000 units/mL of penicillin and 10,000 µg/mL of streptomycin, and we added this solution to the culture media to make 100 units/mL of penicillin and 100 µg/mL of streptomycin for the final concentration.

Amendments in the manuscript:
In Methods section, Page 22, line 395

(Comment 2.13) I was unable to find information about the BDsprint cDNA syntesis kit online and did not understand why SYBR green was included during cDNA synthesis (line 403)?

Response 2.13 =>

Thank the Reviewer for the comment. We provided the old version of the kit and method by mistake. We include the new kit information (SMARTer Pico PCR cDNA Synthesis Kit, Takara Bio) in the Methods section and correct the cDNA synthesis and qRT-PCR methods.

Amendments in the manuscript:
In Methods section, Page 23, line 415-417

Reference

- 1 Gur, G. *et al.* LRIG1 restricts growth factor signaling by enhancing receptor ubiquitylation and degradation. *EMBO J.* **23**, 3270-3281 (2004).
- 2 Stutz, M. A., Shattuck, D., Laederich, M., Carraway, K. & Sweeney, C. LRIG1 negatively regulates the oncogenic EGF receptor mutant EGFRvIII. *Oncogene* **27**, 5741-5752 (2008).
- 3 Jeong, D. *et al.* LRIG1-mediated inhibition of EGF receptor signaling regulates neural precursor cell proliferation in the neocortex. *Cell Rep.* **33**, 108257 (2020).
- 4 Freudlsperger, C. *et al.* EGFR–PI3K–AKT–mTOR signaling in head and neck squamous cell carcinomas: attractive targets for molecular-oriented therapy. *Expert opinion on therapeutic targets* **15**, 63-74 (2011).
- 5 Davis, N. M. *et al.* Deregulation of the EGFR/PI3K/PTEN/Akt/mTORC1 pathway in breast cancer: possibilities for therapeutic intervention. *Oncotarget* **5**, 4603 (2014).
- 6 Zaiss, D. M. *et al.* Amphiregulin enhances regulatory T cell-suppressive function via the epidermal growth factor receptor. *Immunity* **38**, 275-284 (2013).
- 7 Wang, S. *et al.* Amphiregulin confers regulatory T cell suppressive function and tumor invasion via the EGFR/GSK-3β/Foxp3 axis. *J. Biol. Chem.* **291**, 21085-21095 (2016).
- 8 Liu, H. *et al.* ERK differentially regulates T h17-and T reg-cell development and contributes to the pathogenesis of colitis. *European journal of immunology* **43**, 1716-1726 (2013).
- 9 Wang, N. *et al.* CD226 attenuates Treg proliferation via Akt and Erk signaling in an EAE model. *Frontiers in Immunology* **11**, 1883 (2020).
- 10 Chapman, N. M. & Chi, H. mTOR signaling, Tregs and immune modulation. *Immunotherapy* **6**, 1295-1311 (2014).
- 11 Yang, L., Qiao, G., Ying, H., Zhang, J. & Yin, F. TCR-induced Akt serine 473 phosphorylation is regulated by protein kinase C-α. *Biochemical and biophysical research communications* **400**, 16-20 (2010).

- 12 De Mesquita, D. D., Zhan, Q., Crossley, L. & Badwey, J. A. p90-RSK and Akt may promote rapid phosphorylation/inactivation of glycogen synthase kinase 3 in chemoattractant-stimulated neutrophils. *FEBS letters* **502**, 84-88 (2001).
- 13 Lučić, I. *et al.* Conformational sampling of membranes by Akt controls its activation and inactivation. *Proceedings of the National Academy of Sciences* **115**, E3940-E3949 (2018).
- 14 Shi, J.-Y. *et al.* FOXP3 Is a HCC suppressor gene and Acts through regulating the TGF- β /Smad2/3 signaling pathway. *BMC cancer* **17**, 1-10 (2017).
- 15 Zhang, H. *et al.* TGF- β 1/Smad2/3/Foxp3 signaling is required for chronic stress-induced immune suppression. *Journal of neuroimmunology* **314**, 30-41 (2018).
- 16 Herdenberg, C. *et al.* LRIG proteins regulate lipid metabolism via BMP signaling and affect the risk of type 2 diabetes. *Communications biology* **4**, 1-15 (2021).
- 17 Suzuki, Y. *et al.* Targeted disruption of LIG-1 gene results in psoriasiform epidermal hyperplasia. *FEBS letters* **521**, 67-71 (2002).

REVIEWER COMMENTS

Reviewer #2 (Remarks to the Author):

In the revised manuscript, the authors have made several important improvements and clarifications. It remains a highly interesting study. However, several important issues remain unresolved.

1. In my previous comment 2.6, I expressed doubt about the apparent “non-significance” of the differences between Lrig1+/- and Lrig1-/- mice shown in Figure 3f. Now, the authors provide additional statistical tests in the form of Student’s t-tests, confirming the lack of significant differences. Nevertheless, I remain unconvinced, especially regarding the non-significance of CD4+IL4+ T-cells and CD4+IL17+ T-cells. Does it make sense to state that “the level of various T cell subsets in the spleen were unchanged” when we observe an approximately 73% downregulation in the proportion of CD4+IL4+ T cells? Is it a power problem (only three experiments)? Or, maybe part of my confusion stems from their use of SEM rather than SD or them not showing all the data points for the three independent experiments? In any case, and because of my confusion, it would be desirable to get access to the actual results of the three experiments, to enable an independent statistical evaluation.

2. Regarding my previous comment 2.7, “The mechanistic claim that Lrig1 should induce Tregs via the downregulation of EGFR, resulting in reduced pAKT signaling, which in turn would enhance TGFb-pSmad2/3 signaling that drives the differentiation of the Tregs remains speculative and is not firmly supported by the data”, the criticism remains unresolved: (i) Whereas the authors provide confirmatory data suggesting an involvement of TGFb-pSmad2/3 signaling, through the use of a specific Smad3 inhibitor (SIS3), the corresponding confirmation for an involvement of EGFR, is lacking. If their model is correct, the inhibition of EGFR should yield the same result as the treatment with the Lrig1 antibody 6F01. If this is the case, it could easily be confirmed through the use of a specific EGFR inhibitor such as gefitinib or erlotinib, thereby supporting an involvement of EGFR signaling in the pathway. (ii) Furthermore, the model summarized in supplementary Figure 16 suggests that the downregulation of EGFR protein levels is the cause of the observed

reduction in pAKT levels. However, whereas the reduction in pAKT is seen already after 1 minute (supplementary Figure 15), the downregulation of EGFR is demonstrated after 2-3 days (Figure 5e). If the downregulation of EGFR would be the cause of the reduction in pAKT levels, then the downregulation of EGFR should be present prior to, or at the time of, the pAKT effect. Are the EGFR levels downregulated already after 1 minute?

3. In the rebuttal, the authors explained why a rapid decline in pAKT levels was seen also in untreated cells (supplementary Figure 15). That is, if I understood them right, because of the withdrawal of CD3/28 stimulation. It would be helpful for the readers if this information was included in the figure legend.

4. Comment 2.9, regarding the genetic background of the Lrig1 knockout mice, the authors explained that the background was a clean C57BL/6 strain. However, the specific C57BL/6 substrain should be indicated. Second, this information should be clearly indicated in the manuscript.

5. My previous comment 2.11, regarding possible side effects of the Lrig1 treatment, is not discussed properly. If anti-Lrig1 treatment is supposed to be given to humans, scientific honesty calls for a thorough discussion about possible side effects based on the current knowledge. In my opinion, at least three aspects should be discussed. First, there are numerous reports suggesting or showing that Lrig1 functions as a tumor suppressor in different tissues (e.g., Powell et al., 2014; Mao et al., 2018; and more). How may the anti-Lrig1 treatment affect the tumor suppressive function of Lrig1? Or is the tumor suppressive function of Lrig1 irrelevant here? Second, it was recently shown that LRIG1 gene variants are associated with the risk of developing type 2 diabetes in humans (Herdenberg et al., 2021). Could the anti-LRIG1 treatment affect this risk? Third, there are several reports showing that Lrig1 regulates stem cell quiescence (reviewed in Herdenberg and Hedman, 2023). How may the anti-Lrig1 treatment affect these different stem cell compartments?

Title: Lrig1 enriched in the suppressive population of CD4⁺ cells is essential for alleviating autoimmunity via the Smad2/3/Foxp3 axis

We appreciate your excellent review of our manuscript. Your and reviewer #2's valuable comments helped us to make a better manuscript. Significant changes were made or added to the revised manuscript to answer reviewer #2's comments. Please find our explanations and answers that may hopefully resolve the issues raised by reviewer #2.

REVIEWER COMMENTS

Reviewer #2 (Remarks to the Author):

In the revised manuscript, the authors have made several important improvements and clarifications. It remains a highly interesting study. However, several important issues remain unresolved.

MAJOR AND MINOR COMMENTS:

(Comment 2.1) In my previous comment 2.6, I expressed doubt about the apparent “non-significance” of the differences between Lrig1^{+/+} and Lrig1^{-/-} mice shown in Figure 3f. Now, the authors provide additional statistical tests in the form of Student's t-tests, confirming the lack of significant differences. Nevertheless, I remain unconvinced, especially regarding the non-significance of CD4⁺IL4⁺ T-cells and CD4⁺IL17⁺ T-cells. Does it make sense to state that “the level of various T cell subsets in the spleen were unchanged” when we observe an approximately 73% downregulation in the proportion of CD4⁺IL4⁺ T cells? Is it a power problem (only three experiments)? Or, maybe part of my confusion stems from their use of SEM rather than SD or them not showing all the data points for the three independent experiments? In any case, and because of my confusion, it would be desirable to get access to the actual results of the three experiments, to enable an independent statistical evaluation.

Response 2.1 =>

We appreciate the Reviewer's comments and concerns. As the reviewer commented, we also thought that the issue about the non-significance of this result might be due to the small size of the experiments (3 samples). To resolve this issue, we analyzed the level of CD4⁺IFN γ ⁺, CD4⁺IL-4⁺, CD4⁺IL-17A⁺, or CD4⁺Foxp3⁺ T cells in the spleen of 6 more Lrig1 KO and heterozygous mice and performed the two-tailed Student's t-test. As shown below, statistical significance was not shown in all four T cell populations between Lrig1^{+/+} and Lrig1^{-/-} mice in the new 6 samples and the combined 9 samples with the previous 3 samples. The actual results of all the experiments are included. Therefore, the level of various T cell subsets in the spleen of Lrig1 KO was unchanged. The original Fig.3f was replaced with the revised graph containing 9 data points. Also, we added the exact p-value in the revised figure legend of Fig. 3f; IFN γ ; P=0.9341, IL-4; P=0.1721, IL-17A; P=0.0934, Foxp3; P=0.2866.

Data of 6 newly analyzed mice (Data now shown in the manuscript)

Revised Fig. 3f

Raw data of Fig. 3f (Data now shown in the manuscript)

	IFN γ ⁺		IL-4 ⁺		IL-17 ⁺		Foxp3 ⁺	
	Lrig1 ^{+/-}	Lrig1 ^{-/-}	Lrig1 ^{+/-}	Lrig1 ^{-/-}	Lrig1 ^{+/-}	Lrig1 ^{-/-}	Lrig1 ^{+/-}	Lrig1 ^{-/-}
Added data	0.75	1.58	3.48	0.84	3.78	2.53	18	13.4
	1.15	2.2	0.95	0.41	2.58	2.38	19.9	17.7
	2.07	0.83	1.81	0.72	3.24	2.46	16.5	13.7
	1.83	2.07	1.69	1.68	4.18	3.73	10.9	9.21
	1.88	2.01	1.48	1.91	4.19	3.4	10.9	9.25
	1.83	1.57	3.62	2.3	3.1	3.93	11.6	10.6
	1.84	1.32	2.97	2.81	3.68	3.71	11.2	10.2
	1.67	1.48	1.65	1.59	3.77	3.19	10.5	11
	1.69	1.5	1.7	1.81	3.96	2.85	12.8	12.4

Amendments in the manuscript:

In Figure Legends section, Page 49, line 1029-1032

(Comment 2.2) Regarding my previous comment 2.7, “The mechanistic claim that Lrig1 should induce Tregs via the downregulation of EGFR, resulting in reduced pAKT signaling, which in turn would enhance TGF β -pSmad2/3 signaling that drives the differentiation of the Tregs remains speculative and is not firmly supported by the data”, the criticism remains unresolved:

(i) Whereas the authors provide confirmatory data suggesting an involvement of TGFb-pSmad2/3 signaling, through the use of a specific Smad3 inhibitor (SIS3), the corresponding confirmation for an involvement of EGFR, is lacking. If their model is correct, the inhibition of EGFR should yield the same result as the treatment with the Lrig1 antibody 6F01. If this is the case, it could easily be confirmed through the use of a specific EGFR inhibitor such as gefitinib or erlotinib, thereby supporting an involvement of EGFR signaling in the pathway.

Response 2.2 (i) =>

- 1) We thank the Reviewer's comment and suggestion for using the EGFR inhibitors to confirm the involvement of EGFR signaling in the Lrig1 pathway in T_{reg} cells. As shown in the results below (Revised Supplementary Fig. 17), when sub-optimally induced T_{reg} cells were treated with different concentrations of EGFR-specific inhibitors (Gefitinib and Erlotinib) for 1 day, the p-AKT level was reduced by 6F01 treatment (included in Fig. 5f) or EGFR inhibitors. However, EGFR inhibitors treatment did not increase the population of p-Smad2/3⁺ or Foxp3⁺ T cells, which was consistently increased by the 6F01 treatment. These results suggest that EGFR may not be a primary signal mediator in the Lrig1-mediated intracellular signaling pathway in T_{reg} cells. This result is consistent with the result of the previous Comment 2.3 in the first revision to address this issue using the recombinant EGF protein. The data showed that the level of Foxp3 in iT_{reg} cells was not changed by EGF (See below figure).
- 2) The possibility that EGFR may not be a primary signal mediator in the Lrig1-mediated intracellular signaling pathway in T_{reg} cells can be explained by the fact that the level of EGFR is very low on the surface of normal T_{reg} cells. Alternatively, intracellular signaling context and transcriptional factor functional network constellation might be quite different between cancer cells and T_{reg} cells. It may be possible that other or unidentified members of the receptor tyrosine kinase (RTK) family, such as c-MET, AXL, MERTK, or VEGFR, or an unknown surface protein associated with tyrosine kinase, which can activate the AKT-mTOR signaling pathway, can be involved in the Lrig1-mediated intracellular signaling pathway in T_{reg} cells.¹⁻³

Revised Supplementary Fig. 17

Previous Comment 2.3

Amendments in the manuscript:

In Results section, Page 14, line 227-238

In Discussion section, Page 19, line 328-342

(ii) Furthermore, the model summarized in supplementary Figure 16 suggests that the downregulation of EGFR protein levels is the cause of the observed reduction in pAKT levels. However, whereas the reduction in pAKT is seen already after 1 minute (supplementary Figure 15), the downregulation of EGFR is demonstrated after 2-3 days (Figure 5e). If the downregulation of EGFR would be the cause of the reduction in pAKT levels, then the downregulation of EGFR should be present prior to, or at the time of, the pAKT effect. Are the EGFR levels downregulated already after 1 minute?

Response 2.2 (ii) =>

We appreciate the Reviewer's comment. To address this concern, we measured EGFR levels at different time points after stimulation of iT_{reg} cells with the 6F01 antibody. As shown in the result below (Revised Supplementary Fig. 14), downregulation of EGFR level was not observed at 1 min, 5 min, or 15 min after 6F01 stimulation. However, there is an apparent downregulation of EGFR protein at 30 min after 6F01 stimulation. This pattern of EGFR degradation was reported in the previous studies⁴⁻⁶. EGFR degradation by ubiquitination occurs not immediately but rapidly 15-30 min after stimulation. Therefore, Lrig1's function to recruit c-Cbl to the cytoplasmic domain of RTK, such as EGFR or receptor associated with tyrosine kinase, is well maintained in T_{reg} cells, although the level of EGFR is very low in iT_{reg} cells. Upon 6F01 stimulation, Lrig1 induces the binding of c-Cbl to EGFR, and the ubiquitination process will be initiated, and degradation of EGFR could be detected at 30 min after stimulation. Regarding dephosphorylation of p-AKT at the very early time point (at 1 min) of 6F01 stimulation in iT_{reg} cells, when Lrig1 is agonistically stimulated by 6F01, EGFR will undergo significant conformational changes to initiate c-Cbl binding and ubiquitination, thereby terminating the function to phosphorylate AKT immediately. Therefore, the level of p-AKT phosphorylation can decrease at very early time points of 6F01 stimulation in iT_{reg} cells, and degradation of EGFR proceeds accordingly.

Revised Supplementary Fig. 14

Amendments in the manuscript:
In Result section, Page 14, line 217-223

(Comment 2.3) In the rebuttal, the authors explained why a rapid decline in pAKT levels was seen also in untreated cells (supplementary Figure 15). That is, if I understood them right, because of the withdrawal of CD3/28 stimulation. It would be helpful for the readers if this information was included in the figure legend.

Response 2.3 =>

We appreciate the Reviewer's comment. We also agree with the Reviewer's suggestion that we would add the information that the withdrawal of CD3/28 stimulation rapidly induces the dephosphorylation of AKT in primary T cells to the manuscript. We included this information in the Result section as shown below.

“We observed p-AKT starts to dephosphorylate very quickly by 6F01 treatment within 30 min after stimulation (Supplementary Fig. 15). Although no treatment or isotype IgG also induced dephosphorylation of AKT because the transferred iT_{reg} cells could not receive the activation signals through T cell receptor stimulated with anti-CD3/28 antibodies or cytokines, 6F01 treated-iT_{reg} cells showed the quicker dephosphorylation of AKT than no or isotype treatment.”

Amendments in the manuscript:
In Results section, Page 13, line 210-215

(Comment 2.4) Comment 2.9, regarding the genetic background of the Lrig1 knockout mice, the authors explained that the background was a clean C57BL/6 strain. However, the specific C57BL/6 substrain should be indicated. Second, this information should be clearly indicated in the manuscript.

Response 2.4 =>

We appreciate the Reviewer's comment. We now provide the specific C57BL/6N as the substrain of the Lrig1 knockout mice. We added this information in the Methods section.

Amendments in the manuscript:
In Methods section, Page 24, line 434-435

(Comment 2.5) My previous comment 2.11, regarding possible side effects of the Lrig1 treatment, is not discussed properly. If anti-Lrig1 treatment is supposed to be given to humans, scientific honesty calls for a thorough discussion about possible side effects based on the current knowledge. In my opinion, at least three aspects should be discussed. First, there are numerous reports suggesting or showing that Lrig1 functions as a tumor suppressor in different tissues (e.g., Powell et al., 2014; Mao et al., 2018; and more). How may the anti-Lrig1

treatment affect the tumor suppressive function of Lrig1? Or is the tumor suppressive function of Lrig1 irrelevant here? Second, it was recently shown that LRIG1 gene variants are associated with the risk of developing type 2 diabetes in humans (Herdenberg et al., 2021). Could the anti-LRIG1 treatment affect this risk? Third, there are several reports showing that Lrig1 regulates stem cell quiescence (reviewed in Herdenberg and Hedman, 2023). How may the anti-Lrig1 treatment affect these different stem cell compartments?

Response 2.5 =>

We appreciate the Reviewer's comment. We also agree with the comment that proper discussions about possible side effects of anti-Lrig1 antibody treatment should be addressed in the Discussion section. Although 6F01 mAb did not show any single dose- and repeated dose-toxicity in mice from our preliminary data and *Lrig1* KO mice did not show any significant physiological abnormality, there would be possible side effects of anti-Lrig1 monoclonal antibody therapy in humans based on the current knowledge. We demonstrated these potential side effects in the Discussion section.

“Although we have tested the isotype specificity and non-toxicity of the anti-Lrig1 monoclonal antibody, there would be possible side effects of the LRIG1 antibody treatment in humans based on the current knowledge of LRIG1 expression and function in other cell types. 1) Previously, Lrig1 has been well known as a tumor suppressor⁷⁻⁹. When anti-LRIG1 antibody is administered to patients with autoimmune disease, there is a possibility that the antibody binds to LRIG1⁺ cells, such as epithelial or intestinal stem cells in other tissues and may affect the proliferation status of the cells. 2) Previous report has shown that *LRIG1* gene variants have a risk and association with type 2 diabetes¹⁰. Because the functional importance of *Lrig1* in various metabolic diseases such as type 2 diabetes has been explored very recently, it remains to be seen what therapeutic efficacy agonistic stimulation of *Lrig1* would show to type 2 diabetes. 3) In many previous studies, it has been known that *Lrig1* regulates various stem cell quiescence such as epidermal interfollicular stem cells, intestinal stem cells, or neural stem cells¹¹. Although this mechanism has not yet been fully elucidated, one possibility is that agonistic stimulation of *Lrig1* may enhance the sensitization of BMP signal in stem cells for regulating their quiescence status.”

Amendments in the manuscript:
In Discussion section, Page 22, line 404-418

Reference

- 1 Palle, J. *et al.* Targeting HGF/c-Met Axis Decreases Circulating Regulatory T Cells Accumulation in Gastric Cancer Patients. *Cancers* **13**, 5562 (2021).
- 2 Zhao, G.-j. *et al.* Growth arrest-specific 6 enhances the suppressive function of CD4⁺ CD25⁺ regulatory T cells mainly through Axl receptor. *Mediators of Inflammation* **2017** (2017).
- 3 Shin, J.-Y., Yoon, I.-H., Kim, J.-S., Kim, B. & Park, C.-G. Vascular endothelial growth factor-induced chemotaxis and IL-10 from T cells. *Cellular immunology* **256**, 72-78 (2009).
- 4 Carter, R. E. & Sorkin, A. Endocytosis of functional epidermal growth factor receptor-green fluorescent protein chimera. *Journal of Biological Chemistry* **273**, 35000-35007 (1998).
- 5 Huang, F., Kirkpatrick, D., Jiang, X., Gygi, S. & Sorkin, A. Differential regulation of EGF receptor internalization and degradation by multiubiquitination within the kinase domain. *Molecular cell* **21**, 737-748 (2006).
- 6 Zheng, J. *et al.* Folliculin interacts with Rab35 to regulate EGF-induced EGFR degradation. *Frontiers in Pharmacology* **8**, 688 (2017).
- 7 Mao, F. *et al.* Lrig1 is a haploinsufficient tumor suppressor gene in malignant glioma. *Oncogenesis* **7**, 13 (2018).
- 8 Powell, A. E. *et al.* Inducible loss of one Apc allele in Lrig1-expressing progenitor cells results in multiple distal colonic tumors with features of familial adenomatous polyposis. *American Journal of Physiology-Gastrointestinal and Liver Physiology* **307**, G16-G23 (2014).
- 9 Li, Q. *et al.* LRIG1 is a pleiotropic androgen receptor-regulated feedback tumor suppressor in prostate cancer. *Nature communications* **10**, 5494 (2019).
- 10 Herdenberg, C. *et al.* LRIG proteins regulate lipid metabolism via BMP signaling and affect the risk of type 2 diabetes. *Communications Biology* **4**, 90 (2021).

- 11 Herdenberg, C. & Hedman, H. Hypothesis: Do LRIG proteins regulate stem cell quiescence by promoting BMP signaling? *Stem Cell Reviews and Reports* **19**, 59-66 (2023).

Title: Lrig1 enriched in the suppressive population of CD4⁺ cells is essential for alleviating autoimmunity via the Smad2/3/Foxp3 axis

We appreciate your excellent review of our manuscript. Your and reviewer #2's valuable comments helped us to make a better manuscript. Significant changes were made or added to the revised manuscript to answer reviewer #2's comments. Please find our explanations and answers that may hopefully resolve the issues raised by reviewer #2.

REVIEWER COMMENTS

Reviewer #2 this time haven't had any practical comments to the authors but in the comments to editors section they expressed serious doubts about the proposed signalling mechanism. Since both EGFR inhibition and Lrig-1 stimulation lead to the same effect on Akt phosphorylation but not the same effect on the Treg phenotype, this makes it unlikely that Akt plays a role at all, even if Lrig-1 agonism does affect Akt phosphorylation. You may entirely remove the signal transduction section alongside Fig.5 e-h and Sup Fig. 14-18 and wrap up the story without claiming understanding exactly how Lrig-1 regulates Treg function, or leave in the pAkt data but interpret it as likely irrelevant for the phenotype. Either way, please re-write the abstract, results and discussion sections accordingly.

Response =>

We appreciate the Reviewer's comments and concerns. As the reviewer suggested, we have now amended the manuscript to describe that the reduction of p-AKT by agonistic stimulation of Lrig1 in T_{reg} cells may be likely irrelevant to the Lrig1-mediated functional enhancement of T_{reg} cells.

Amendments in the manuscript:

In Abstract section, Page 3

The sentence

“Furthermore, monoclonal antibody 6F01 specific to Lrig1 significantly improved the symptoms of experimental autoimmune encephalomyelitis (EAE) via the increase of Smad2/3 phosphorylation and the decrease of phosphorylated AKT, mTOR, or FoxO1, and thus inducing the expression of Foxp3 in T_{reg} cell.”
was changed into

“Furthermore, monoclonal antibody 6F01 specific to Lrig1 significantly improved the symptoms of experimental autoimmune encephalomyelitis (EAE) via the increase of Smad2/3 phosphorylation and Foxp3 expression in T_{reg} cell.”

In Introduction section, Page 5, line 44-49

The sentences

“This therapeutic potential through Lrig1 is mediated by the increase of Foxp3⁺ T cells via induction of Smad2/3 phosphorylation leading to Foxp3 expression at the transcription level. Also, we demonstrated that the induction of Smad2/3 phosphorylation or Foxp3 expression by 6F01 stimulation was exerted by the decrease of the phosphorylation of AKT, mTOR, or FoxO1 that act as a suppressor of Smad2/3 phosphorylation or Foxp3 expression.”

was changed into

“This therapeutic potential through Lrig1 is mediated by the increase of Foxp3⁺ T cells via induction of Smad2/3 phosphorylation leading to Foxp3 expression at the transcription level.”

In Results section, Page 14, line 229-234

The sentences

“These results suggest that EGFR may not be a primary signal mediator in the Lrig1-mediated intracellular signaling pathway in T_{reg} cells. This result is consistent with our result showing that the level of Foxp3 in iT_{reg} cells was not changed by EGF (data not shown). Together, these findings demonstrate that 6F01-mediated Lrig1 stimulation inhibits AKT signaling, thereby increasing the number of Foxp3⁺ T cells through the induction of Smad2/3 phosphorylation and subsequent Foxp3 transcription independent of EGFR degradation in T_{reg} cells.”

“These results suggest that EGFR and p-AKT may not be a primary signal mediator in the Lrig1-mediated intracellular signaling pathway in T_{reg} cells. This result is consistent with our result showing that the level of Foxp3 in iT_{reg} cells was not changed by EGF (data not shown). These findings demonstrate that 6F01-mediated Lrig1 stimulation can induce EGFR degradation and reduce p-AKT/mTOR signaling. However, reduction of p-AKT may be likely irrelevant to the increase in phosphorylation of Smad2/3 and the expression of Foxp3.”

In Discussion section, Page 19, line 321-322

The sentence

“Agonistic T_{reg}-ness activation of Lrig1 by 6F01 mAb induces the phosphorylation of Smad2/3, followed by the enhanced Foxp3 transcription through AKT signaling.”

was changed into

“Agonistic T_{reg}-ness activation of Lrig1 by 6F01 mAb induces the phosphorylation of Smad2/3, followed by the enhanced Foxp3 transcription.”

In Discussion section, Page 19, line 335-338

The sentence

“It may be possible that other or unidentified members of the receptor tyrosine kinase (RTK) family, such as c-MET, AXL, MERTK, or VEGFR, or an unknown surface protein associated with tyrosine kinase, which can activate the AKT-mTOR signaling pathway, can be involved in the Lrig1-mediated intracellular signaling pathway in T_{reg} cells.”

was changed into

“It may be possible that other or unidentified members of the receptor tyrosine kinase (RTK) family, such as c-MET, AXL, MERTK, or VEGFR, or an unknown surface protein associated with tyrosine kinase, can be involved in the Lrig1-mediated intracellular signaling pathway in T_{reg} cells.”

In Discussion section, Page 20, line 355-366

The sentences

“Contrary to previous studies, our study mainly demonstrates the new signaling pathway in T_{reg} cells during T_{reg} cell differentiation and activation through the Lrig1/RTKs/Smad2/3/Foxp3 axis. AKT, a downstream signal mediator of EGFR or RTKs, is known to interact with Smad3 and inhibit its phosphorylation, thereby suppressing the TGF- β receptor-mediated signaling. AKT can also inhibit the functions of FoxO protein essential for Foxp3 expression and activate the mTORC1 pathway to suppress the induction of Foxp3 (Supplementary Fig. 18a). Based on our results, the molecular mechanism to explain how Lrig1 is driving T_{reg}-ness or suppressive function is suggested in Supplementary Fig. 18. In T_{reg} cells without Lrig1 stimulation, EGF or other functional equivalents in the immunological microenvironment activates unknown RTK or its family members, which suppresses the expression of Foxp3 via functional inhibition of FoxO and Smad2/3. Upon stimulation of Lrig1, AKT phosphorylation is inhibited by the degradation of unknown RTK on T_{reg} cells, and it sequentially suppresses the phosphorylation of mTOR. Thereby, Foxp3 expression can be induced by lifting the AKT/mTOR-mediated functional suppression of Smad2/3 and FoxO (Supplementary Fig. 18b).”

were changed into

“AKT, a downstream signal mediator of EGFR or RTKs, is also known to play an essential role in T cell development or activation. However, AKT interacts with Smad3 and inhibits its phosphorylation, thereby suppressing the TGF- β receptor-mediated signaling. AKT can also inhibit the functions of FoxO protein essential for Foxp3 expression and activate the mTORC1 pathway to suppress the induction of Foxp3. The results from our study demonstrate that agonistic stimulation of Lrig1 in T_{reg} cells induced the degradation of EGFR and reduction of p-AKT. However, EGFR-specific inhibitors reduced the p-AKT level, but the population of p-Smad2/3⁺ or Foxp3⁺ T cells was not increased. Therefore, the reduction of p-AKT by agonistic stimulation of Lrig1 in T_{reg} cells may be likely irrelevant to the Lrig1-mediated functional enhancement of T_{reg} cells. Accordingly, the intracellular signaling pathway for the increase of p-Smad2/3 and Foxp3 by Lrig1 stimulation in T_{reg} cells needs to be further elucidated.”

Supplementary Fig. 18 was removed.

REVIEWERS' COMMENTS

Reviewer #2 (Remarks to the Author):

This very interesting manuscript has improved considerably. My remaining (minor) points are as follows:

1. Now, when the authors admit that the EGFR > pAKT pathway is probably not what drives the Treg stimulatory activity of LRIG1, the discussion about EGFR, ERBBs, RTKs, and AKT on page 19 and 20 (lines 376-382 and 387-408) is far too long. This could instead be summarized in a couple of sentences, I suggest.

2. The speculation about EGFR conformations and pAKT dephosphorylation (lines 258-264) remains a free fantasy, without any experimental support, and should be removed from the Results.

3. In the Results it is stated that “We observed p-AKT starts to dephosphorylate very quickly by 6F01 treatment within 30 min after stimulation (Supplementary Fig. 15)” (lines 251-253). As far as I can see, dephosphorylation starts within 1 min, rather than within 30 min. Hence, the text should be revised accordingly.

4. Apparently, LRIG1 was important for the studied Treg functions. However, this does not mean that these functions were completely dependent on LRIG1, as evident from the fact that also LRIG1-low/negative Tregs showed some activity, albeit at reduced levels. Therefore, definitive words like “essential” (lines 43, 149, 162), “abrogates” (line 152), “depend” (although in a different context, line 411), and “required” (line 458) should be avoided and replaced with more moderate words, such as “important”, “impair”, “regulate”, etc.

5. Similarly, the last sentence in the Discussion (lines 463-463), stating that the targeting of LRIG1 “...will provide a new therapeutic regime for treating autoimmunity” is too strong and should be modified to “...may provide a new therapeutic...”, or similar.

6. A last minor comment regarding the possible side effects listed on lines 442-454. Indeed, it is important to discuss the potential side effects, however, because we do not know whether the antibody is “agonistic” or “antagonistic” one could maybe also speculate that the antibody might have beneficial effects as well as detrimental effects; i.e., maybe also function as an agent against cancer and type 2 diabetes?

Title: Lrig1-expression confers suppressive function to CD4⁺ cells and is essential for averting autoimmunity via the Smad2/3/Foxp3 axis

We appreciate your excellent review of our manuscript. Your and reviewer #2's valuable comments helped us to make a better manuscript. Significant changes were made or added to the revised manuscript to answer reviewer #2's comments. Please find our explanations and answers that may hopefully resolve the issues raised by reviewer #2.

REVIEWERS' COMMENTS

Reviewer #2 (Remarks to the Author):

This very interesting manuscript has improved considerably. My remaining (minor) points are as follows:

MINOR COMMENTS:

(Comment 2.1) Now, when the authors admit that the EGFR > pAKT pathway is probably not what drives the Treg stimulatory activity of LRIG1, the discussion about EGFR, ERBBs, RTKs, and AKT on page 19 and 20 (lines 376-382 and 387-408) is far too long. This could instead be summarized in a couple of sentences, I suggest.

Response 2.1 =>

We appreciate the Reviewer's comment. We removed unnecessary sentences including the discussion about EGFR family and RTKs.

Amendments in the manuscript:
In Discussion section, Page 20 line 336-348

(Comment 2.2) The speculation about EGFR conformations and pAKT dephosphorylation (lines 258-264) remains a free fantasy, without any experimental support, and should be removed from the Results.

Response 2.2 =>

We appreciate the Reviewer's comment. We now removed the sentences from the Results section.

(Comment 2.3) In the Results it is stated that "We observed p-AKT starts to dephosphorylate very quickly by 6F01 treatment within 30 min after stimulation (Supplementary Fig. 15)" (lines 251-253). As far as I can see, dephosphorylation starts within 1 min, rather than within 30 min. Hence, the text should be revised accordingly.

Response 2.3 =>

We appreciate the Reviewer's comment. We also agree with the Reviewer's suggestion and the stated time was amended from 30 min to 1 min.

Amendments in the manuscript:
In Result section, Page 13 line 208

(Comment 2.4) Apparently, LRIG1 was important for the studied Treg functions. However, this does not mean

that these functions were completely dependent on LRIG1, as evident from the fact that also LRIG1-low/negative Tregs showed some activity, albeit at reduced levels. Therefore, definitive words like “essential” (lines 43, 149, 162), “abrogates” (line 152), “depend” (although in a different context, line 411), and “required” (line 458) should be avoided and replaced with more moderate words, such as “important”, “impair”, “regulate”, etc.

Response 2.4 =>

We appreciate the Reviewer’s comment. To address this concern, we now replaced the words “essential”, “abrogates”, “depend”, and “required” to the words “important”, “impair”, and “regulate” in the manuscript.

Amendments in the manuscript:

In Introduction section, Page 5 line 46

In Result section, Page 8 line 87

In Result section, Page 9 line 105, 108, 118

In Result section, Page 10 line 139

In Discussion section, Page 18 line 304

In Discussion section, Page 22 line 398

(Comment 2.5) Similarly, the last sentence in the Discussion (lines 463-463), stating that the targeting of LRIG1 “...will provide a new therapeutic regime for treating autoimmunity” is too strong and should be modified to “...may provide a new therapeutic...”, or similar.

Response 2.5 =>

We appreciate the Reviewer’s comment. To address this concern, we now replaced the sentence to “Therefore, the reagents targeting Lrig1 protein on the surface of the immuno-suppressive T cell population may provide a new therapeutic regime for treating autoimmunity.”

Amendments in the manuscript:

In Result section, Page 22 line 404

(Comment 2.6) A last minor comment regarding the possible side effects listed on lines 442-454. Indeed, it is important to discuss the potential side effects, however, because we do not know whether the antibody is “agonistic” or “antagonistic” one could maybe also speculate that the antibody might have beneficial effects as well as detrimental effects; i.e., maybe also function as an agent against cancer and type 2 diabetes?

Response 2.6 =>

We appreciate the Reviewer’s comment. We agree with the Reviewer’s comment that the function of antibody is still unknown as “agonistic” or “antagonistic”. Therefore, we added some sentences to address this point in the Discussion section and also changed “agonistic mAb” to “Treg cell-stimulating mAb specific to Lrig1” or “Lrig1 mAb stimulating Treg cell functions”.

Amendments in the manuscript:

In Discussion section, Page 21 line 381- Page 22 line 394